# VERIFICATION AND CO-ALIGNMENT VIA HETEROGENEOUS CONSISTENCY FOR PREFERENCE-ALIGNED LLM ANNOTATIONS

**Cheng Chen**[1,2,3]**, Haiyan Yin**[1,2]**, Ivor W. Tsang**[1,2,4]

[1]Centre for Frontier AI Research, Agency for Science, Technology and Research, Singapore
[2]Institute of High Performance Computing, Agency for Science, Technology and Research, Singapore
[3]Australian Artificial Intelligence Institute, University of Technology Sydney, Australia
[4]Nanyang Technological University (NTU), Singapore
`Cheng.chen-16@student.uts.edu.au`, `{yin_haiyan,ivor_tsang}@a-star.edu.sg`

## ABSTRACT

Large Language Models (LLMs) are increasingly expected to be culturally customisable and personally aligned for natural language understanding (NLU). However, existing methods, from supervised fine-tuning (SFT) to personalised RLHF and prompting, either require costly large-scale annotations or remain constrained by their pretraining distributions. Moreover, acquiring annotations that reflect subjective, diverse, and evolving user preferences is both expensive and labour-intensive. To address these limitations, we propose *Heterogeneous-Consistency Co-Alignment* (HCC), a training-free annotation paradigm that leverages two heterogeneous models: a knowledge-rich yet potentially overconfident LLM and a task-specialised lightweight model guided by a small user preference set. Together, they verify and co-align misaligned outputs over unlabelled corpora. For verification, HCC introduces the reference-free *Consistent-And-Inconsistent* (**CAI**) Ratio, an uncertainty signal derived from inter-model agreements (consistent samples) and disagreements (inconsistent samples) to determine whether refinement is necessary. For co-alignment, HCC employs a non-parametric, embedding-based preference assignment scheme to recalibrate inconsistent samples according to user preferences. Across eight NLU datasets and both open- and closed-source LLMs, HCC consistently improves annotation alignment and, in several tasks, enables *Llama-3-8B* to surpass *GPT-3.5/4o-mini* after co-alignment correction. Moreover, CAI strongly correlates with accuracy and tracks pre- and post-alignment gains, offering a reference-free signal for scaling preference-aligned annotation without ground-truth supervision. **Code:** `https://github.com/858006908cc/VERIFICATION-AND-CO-ALIGNMENT-ICLR26`

# 1 INTRODUCTION

Demand is burgeoning for culturally and personally aligned LLMs across diverse natural-language understanding (NLU) applications, including culturally aware language understanding (Nguyen et al., 2024), personalized recommendation (Li et al., 2023), household robotics (Han et al., 2024), and clinical guidance in healthcare (Kadariya et al., 2019). In Southeast Asia, for example, many regional languages feature unique slang, local expressions, and values; stakeholders therefore seek LLMs aligned with local norms and vernacular. In personalized recommendation (Li et al., 2023), users expect models to respect user-specific facts (e.g., names, titles, birthdays, preferences in music and film) that are not common knowledge. In preference-aware household robotics (Han et al., 2024) and clinical guidance (Kadariya et al., 2019), preferences are proprietary, diverse, and highly individualized (e.g., how dishes are organized, when and where laundry is handled, where cups are stored). Reinforcement Learning from Human Feedback (RLHF) optimizes a single global reward that collapses heterogeneous user preferences, which under-represents individual tastes and limiting generalization to unseen, user-specific preferences (Chakraborty et al., 2024; Ouyang et al., 2022a). Supervised fine-tuning (SFT) can encode specific behaviors when sufficient labeled data are available, but assembling large, high-quality annotations is costly (Ouyang et al., 2022b; Wei et al., 2021; Taori et al., 2023; Tan et al., 2024a; Salemi et al., 2024; Zhang et al., 2023a; Tu et al., 2025). To address personalization with RLHF, Poddar et al. (2024) propose training a latent-conditioned re-

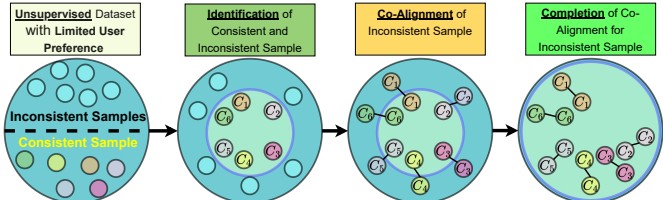

Figure 1: **Schematic overview of *Heterogeneous-Consistency Co-Alignment* (HCC).** The inner circle denotes the consistent set $\mathcal{C}$ with higher reliability, while the outer region corresponds to inconsistent samples $\mathcal{I}$ with lower reliability. Each intent cluster is represented by $C_i$. HCC identifies consistency via inter-model comparison and performs co-alignment using a non-parametric, embedding-based preference assignment scheme to refine inconsistent samples under limited preference supervision.

ward model on pluralistic preference corpora and then adapting to new users with a few additional queries. Nonetheless, it still requires training on large, diverse preference corpora. In addition, when a new user's preferences diverge substantially from that training distribution, reward-model generalization may degrade. Yet in practice, pretrained LLMs may hallucinate seemingly plausible but incorrect answers due to reliance on population-level statistics (Kalai et al., 2025). Lastly, conventional prompt-based approaches Wei et al. (2021; 2022); Huang et al. (2022); Yao et al. (2022); Diao et al. (2024); Liu et al. (2023); Wang et al. (2023); Yao et al. (2023); Long (2023); Huang et al. (2023); Madaan et al. (2024); Shinn et al. (2024); Chen & Tsang (2024) excel at producing factual responses to widely known questions, but they remain unreliable for queries involving personal or user-specific information and preferences.

Therefore, it is crucial to build an efficient and economical pipeline for generating preference-aligned annotations from a small, user-labeled preference set and propagating them across large unlabeled corpora which are typically more scalable and cheaper to obtain (van Engelen & Hoos, 2020; Zhu, 2020; Chen et al., 2025), so that LLMs move beyond population-level statistics toward user-specific expectations. In addition, conventional reference-based metrics (see Table 8 in the Appendix) fall short in reference-free settings, while naive self-evaluation methods within LLMs often exhibit overconfidence and inconsistencies (Xiong et al., 2023; Stureborg et al., 2024; Zhou et al., 2024), as well as vulnerabilities to prompt format. Moreover, relying solely on the LLM introduces reproducibility issues due to frequent updates of proprietary closed-source models Ito et al. (2025). Collectively, these limitations highlight a critical need for robust, reference-free evaluation and co-alignment frameworks capable of effectively aligning with nuanced, personalized user preferences in semi-supervised settings. In sum, to train user-preferred, competent LLMs for diverse individual users, especially when only a small number of user-preference exemplars are available, requires two key components: **(i)** an evaluation framework centered on an interpretable, personalization-aware, reference-free metric that explicitly accounts for the plurality of human preferences and ensures that generated annotations are accurate and useful. Because collecting large, personalized corpora is prohibitively expensive, training evaluators or reward models (e.g., regression models) from scratch is often infeasible (Ito et al., 2025), which underscores the importance of such a reference-free metric in the limited-sample regime. **(ii)** a model-agnostic, low-overhead annotation-alignment paradigm that effectively rectifies misaligned annotation.

- **(i) Reference-Free Uncertainty Evaluation in LLMs:** *How can we assess whether LLM-generated outputs are consistent with user preferences in a general reference-free setting, without access to ground-truth labels or external references?*

- **(ii) Semi-Supervised Co-Alignment for Categorical NLU:** *How can we design a co-alignment mechanism that recalibrates LLM-generated annotations for categorical NLU tasks based on a small set of user preference examples?*

To address these challenges, we propose the *Heterogeneous-Consistency Co-Alignment* (HCC), a training-free framework consisting of (i) reference-free evaluation and (ii) a co-alignment mechanism for LLM-generated annotations. **For evaluation**, we introduce the *Consistent-and-Inconsistent (CAI) Ratio*, a reference-free metric for assessing annotation alignment in preference-based labeling of unlabeled textual datasets. CAI measures the odds of agreement between two *heterogeneous annotators*: (1) an LLM that produces responses driven by next-token probabilities, and (2) a task-specific/lightweight model that operates in embedding space. By cross-checking their out-

puts, CAI identifies where token-likelihood-derived decisions and embedding-similarity judgments *agree* (consistent samples) versus *disagree* (inconsistent samples), providing a reliability signal that mitigates the overconfidence observed when relying on the LLM alone. **For co-alignment**, HCC applies *MV-VTES* (see §4.3.2): a nonparametric, embedding-based assignment that takes majority votes among the top-$k$ nearest neighbors within clusters seeded by user-preferred exemplars. This propagates preferences to initially unlabeled data. Next, HCC compares the specialized model's assigned annotations with the LLM's outputs and performs *divide-and-conquer co-alignment (DCCA)*, again using MV-VTES, to repair disagreements. Crucially, HCC avoids dependence on ad-hoc confidence thresholds derived from next-token probabilities, thereby reducing overconfidence errors. Our main contributions are as follows:

- We propose *Heterogeneous-Consistency Co-Alignment* (**HCC**), a novel training-free framework that (i) enables reference-free uncertainty evaluation in LLMs and (ii) provides an efficient co-alignment mechanism to recalibrate LLM-produced annotations in categorical NLU tasks using only a small set of user preference examples.

- We introduce the reference-free *Consistent-And-Inconsistent* (CAI) Ratio, a metric that quantifies inter-model agreement as a proxy for preference-alignment uncertainty, enabling evaluation without ground-truth labels and addressing the challenge of assessing LLM-generated annotations under limited user-labelled examples.

- We validate the effectiveness of HCC on eight domain-specific categorical NLU datasets, demonstrating substantial and consistent improvements in annotation alignment.

## 2 RELATED WORKS

**LLMs for Data Annotation.** Large Language Models (LLMs) have demonstrated strong capabilities in text and data annotation across a wide range of open-domain tasks (Meng et al., 2022; Ye et al., 2022; Wang et al., 2024; Liu et al., 2024; Wu et al., 2024), including spoken language understanding (Chen et al., 2023; 2024). In many cases, LLM-generated labels rival or exceed crowdsourced and manual annotations without task-specific training (Gilardi et al., 2023). Consequently, such annotations have been widely adopted for supervised fine-tuning, alignment optimisation, and downstream inference (Tan et al., 2024b). However, relying solely on LLMs for annotation can be risky. Without explicit verification mechanisms, LLM outputs may exhibit overconfidence, systematic bias, or misalignment with user-defined preferences (Xiong et al., 2023; Stureborg et al., 2024; Zhou et al., 2024). This concern becomes particularly salient in preference-aligned NLU settings, where annotations must reflect personalised or task-specific intent. In practice, many real-world scenarios operate in a semi-supervised regime, where a small set of labelled preference examples guides annotation over large unlabelled corpora. While this setting reduces the need for extensive supervision, reliable evaluation remains challenging when labelled data are scarce. Moreover, when the unlabelled corpus is large, repeatedly querying closed-source LLMs may be prohibitively expensive, whereas smaller open-source models may lack sufficient reliability for direct deployment. These challenges highlight the need for training-free frameworks that combine strong or weak LLMs with accessible task-specialised models, while incorporating reference-free evaluation and alignment mechanisms.

**Verification Gap in Model-Guided LLM Alignment.** While pretrained task-specific models can efficiently annotate unlabelled data using few examples, their performance often degrades when user preferences diverge from the original training distribution. Consequently, using these specialised models to guide LLMs is risky: it implicitly assumes the model's predictions are preference-consistent. Without verification, these errors can propagate to the LLM, amplifying misalignment. A similar flaw exists in conventional clustering methods, which assume geometric proximity in embedding space ensures semantic correctness. In practice, personalised preferences rarely align perfectly with embedding homogeneity. Even LLM-guided clustering methods like ClusterLLM (Zhang et al., 2023b) remain bound by an initial embedding hierarchy, ultimately lacking a principled, reference-free mechanism to verify and correct preference misalignment under limited supervision. Conversely, HCC treats embedding similarity as a structural prior for propagation rather than as a proxy for correctness, and it retains or revises annotations based on the agreement between the LLM and embedding-induced hypotheses, thereby avoiding blind reliance on geometric proximity.

**Prompt-based Approaches.** Prompt-based methods leverage in-context learning to guide LLM outputs without parameter updates. Self-Consistency (Wang et al., 2022) enhances reliability by

sampling multiple reasoning paths and selecting the most consistent output. While effective in improving response accuracy, this approach assumes the underlying LLM is sufficiently capable and typically requires repeated sampling, resulting in substantial computational overhead. Moreover, consistency across reasoning paths does not guarantee alignment with user-defined preferences. Chain-of-Thought (CoT) prompting (Wei et al., 2022) improves interpretability and performance by explicitly modelling intermediate reasoning steps, while Few-shot Prompting (Brown et al., 2020) incorporates a small number of illustrative examples to steer outputs toward task-specific intent. Despite their empirical success, these methods primarily rely on the intrinsic capabilities of the LLM and the quality of prompt construction. They lack explicit mechanisms for reference-free reliability assessment or systematic co-alignment with personalised preference distributions under limited supervision. KATE (Liu et al., 2022) improves prompting by retrieving the top-$k$ semantically similar labelled examples as demonstrations. However, when the seed labelled set is small, the retrieved examples may fail to capture the diversity and ambiguity of user preferences, limiting its adaptability to personalised annotation tasks. Finally, Self-Refine (Madaan et al., 2024) iteratively improves initial outputs through self-correction. While promising, its success depends heavily on both the capability of the LLM and the suitability of any external tools used for the target task.

## 3 PROBLEM SETTING

Let the unlabeled training corpus and the test corpus be denoted as $\mathcal{D}_u = \{x_1, \ldots, x_N\}$ and $\mathcal{D}_{\text{test}} = \{x'_1, \ldots, x'_L\}$, where each instance satisfies $x \in \mathcal{X} \subseteq \mathbb{R}^d$. A small set of user-preference annotations (see Section 4.1) serves as the alignment reference set, with the goal of propagating a preference label $\bar{y} \in \mathcal{Y} = \{1, \ldots, k\}$ to each $x \in \mathcal{D}_u$. We assume that the distribution $\mathcal{D}_u$ can be partitioned into two subsets: consistent samples $\mathcal{C}$ and inconsistent samples $\mathcal{I}$, such that $\mathcal{C}, \mathcal{I} \subseteq \mathcal{D}_u, \mathcal{C} \cap \mathcal{I} = \emptyset$, and $|\mathcal{C}| + |\mathcal{I}| = |\mathcal{D}_u|$. In practice, the subsets are unknown and must be estimated (see Section §4) using a specialized embedding-based model $\mathcal{S}$ and an LLM $\mathcal{T}$, given a small user-preference set $H$. Formally, $H$ is clustered into $k$ disjoint subsets $\{C_1, \ldots, C_k\}$, each representing a semantically coherent region of the embedding space. Alongside the samples, a set of preference annotations is also given, denoted as $\mathcal{Y} = \{\bar{y}_1, \bar{y}_2, ..., \bar{y}_k\}$. Hence each cluster is formally denoted as $C_j = \{(x_i, \bar{y}_j) | x_i \in H_j\}$, where $H_j \subseteq H$ and $H = \{(x_i, \bar{y}_i)\}_{i=1}^s$, with $s = 5\%$ of $|\mathcal{D}_u|$. The clusters are disjoint, satisfying $(C_i \cap C_j = \emptyset, \forall i \neq j)$, and their union fully covers $H$. Our goals are: (i) to assess LLM annotation alignment by identifying latent consistent and inconsistent subsets $\mathcal{C}^*$ and $\mathcal{I}^*$, and (ii) to improve annotation accuracy and alignment according to user preferences.

## 4 CO-ALIGNING LLMS WITH *Heterogeneous-Consistency Co-Alignment*

HCC leverages agreements (consistent samples) and disagreements (inconsistent samples) between two heterogeneous models: an LLM annotator, which generates responses from token-level probabilities, and a task-specific model, which captures fine-grained embedding similarities. Cross-checking both models identifies samples where token probabilities and embeddings align, mitigating the overconfidence of LLM-only predictions. For co-alignment, HCC employs MV-VTES, a non-parametric, embedding-based preference assignment that uses majority voting over the top-$k$ nearest neighbors within clusters initialized by user-preferred samples, enabling annotation of unlabeled data. Finally, HCC refines inconsistent samples via a divide-and-conquer co-alignment (DCCA) strategy, again combined with MV-VTES.

### 4.1 SEMANTIC CLUSTERING-BASED ANNOTATION FOR UNLABELLED DATA

We propose a semantic clustering approach to propagate and refine annotations over unlabelled data. By applying majority voting over top-$k$ nearest neighbours in embedding space (see Section 4.3.2), labels from a small user-preference reference set are extended to semantically similar instances. This structured propagation provides local contextual signals for detecting and correcting misaligned annotations, thereby supporting co-alignment in semi-supervised settings.

### 4.1.1 TASK-SPECIFIC SPECIALISED MODEL

We adopt `MiniLM` (Wang et al., 2020), a sentence-transformer model, as the task-specialised model, denoted by $\mathcal{S}$. The model encodes each instance $x_i$ into a sentence embedding: $\mathcal{S}(x_i) = e_i$. We em-

ploy a semantic clustering-based assignment scheme, where annotations are assigned according to similarity in the embedding space without additional model training. Each cluster $C_j$ consists of embeddings of samples associated with label $j$. To measure alignment between an instance and a cluster $C_j$, we compute the average cosine similarity over the top-$k$ most similar embeddings in that cluster:

$$AS(e_i, C_j) = \frac{1}{k} \sum_{e \in \text{Top-}k(C_j, e_i)} \frac{e_i \cdot e}{\|e_i\|\|e\|}. \quad (1)$$

Here, $e_i$ denotes the embedding of $x_i$, and $e$ represents embeddings of samples in cluster $C_j$. The operator Top-$k(C_j, e_i)$ selects the $k$ samples in $C_j$ with the highest cosine similarity to $e_i$. In our experiments, we set $k$ to five. The cluster assigned to $x_i$ is determined by

$$C_{j^*} = \arg\max_{C_j} AS(e_i, C_j). \quad (2)$$

The corresponding cluster label $\bar{y}_{j^*}$ is then assigned to $x_i$, i.e., $\bar{y}_i = \bar{y}_{j^*}$. This defines the annotation function $\mathcal{S}(x_i)$ induced by the specialised model. The resulting annotated dataset is $D_S = \{(x_i, \bar{y}_i^{(S)})\}_{i=1}^N$, where $\bar{y}_i^{(S)}$ denotes the annotation assigned through this semantic clustering-based assignment scheme.

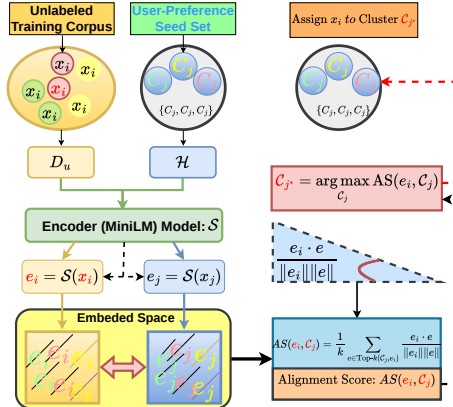

Figure 2: Semantic Clustering-Based Annotation Assignment Scheme Based on Embedding Similarity in Task-Specialised Model.

### 4.1.2 GENERAL LLM ANNOTATION

Given the specialised-model annotated dataset $D_S$, we query a large language model $\mathcal{T}$ to generate annotations via group prompting. We consider both zero-shot and single-shot settings.

**Zero-shot.** The LLM generates annotations independently for each input: $\bar{y}_i^{(\mathcal{T})} = \mathcal{T}(x_i)$, with autoregressive likelihood $P(\bar{y}_i^{(T)}|x_i) = \prod_{j=1}^{L_i} P\left(\bar{y}_{i,j}^{(T)}|x_i, \bar{y}_{i,<j}^{(T)}\right)$, where $L_i$ denotes the length of the generated sequence and $\bar{y}_{i,<j}^{(T)}$ represents previously generated tokens. This produces the LLM-annotated dataset $D_T = \{(x_i, \bar{y}_i^{(T)})\}_{i=1}^N$.

**Single-shot.** We incorporate the specialised-model annotation as contextual guidance: $\hat{y}_i^{(T)} = \mathcal{T}(x_i, \bar{y}_i^{(S)})$, with likelihood $P(\hat{y}_i^{(T)}|x_i, \bar{y}_i^{(S)}) = \prod_{j=1}^{L_i} P\left(\hat{y}_{i,j}^{(T)}|x_i, \bar{y}_i^{(S)}, \hat{y}_{i,<j}^{(T)}\right)$. This yields the augmented annotated dataset $\hat{D}_T = \{(x_i, \hat{y}_i^{(T)})\}_{i=1}^N$.

### 4.2 IDENTIFICATION OF CONSISTENT AND INCONSISTENT SAMPLES

After obtaining the specialised-model dataset $D_S$, the LLM-generated dataset $D_T$, and the augmented dataset $\hat{D}_T$, we introduce **Consistent-and-Inconsistent (CAI) Identification**, an agreement-based procedure that partitions samples into consistent and inconsistent subsets via cross-model annotation comparison. Based on this partition, we define the **CAI Ratio** as a reference-free reliability metric that quantifies annotation stability by measuring the relative proportion of consistent samples within the unlabeled corpus. In contrast to conventional clustering similarity measures, such as the split and merge framework of Xiang et al. (2012), which decomposes partition similarity via meet-weighted local aggregation, CAI applies an instance-level aggregation principle to quantify inter-annotator consistency. For each $x \in \mathcal{D}_u$, the annotation label from the task-specialised model is denoted as $\bar{y}^{(S)}$, and that for the LLM is denoted as $\bar{y}^{(T)}$ (zero-shot) and $\hat{y}^{(T)}$ (single-shot). A sample is considered *consistent* if:

$$\bar{y}^{(S)} = \bar{y}^{(T)} = \hat{y}^{(T)} \quad \Rightarrow \quad x \in \mathcal{C},$$

where $\mathcal{C}$ represents the set of consistent samples, while conversely, a sample is considered *inconsistent* if at least one of the annotations differs:

$$\exists (y, y') \in \{\bar{y}^{(S)}, \bar{y}^{(T)}, \hat{y}^{(T)}\}, \quad y \neq y' \quad \Rightarrow \quad x \in \mathcal{I},$$

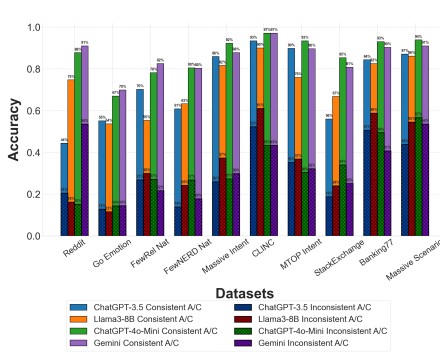

Figure 3: **An illustrative figure highlighting the importance of *consistent-and-inconsistent* sample identification in evaluating LLM performance.** LLM annotations on inconsistent samples (**dark-colored bars**) exhibit significantly lower accuracy compared to those on consistent samples (**light-colored bars**).

Figure 4: **Visualization of t-SNE Clustering** (better viewed in color, enlarged) comparing LLM vs Ground-Truth Annotations on *Go_Emotion* Dataset. LLM outputs exhibit **high similarity** with ground-truth labels on **consistent** samples, while showing ***significant divergence*** on **inconsistent** samples.

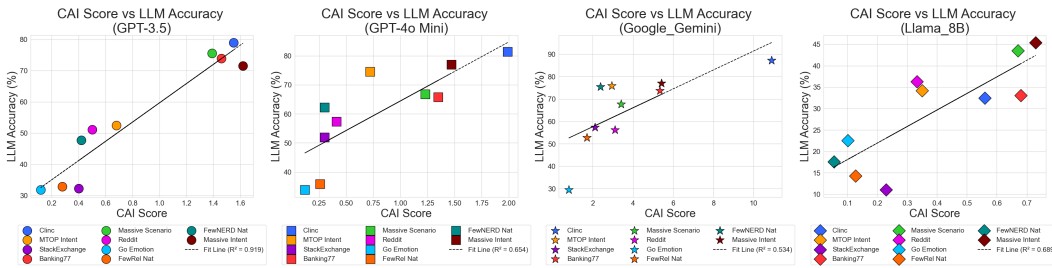

Table 1: **Correlation analysis between LLM annotation accuracy and the CAI ratio**, evaluated across four principled LLMs (also see statistical test results in Sec 6.1). The Pearson correlation coefficients and corresponding p-values confirm the statistical significance of the positive correlation between the CAI ratio and LLMs accuracy.

where $\mathcal{I}$ represents the set of inconsistent samples. Identifying annotation inconsistencies and evaluating the reliability of LLM outputs without ground truth remains a central challenge in semi-supervised LLM annotation. Our HCC framework addresses this by leveraging disagreement between the task-specialised model and the LLM to identify unreliable annotations and trigger co-alignment. We introduce the *Consistent-and-Inconsistent (CAI) Ratio*, a novel HCC metric for assessing the trustworthiness of LLM-generated labels without access to ground truth. For the formal definition of CAI, we fix the LLM hypothesis (either zero-shot or single-shot), ensuring consistency with Definition 4.1 and Proposition 4.2.

**Definition 4.1 (Consistent-and-Inconsistent (CAI) Ratio)** *Let $D_u = \{x_i\}_{i=1}^n$ be an unlabelled dataset. Given an LLM hypothesis $T$ and a task-specialised model hypothesis $S$, define $\mathcal{C} := \{x_i \in D_u : \mathcal{T}(x_i) = \mathcal{S}(x_i)\}$, $\mathcal{I} := \{x_i \in D_u : \mathcal{T}(x_i) \neq \mathcal{S}(x_i)\}$. Let $N_C := |\mathcal{C}|$ and $N_{IC} := |\mathcal{I}|$. The CAI Ratio is defined as $\mathrm{CAI}(D_u; \mathcal{T}, \mathcal{S}) := \frac{N_C}{N_{IC}+\varepsilon}$, where $\varepsilon > 0$ is a small constant to avoid division by zero.*

The CAI Ratio leverages heterogeneous consistency, combining likelihood-derived signals from the LLM with embedding-based agreement from a task-specialised model, to evaluate the reliability of LLM-generated annotations without labelled supervision. A markedly high CAI Ratio ($\mathrm{CAI} \gg 1$) may indicate systematic agreement that could stem from overconfidence or shared bias, whereas a low CAI Ratio ($\mathrm{CAI} \ll 1$) reflects frequent disagreement and thus elevated uncertainty in the model's outputs. In both regimes, the ratio serves as a reference-free indicator of annotation align-

ment, guiding decisions on whether LLM-generated labels should be refined through additional human supervision or enriched prior knowledge. Furthermore, we formalise the relationship between the CAI Ratio and annotation accuracy through the *Consistency Principle*. Under mild assumptions that the LLM and the task-specialised model share a common latent preference-aligned target and exhibit bounded disagreement, the proportion of consistent samples is expected to exceed that of inconsistent samples as the dataset size $|D_u|$ grows. This asymptotic behaviour provides theoretical support for interpreting CAI as a proxy for annotation reliability.

**Proposition 4.2 (Consistency Principle)** *Let $D_u = \{x_i\}_{i=1}^n$ be i.i.d. samples drawn from $\mathcal{D}$. Let $\mathcal{T}$ and $S$ denote the LLM-induced and embedding-induced hypotheses, respectively, and assume both are optimal under the same target distribution. Define $p_{\text{agree}} := \Pr_{x \sim \mathcal{D}}[\mathcal{T}(x) = \mathcal{S}(x)]$ and $p_{\text{dis}} := \Pr_{x \sim \mathcal{D}}[\mathcal{T}(x) \neq \mathcal{S}(x)]$. Let $N_C = \sum_{i=1}^n \mathbb{I}[\mathcal{T}(x_i) = \mathcal{S}(x_i)]$ and $N_{IC} = \sum_{i=1}^n \mathbb{I}[\mathcal{T}(x_i) \neq \mathcal{S}(x_i)]$. If $p_{\text{agree}} > p_{\text{dis}}$, then $\lim_{|D_u| \to \infty} \Pr(N_C > N_{IC}) = 1$.*

## 4.3 HETEROGENEOUS-CONSISTENCY CO-ALIGNMENT

Given the consistent samples with high-quality annotations $\mathcal{C} = \{(x_i, \bar{y}_i^c)\}_{i=1}^{|\mathcal{C}|}$ and the inconsistent samples $\mathcal{I} = \{(x_i, \bar{y}_i^I)\}_{i=1}^{|\mathcal{I}|}$ identified via CAI, together with the user-preference set $H$, we perform co-alignment of the annotations in $\mathcal{I}$ using a divide-and-conquer strategy. Here, $\bar{y}_i^c$ and $\bar{y}_i^I$ denote the annotations associated with consistent and inconsistent samples, respectively. As illustrated in Fig. 3, the samples in $\mathcal{C}$ exhibit substantially higher accuracy than those in $\mathcal{I}$. We therefore leverage the reliable annotations in $\mathcal{C}$ to guide the correction and realignment of the inconsistent samples in $\mathcal{I}$.

### 4.3.1 DIVIDE-AND-CONQUER CO-ALIGNMENT (DCCA)

**Round 1.** (*co-aligning $\mathcal{I}$ to obtain $\mathcal{I}^{(1)}$*). For each $(x, \bar{y}^I) \in \mathcal{I}$, we apply the **(MV-VTES)** (Section 4.3.2) using the reference set $\mathcal{C} \cup H$. Let $\hat{y}$ denote the reassigned annotation produced by MV-VTES. We define $\mathcal{I}^{(1)} = \{(x, \hat{y}) \mid (x, \bar{y}^I) \in \mathcal{I}\}$. Thus, $\mathcal{I}^{(1)}$ represents the originally inconsistent samples after first-round reassignment. We then perform CAI identification by comparing the original annotation $\bar{y}^I$ in $\mathcal{I}$ with the updated annotation $\hat{y}$ in $\mathcal{I}^{(1)}$, yielding the partition $\mathcal{I} = \mathcal{CI} \cup \mathcal{II}, \mathcal{CI} \cap \mathcal{II} = \varnothing$. Here, $\mathcal{CI}$ contains the samples that become consistent after the first alignment pass, while $\mathcal{II}$ contains those that remain inconsistent.

**Round 2.** (*Co-aligning $\mathcal{II}$ to obtain $\mathcal{II}^{(1)}$*). For each remaining $(x, \bar{y}^I) \in \mathcal{II}$, we again apply MV-VTES, this time using the expanded reference set $\mathcal{C} \cup \mathcal{CI} \cup H$. Let $\tilde{y}$ denote the newly assigned annotation, and define $\mathcal{II}^{(1)} = \{(x, \tilde{y}) \mid (x, \bar{y}^I) \in \mathcal{II}\}$. After the second pass, the fully aligned version of the originally inconsistent set is $\mathcal{I}^{(2)} = \mathcal{CI} \cup \mathcal{II}^{(1)}$. Finally, the self-corrected dataset is $D^{(\text{final})} = \mathcal{C} \cup \mathcal{CI} \cup \mathcal{II}^{(1)}$. Overall, DCCA partitions $\mathcal{I}$ into $\mathcal{CI}$ and $\mathcal{II}$, resolves the easier inconsistencies in the first pass and the more challenging ones in the second, and produces the refined dataset $D^{(\text{final})}$ through iterative co-alignment.

### 4.3.2 MAJORITY VOTING VIA TOP-NEAREST EMBEDDING SCHEME (MV-VTES)

Our divide-and-conquer co-alignment (DCCA) strategy relies on the MV-VTES method, which refines annotations of inconsistent samples by selecting labels from the most semantically similar references in $C \cup H$. Given a query sample $x \in D_u$, we first determine its consistency status using CAI. For a sample identified as inconsistent ($x \in I$), we refine its annotation via MV-VTES using the labeled reference set $\mathcal{D}_e = C \cup H \subset \mathcal{X} \times \mathcal{Y}$, where each element is a pair $(a, \bar{y})$ consisting of an input $a$ and its high-quality annotation $\bar{y}$. Let $\mathcal{A}_e = \{a \in \mathcal{X} \mid (a, \bar{y}) \in \mathcal{D}_e\}$ denote the set of reference inputs. We retrieve the top-$k$ most similar reference inputs to $x$ according to cosine similarity in embedding space:

$$\{a_i\}_{i=1}^k = \operatorname*{TopK}_{a \in \mathcal{A}_e} \left( \frac{\mathcal{S}(a) \cdot \mathcal{S}(x)}{\|\mathcal{S}(a)\| \|\mathcal{S}(x)\|} \right), \tag{3}$$

where $\mathcal{S}(\cdot)$ denotes the embedding function. For each retrieved input $a_i$, let $\bar{y}_i$ denote its associated annotation in $\mathcal{D}_e$. The refined annotation $\hat{y}$ for $x$ is then obtained via majority voting over the retrieved labels. Specifically, let $A = \{\bar{y}_1, \ldots, \bar{y}_k\}$ be the set of unique labels among the retrieved

**(a) GPT-3.5 Turbo (Closed-source LLM).** HCC achieves the best accuracy on 5 datasets.

| Dataset | Spec. Model | LLM Zero-shot | Spec.+LLM | Clust. | CoT | FoT | Self Cons. | Self Ref. | HCC w/o Corr. | HCC w/ Corr. | CAI Before→After |
|---|---|---|---|---|---|---|---|---|---|---|---|
| Clinc | $79.01_{\pm1.08}$ | $66.58_{\pm3.36}$ | $76.82_{\pm1.51}$ | $78.58_{\pm0.41}$ | $45.46_{\pm0.55}$ | $46.66_{\pm0.46}$ | $76.56_{\pm0.03}$ | $64.39_{\pm0.14}$ | $81.32_{\pm0.46}$ | $\mathbf{85.49}_{\pm0.19}$ | 1.55→5.50 |
| Massive Scenario | $75.55_{\pm1.76}$ | $60.89_{\pm0.62}$ | $70.23_{\pm1.64}$ | $60.85_{\pm4.33}$ | $52.01_{\pm0.28}$ | $56.06_{\pm0.36}$ | $63.44_{\pm0.12}$ | $47.70_{\pm0.26}$ | $69.25_{\pm0.03}$ | $\mathbf{76.43}_{\pm2.47}$ | 1.39→4.72 |
| MTOP Intent | $52.49_{\pm2.52}$ | $64.95_{\pm0.21}$ | $55.12_{\pm3.08}$ | $37.22_{\pm1.18}$ | $58.41_{\pm0.90}$ | $59.64_{\pm0.73}$ | $68.00_{\pm0.26}$ | $39.99_{\pm0.08}$ | $\mathbf{79.57}_{\pm0.42}$ | $69.06_{\pm1.10}$ | 0.68→1.78 |
| StackExchange | $32.27_{\pm0.65}$ | $30.10_{\pm0.10}$ | $30.92_{\pm2.21}$ | $\mathbf{47.75}_{\pm1.24}$ | $9.71_{\pm0.34}$ | $13.50_{\pm0.19}$ | $37.18_{\pm0.70}$ | $21.21_{\pm0.76}$ | $29.76_{\pm0.19}$ | $41.45_{\pm2.56}$ | 0.40→0.85 |
| Banking77 | $73.93_{\pm0.81}$ | $65.12_{\pm0.30}$ | $75.39_{\pm0.32}$ | $71.20_{\pm1.59}$ | $27.24_{\pm0.05}$ | $32.34_{\pm0.28}$ | $56.10_{\pm0.05}$ | $36.74_{\pm0.07}$ | $73.56_{\pm0.20}$ | $\mathbf{82.45}_{\pm0.48}$ | 1.36→4.03 |
| Reddit | $51.73_{\pm0.62}$ | $51.12_{\pm1.27}$ | $51.64_{\pm0.18}$ | $57.02_{\pm1.59}$ | $22.69_{\pm0.51}$ | $27.52_{\pm0.75}$ | $41.15_{\pm0.26}$ | $26.88_{\pm0.51}$ | $43.90_{\pm1.59}$ | $\mathbf{58.77}_{\pm0.29}$ | 0.50→1.40 |
| FewRel-Nat | $35.35_{\pm0.02}$ | $32.87_{\pm1.72}$ | $37.37_{\pm0.13}$ | $\mathbf{51.22}_{\pm1.43}$ | $18.36_{\pm0.14}$ | $17.34_{\pm0.41}$ | $27.52_{\pm0.03}$ | $15.68_{\pm0.52}$ | $49.24_{\pm0.63}$ | $44.88_{\pm0.05}$ | 0.28→0.89 |
| Massive Intent | $61.80_{\pm1.04}$ | $71.52_{\pm0.95}$ | $64.54_{\pm0.02}$ | $60.69_{\pm0.02}$ | $52.52_{\pm1.33}$ | $55.89_{\pm0.36}$ | $\mathbf{74.88}_{\pm0.36}$ | $55.12_{\pm0.26}$ | $73.41_{\pm1.84}$ | $71.72_{\pm0.40}$ | 1.62→2.81 |

**(b) GPT-4o Mini (Closed-source LLM).** HCC achieves the best accuracy on 6 datasets.

| Dataset | Spec. Model | LLM Zero-shot | Spec.+LLM | Clust. | CoT | FoT | Self Cons. | Self Ref. | HCC w/o Corr. | HCC w/ Corr. | CAI Before→After |
|---|---|---|---|---|---|---|---|---|---|---|---|
| Clinc | $79.01_{\pm1.08}$ | $81.44_{\pm0.44}$ | $78.58_{\pm1.35}$ | $78.58_{\pm0.41}$ | $74.93_{\pm0.40}$ | $77.17_{\pm0.19}$ | $84.22_{\pm0.88}$ | $80.06_{\pm0.65}$ | $85.23_{\pm0.98}$ | $\mathbf{87.93}_{\pm0.53}$ | 2.06→5.20 |
| Massive Scenario | $75.55_{\pm1.76}$ | $66.83_{\pm1.31}$ | $77.62_{\pm0.74}$ | $60.85_{\pm4.33}$ | $62.96_{\pm0.28}$ | $70.17_{\pm0.24}$ | $68.99_{\pm0.76}$ | $50.27_{\pm0.95}$ | $79.60_{\pm0.85}$ | $\mathbf{80.18}_{\pm0.45}$ | 1.39→4.65 |
| MTOP Intent | $52.49_{\pm2.52}$ | $75.03_{\pm1.35}$ | $57.01_{\pm0.37}$ | $37.22_{\pm1.18}$ | $74.48_{\pm0.29}$ | $78.65_{\pm0.26}$ | $73.32_{\pm1.10}$ | $39.98_{\pm0.22}$ | $\mathbf{80.16}_{\pm0.85}$ | $67.10_{\pm0.32}$ | 0.74→1.66 |
| StackExchange | $32.27_{\pm0.65}$ | $\mathbf{51.90}_{\pm0.75}$ | $45.49_{\pm0.94}$ | $47.75_{\pm1.24}$ | $39.42_{\pm0.20}$ | $40.99_{\pm0.17}$ | $48.06_{\pm0.12}$ | $25.98_{\pm0.10}$ | $35.63_{\pm0.51}$ | $45.22_{\pm0.15}$ | 0.31→0.66 |
| Banking77 | $73.93_{\pm0.81}$ | $65.12_{\pm0.30}$ | $75.39_{\pm0.32}$ | $71.20_{\pm1.59}$ | $54.41_{\pm0.46}$ | $56.33_{\pm0.51}$ | $66.82_{\pm0.28}$ | $40.23_{\pm1.06}$ | $73.56_{\pm0.20}$ | $\mathbf{82.45}_{\pm0.48}$ | 1.36→4.03 |
| Reddit | $51.73_{\pm0.62}$ | $57.40_{\pm1.96}$ | $53.25_{\pm0.35}$ | $57.02_{\pm1.59}$ | $38.34_{\pm0.44}$ | $41.01_{\pm0.66}$ | $44.60_{\pm1.23}$ | $24.66_{\pm0.09}$ | $44.47_{\pm0.69}$ | $\mathbf{60.94}_{\pm0.11}$ | 0.51→1.90 |
| FewRel-Nat | $35.35_{\pm0.02}$ | $35.87_{\pm0.03}$ | $37.11_{\pm0.49}$ | $\mathbf{51.22}_{\pm1.43}$ | $28.47_{\pm0.57}$ | $33.95_{\pm0.85}$ | $43.57_{\pm0.28}$ | $23.49_{\pm0.13}$ | $49.53_{\pm0.35}$ | $44.94_{\pm0.02}$ | 0.26→0.90 |
| Massive Intent | $61.80_{\pm1.04}$ | $76.93_{\pm1.05}$ | $66.02_{\pm0.12}$ | $60.69_{\pm0.02}$ | $60.91_{\pm0.14}$ | $65.44_{\pm0.57}$ | $74.43_{\pm0.10}$ | $53.32_{\pm0.12}$ | $\mathbf{78.93}_{\pm0.50}$ | $72.49_{\pm0.40}$ | 1.47→3.30 |

**(c) Meta-Llama 3-8B Instruct (Open-source LLM).** HCC achieves the best accuracy on 7 datasets.

| Dataset | Spec. Model | LLM Zero-shot | Spec.+LLM | CoT | FoT | Self Cons. | Self Ref. | HCC w/o Corr. | HCC w/ Corr. | CAI Before→After |
|---|---|---|---|---|---|---|---|---|---|---|
| Clinc | $79.01_{\pm1.08}$ | $32.49_{\pm6.73}$ | $69.40_{\pm7.28}$ | $31.07_{\pm0.21}$ | $38.08_{\pm0.99}$ | $52.53_{\pm0.27}$ | $48.02_{\pm1.07}$ | $63.41_{\pm3.19}$ | $\mathbf{82.43}_{\pm0.20}$ | 0.56→4.43 |
| Massive Scenario | $75.55_{\pm1.76}$ | $43.52_{\pm1.85}$ | $66.74_{\pm0.98}$ | $44.29_{\pm1.26}$ | $43.10_{\pm1.10}$ | $58.11_{\pm0.15}$ | $54.65_{\pm1.29}$ | $70.06_{\pm1.12}$ | $\mathbf{78.13}_{\pm0.74}$ | 0.67→4.88 |
| MTOP Intent | $52.49_{\pm2.52}$ | $34.17_{\pm6.70}$ | $48.23_{\pm0.25}$ | $53.66_{\pm0.03}$ | $61.19_{\pm0.07}$ | $68.18_{\pm0.20}$ | $39.93_{\pm0.26}$ | $\mathbf{66.39}_{\pm0.70}$ | $63.39_{\pm1.47}$ | 0.35→1.46 |
| StackExchange | $32.27_{\pm0.65}$ | $11.02_{\pm2.78}$ | $26.26_{\pm2.16}$ | $15.05_{\pm1.58}$ | $16.04_{\pm1.38}$ | $5.04_{\pm0.21}$ | $21.26_{\pm0.76}$ | $16.03_{\pm0.13}$ | $\mathbf{38.88}_{\pm0.27}$ | 0.23→0.53 |
| Banking77 | $73.93_{\pm0.81}$ | $33.06_{\pm1.92}$ | $69.66_{\pm1.74}$ | $27.24_{\pm0.05}$ | $32.53_{\pm0.49}$ | $56.07_{\pm0.05}$ | $36.69_{\pm0.07}$ | $64.29_{\pm1.24}$ | $\mathbf{77.71}_{\pm0.25}$ | 0.68→4.20 |
| Reddit | $51.73_{\pm0.62}$ | $36.31_{\pm0.97}$ | $46.00_{\pm2.51}$ | $16.65_{\pm0.29}$ | $26.29_{\pm1.45}$ | $40.34_{\pm0.92}$ | $40.30_{\pm0.29}$ | $40.29_{\pm0.55}$ | $\mathbf{58.81}_{\pm0.29}$ | 0.33→1.58 |
| FewRel-Nat | $35.35_{\pm0.02}$ | $14.25_{\pm0.36}$ | $30.07_{\pm4.45}$ | $15.13_{\pm0.14}$ | $18.66_{\pm0.26}$ | $19.84_{\pm0.17}$ | $19.41_{\pm0.24}$ | $31.80_{\pm0.34}$ | $\mathbf{42.92}_{\pm0.06}$ | 0.13→0.85 |
| Massive Intent | $61.80_{\pm1.04}$ | $45.41_{\pm0.06}$ | $56.03_{\pm0.08}$ | $35.02_{\pm0.76}$ | $43.05_{\pm0.40}$ | $\mathbf{74.63}_{\pm0.19}$ | $54.93_{\pm0.36}$ | $67.49_{\pm0.10}$ | $67.75_{\pm0.43}$ | 0.73→2.87 |

Table 2: **Comparative Performance Across LLMs.** Accuracy (%) on eight benchmark datasets for GPT-3.5 Turbo, GPT-4o Mini, and Meta-Llama 3-8B Instruct. We compare (i) zero-shot inference, (ii) prompting and clustering baselines, and (iii) HCC before and after correction. HCC achieves the highest or near-highest accuracy on most datasets (5–7 per model) and consistently increases the CAI score (Before→After), indicating stronger cross-model consistency. Results are reported as mean ± standard deviation over three runs.

neighbors. For each $a \in A$, define its frequency as $n_a = \sum_{i=1}^{k} \mathbb{I}[\bar{y}_i = a]$. The refined annotation is assigned as

$$\hat{y} = \arg\max_{a \in A} n_a. \tag{4}$$

Hence, our method treats consistent samples as structural anchors for correcting inconsistent ones. Rather than discarding noisy or conflicting annotations, HCC systematically co-aligns them to improve global annotation consistency.

## 5 EXPERIMENTS

We conduct extensive comparisons against the following baselines: (1) Specialised Model (Reimers & Gurevych, 2019); (2) LLM (Grattafiori et al., 2024); (3) Specialised Model + LLM; (4) Clustering (Zhang et al., 2023b); (5) CoT (Wei et al., 2022); (6) FoT (Brown et al., 2020); (7) Self-Consistency (Wang et al., 2022); and (8) Self-Refine (Madaan et al., 2024). **Details of the baselines, datasets, and evaluation metrics are provided in Appendices D and E.**

### 5.1 EVALUATION RESULTS

**GPT-3.5 Turbo:** We have two key findings based on the experimental results from Table 2 a. First, our method (HCC) consistently outperforms baseline methods, achieving notable gains on `Clinc` (**+4.17%**), `Massive Scenario` (**+0.88%**), and `Bank77` (**+2.99%**). For the `MTOP Intent` and `StackExchange`, HCC surpasses the non-collaborative baselines, i.e., Only Specialised Model and Only LLMs (GPT 3.5), by **+16%/+4.11%** and **+9.18%/+11.35%**, respectively. This underscores the effectiveness of HCC in improving annotation accuracy through the collaborative refinement of a specialized model and an LLM. In particular, knowledge distillation between the specialized model (BERT) and the LLM using consistent samples (denoted as HCC w/o Corr) achieved the highest annotation accuracy on the `MTOP Intent` dataset.

**GPT-4o Mini:** Based on the results from Table 2 b, there are two key findings. First, *Heterogeneous-Consistency Co-Alignment* (HCC) consistently outperforms all baselines on `Clinc` (**+2.7%**),

| Dataset | HCC (MiniLM) | HCC (BERT) | GPT-4o Mini | GPT-3.5 Turbo | MiniLM Outcome | BERT Outcome |
|---|---|---|---|---|---|---|
| CLINC | **82.43** | 82.40 | 81.44 | 66.58 | **Beats Both** | **Beats Both** |
| Massive Scenario | **78.13** | 74.61 | 66.83 | 60.89 | **Beats Both** | **Beats Both** |
| MTop Intent | 63.39 | 65.62 | **75.03** | 64.95 | Lower than Both | Beats GPT-3.5 Only |
| StackExchange | 38.88 | 39.41 | **51.90** | 30.10 | Beats GPT-3.5 Only | Beats GPT-3.5 Only |
| Banking77 | **77.71** | 71.62 | 65.12 | 65.12 | **Beats Both** | **Beats Both** |
| Reddit | **58.81** | 58.13 | 57.40 | 51.12 | **Beats Both** | **Beats Both** |
| FewRel-Nat | **42.92** | 42.05 | 35.87 | 32.87 | **Beats Both** | **Beats Both** |
| Massive Intent | 67.75 | 62.61 | **76.93** | 71.52 | Lower than Both | Lower than Both |

Table 3: **HCC under MiniLM and BERT backbones compared with closed-source LLMs.** The highlighted columns report HCC performance under two specialized-model backbones. Green cells indicate cases where HCC outperforms both closed-source LLMs, blue cells indicate improvements over GPT-3.5 Turbo only, and orange cells indicate cases where HCC remains below both closed-source LLMs.

`Massive Scenario` (**+0.58%**), `Bank77` (**+7.06%**) and `Reddit` (**+3.92%**). Additionally, HCC without alignment achieved the highest annotation accuracy on `MTOP Intent`. HCC (w/ Corr) and HCC (w/o Corr) can be used interchangeably to enhance annotation accuracy.

**Meta-Llama3-8B Instruct:** HCC with correction achieves the highest accuracy on 7 of the 8 benchmark datasets. Despite the relatively weak zero-shot performance of Llama3-8B, our method substantially improves both accuracy and CAI, often surpassing the base LLM and the standalone specialised model. These results indicate that HCC remains effective even when the underlying LLM is comparatively weak, showing strong robustness to model quality.

## 5.2 HCC under Weaker (MiniLM and BERT-Base-Uncased) and Stronger (Llama-3-8B Instruct) Model Regimes

Overall, HCC consistently outperforms prompt-based baselines such as Self-Refine. Prompting methods often underperform on categorical NLU tasks: unlike reasoning benchmarks where executable code can verify solutions, NLU lacks an external correctness oracle, and self-generated feedback may even degrade annotation quality (Huang et al., 2023). Moreover, prompt-based approaches depend heavily on strong proprietary LLMs, with performance degrading on more complex tasks or weaker open-source models (e.g., Llama-3-8B). To evaluate robustness across backbone capacities, we pair two task-specialised models of different strengths—`MiniLM` (stronger) and `BERT-Base-Uncased` (weaker), with `Llama-3-8B Instruct`. This setting tests whether HCC remains effective under varying structural representation quality.

**`BERT-Base-Uncased`.** As shown in Table 3, HCC significantly improves annotation quality even with this weaker backbone. In this regime, `Llama-3-8B Instruct` combined with HCC outperforms `GPT-4o Mini` on five datasets and surpasses `GPT-3.5 Turbo` on seven datasets. These results indicate that HCC remains effective even when the specialised encoder provides comparatively weaker structural signals.

**`MiniLM`.** When paired with the stronger MiniLM backbone, HCC further improves performance consistency. Under this configuration, HCC combined with `Llama-3-8B Instruct` surpasses `GPT-3.5 Turbo` on six datasets and outperforms `GPT-4o Mini` on five datasets. This suggests that stronger structural representations amplify the gains from heterogeneous consistency while preserving robustness across datasets. Overall, these findings demonstrate that HCC enables competitive performance even when paired with weaker open-source LLMs, reducing reliance on proprietary models.

### 5.2.1 Imbalanced User-Preference Samples

In real-world scenarios, the distribution of user-preferred samples is often imbalanced. To assess the robustness of our method, we conduct a comprehensive evaluation of HCC under various annotation budgets, **1%, 5%, and 10%**, in the presence of imbalanced label distributions. We perform ablation studies to examine the impact of user preference imbalance, as shown in Table 36 (Appendix), using Llama 3-8B Instruct under an imbalance ratio of 60%. Specifically, after assigning one labeled sample per class, 60% of the remaining samples are allocated to the majority classes. This setup reflects a moderately skewed yet realistic user preference distribution. HCC assumes at least one labeled instance per class to align annotations with user intent. Despite the imbalance, HCC consistently outperforms strong baselines such as Self-Consistency (+29.58% on `Clinc`, +33.25%

on `Stackexchange`, +22.56% on `Few_Rel_Nat`), Self-Refine (+34.09% on `Clinc`, +37.63% on `Banking77`), and Clustering (+10.03% on `Banking77`, +22.26% on `Reddit`), all under balanced 5% label supervision. However, HCC does experience some performance degradation due to the imbalanced setup, with accuracy drops observed on `Clinc` (-0.32), `Massive_Scenario` (-0.46), `Stackexchange` (-0.59), `Reddit` (-0.11), and `Few_Rel_Nat` (-0.52). Overall, the HCC framework demonstrates strong robustness under moderately imbalanced conditions.

## 6 CORRELATION RESULTS BETWEEN CAI RATIO AND LLM ACCURACY

We performed a Pearson correlation analysis to investigate the relationship between the CAI Ratio and LLM accuracy. The correlation analysis between the Consistent-and-Inconsistent (CAI) ratio and accuracy across three different LLMs demonstrates a strong relationship between these two metrics. GPT-3.5 shows the highest correlation ($\rho = 0.93$, $p = 8.22 \times 10^{-5}$), indicating a very strong positive relationship between CAI and accuracy, with high statistical significance. GPT-4o Mini shows a strong correlation ($\rho = 0.86$, $p = 1.61 \times 10^{-3}$), suggesting that CAI is a reliable indicator of LLM accuracy for this model. Llama-8B-Instruct ($\rho = 0.81$, $p = 1.44 \times 10^{-2}$) and Google Gemini ($\rho = 0.72$, $p = 1.80 \times 10^{-2}$) exhibits moderate-to-strong correlations with significant statistical confidence.

### 6.1 CAI RATIO EVALUATION AND ANALYSIS

As shown in Table 2, applying HCC consistently increases the CAI ratio across GPT-3.5 Turbo, GPT-4o Mini, and Meta-Llama3-8B Instruct. In many cases, substantial CAI growth coincides with notable accuracy improvements. For closed-source models (GPT-3.5 and GPT-4o Mini), large CAI increases are typically accompanied by strong performance gains. Under GPT-4o Mini, `Banking77` improves from CAI **1.36** to **4.03** with accuracy rising from **65.12%** to **82.45%**. Similarly, `Clinc` increases from **2.06** to **5.20** (**81.44%** → **87.93%**), and `Massive Scenario` from **1.39** to **4.65** (**66.83%** → **80.18%**). A similar trend is observed under GPT-3.5 Turbo, where `Clinc`, `Massive Scenario`, and `Banking77` exhibit both strong CAI growth and corresponding accuracy gains. For the open-source Llama3-8B model, the effect is particularly pronounced. Although zero-shot Llama performance is relatively weak, HCC substantially increases CAI (e.g., **0.56** → **4.43** on `Clinc`, **0.67** → **4.88** on `Massive Scenario`), with large corresponding accuracy improvements (**32.49%** → **82.43%** and **43.52%** → **78.13%**, respectively). This demonstrates that HCC remains effective even when the base LLM exhibits limited initial consistency. However, the relationship is not strictly monotonic. On `MTOP Intent` under GPT-4o Mini, CAI increases from **0.74** to **1.66**, yet accuracy decreases from **75.03%** to **67.10%**. Similarly, on `StackExchange`, CAI improves modestly (e.g., **0.31** → **0.66**), while HCC remains below the LLM-only baseline. These cases indicate that CAI growth alone does not guarantee accuracy improvement. Empirically, substantial CAI increases, particularly when CAI moves well above 1, are often accompanied by meaningful performance gains, whereas marginal increases tend to provide limited structural reinforcement. HCC is generally most effective when the initial CAI already reflects moderate cross-model agreement (e.g., CAI > 1 for closed-source models or > 0.5 for weaker LLMs), suggesting that co-alignment amplifies existing structural consistency rather than creating it from scratch. Across model families, HCC achieves the highest accuracy on the majority of datasets, showing consistent effectiveness across both closed- and open-source LLMs.

## 7 CONCLUSION

This paper addresses reference-free evaluation and co-alignment in semi-supervised NLU annotation under limited user preference supervision. We propose *Heterogeneous-Consistency Co-Alignment* (HCC), a training-free framework that leverages agreement and disagreement between an LLM and a task-specialised model to assess and refine annotation quality. HCC introduces the CAI Ratio as a reference-free indicator of alignment and employs a divide-and-conquer co-alignment strategy to recalibrate inconsistent samples. Extensive experiments across eight NLU benchmarks demonstrate that HCC consistently improves annotation alignment. Notably, HCC enables competitive performance for open-source LLMs when compared with proprietary models. Furthermore, the CAI Ratio correlates strongly with observed alignment gains, supporting its role as a scalable reference-free evaluation signal. Future work will investigate theoretical extensions of heterogeneous consistency under broader preference modelling settings.

ETHICS STATEMENT

This work studies preference-aligned annotation through heterogeneous verification and co-alignment, without additional model training. All experiments are conducted on publicly available datasets and do not involve human subjects, private data, or high-stakes deployment settings.

REPRODUCIBILITY STATEMENT

We provide comprehensive documentation to ensure full reproducibility of our results. The complete Heterogeneous-Consistency Co-Alignment (HCC) procedure is specified in Appendix R (Algorithm Table), detailing the CAI-based identification process, verification mechanism, and divide-and-conquer co-alignment refinement steps. All prompting templates used for LLM annotation are reported in Appendix S (Prompt Instructions), including task formulations and preference-seed integration strategies for both zero-shot and single-shot settings. Additional implementation details, including experimental configurations and hyperparameters, are provided in Appendix G. All datasets used in our experiments are publicly available (Appendix D), and all evaluation metrics, including the proposed CAI Ratio, are formally defined in the main text and Appendix F. No proprietary datasets, hidden annotations, or undisclosed training procedures are involved.

ACKNOWLEDGEMENTS

This work was supported by the National Research Foundation, Singapore under its National Large Language Models Funding Initiative (AISG Award No: AISG-NMLP-2024-004). Any opinions, findings and conclusions or recommendations expressed in this material are those of the author(s) and do not reflect the views of National Research Foundation, Singapore.

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

APPENDIX

Appendix Overview.

# A   NOTATION SUMMARY

Summary of main notation used throughout the paper.

| Symbol | Description |
|---|---|
| $X \subset \mathbb{R}^d$ | Input space of textual instances. |
| $x_i$ | The $i$-th input instance. |
| $D_u = \{x_i\}_{i=1}^N$ | Unlabelled (semi-supervised) corpus. |
| $D_{\text{test}} = \{x_i\}_{i=1}^L$ | Test corpus used for evaluation. |
| $\mathcal{Y} = \{1, \ldots, C\}$ | Discrete label space with $C$ classes. |
| $H = \{(x_i, \bar{y}_i)\}_{i=1}^s$ | Small user-preference seed set. |
| $S$ | Task-specialised embedding model (e.g., MiniLM, BERT). |
| $T$ | Large language model (LLM) annotator. |
| $e_i = S(x_i)$ | Sentence embedding of $x_i$. |
| $\bar{y}_i^{(S)}$ | Annotation assigned by specialised model $S$. |
| $\bar{y}_i^{(T)}$ | Zero-shot annotation assigned by LLM $T$. |
| $\hat{y}_i^{(T)}$ | Single-shot annotation assigned by LLM $T$. |
| $D_S = \{(x_i, \bar{y}_i^{(S)})\}$ | Dataset annotated by specialised model. |
| $D_T = \{(x_i, \bar{y}_i^{(T)})\}$ | Zero-shot LLM-annotated dataset. |
| $\hat{D}_T = \{(x_i, \hat{y}_i^{(T)})\}$ | Single-shot LLM-annotated dataset. |
| C | Set of consistent samples (inter-model agreement). |
| I | Set of inconsistent samples (inter-model disagreement). |
| $C = \{x_i \in D_u : T(x_i) = S(x_i)\}$ | Formal definition of consistent set. |
| $I = \{x_i \in D_u : T(x_i) \neq S(x_i)\}$ | Formal definition of inconsistent set. |
| $N_C = |C|$ | Number of consistent samples. |
| $N_I = |I|$ | Number of inconsistent samples. |
| $\text{CAI}(D_u; T, S)$ | Consistent-and-Inconsistent Ratio. |
| $\text{CAI} = \dfrac{N_C}{N_I + \varepsilon}$ | Reference-free reliability metric. |
| CI | Subset of $I$ corrected in Round 1. |
| II | Subset of $I$ remaining inconsistent after Round 1. |
| $I^{(1)}$ | Inconsistent set after Round 1 reassignment. |
| $\mathcal{II}^{(1)}$ | Hard inconsistent samples corrected in Round 2. |
| $I^{(2)} = CI \cup \mathcal{II}^{(1)}$ | Fully aligned inconsistent set. |
| $D^{(\text{final})}$ | Final corrected dataset: $C \cup CI \cup \mathcal{II}^{(1)}$. |
| $K$ | Number of nearest neighbours used in MV-VTES. |
| $D_e = C \cup H$ | Reference set for embedding-based correction. |
| $\text{TopK}(a \in A_e)$ | Operator selecting $K$ nearest neighbours in embedding space. |
| $A = \{\bar{y}_1, \ldots, \bar{y}_K\}$ | Label set from retrieved neighbours. |
| $n_a = \sum_{j=1}^K \mathbf{1}[\bar{y}_j = a]$ | Frequency of label $a$ among neighbours. |
| $\hat{y} = \arg\max_{a \in A} n_a$ | Majority-vote refined annotation. |

## B  EXPERIMENTAL SETUP

We evaluate the CAI Ratio on multiple LLMs—GPT-3.5-Turbo, GPT-4o-Mini, Google Gemini 1.5 Flash (Model Selection Task), and Llama-3-8B-Instruct—across ten textual datasets. These datasets span a diverse range of domains and task types, including intent classification, topic modeling, sentiment/emotion analysis, and semi-supervised intent discovery (Zhang et al., 2021; 2022). The annotation practices follow Zhang et al. (2023b).

**Intent Classification.**

- **Banking77** (Casanueva et al., 2020)
- **CLINC150**(Larson et al., 2019a)
- **MTOP** (Li et al., 2020)
- **Massive (Intent)** (Larson et al., 2019b; FitzGerald et al., 2022)

**Topic and Question Classification.**

- **StackExchange**(Geigle et al., 2021)
- **Reddit** (Geigle et al., 2021)

**Emotion and Sentiment.**

- **GoEmotions** (Model Selection Task)

**Relation and Entity Classification.**

- **FewRel-Nat** (Han et al., 2018)
- **FewNerd-Nat** (Model Selection Task)

## C  PEARSON CORRELATION TEST FOR CAI SCORES AND LLMS ACCURACY

### C.1  STATISTICAL INFERENCE

We have performed a Pearson correlation, the correlation coefficient $r$ is calculated as:

$$r = \frac{\sum_{i=1}^{n}(x_i - \bar{x})(y_i - \bar{y})}{\sqrt{\sum_{i=1}^{n}(x_i - \bar{x})^2}\sqrt{\sum_{i=1}^{n}(y_i - \bar{y})^2}}$$

where $x_i$ symbolises the CAI ratios. $y_i$ denotes the LLM annotation accuracies. $\bar{x}$ and $\bar{y}$ are the average mean of $x_i$ and $y_i$, accordingly. $n$ is the number of samples we have used for evaluation. To assess the statistical significance, we use a hypothesis test for the correlation coefficient, calculating a t-statistic (Schober et al., 2018):

$$t = r\sqrt{\frac{n-2}{1-r^2}}$$

The P-value is then calculated from the t-distribution with $n - 2$ degrees of freedom.

### C.2  BEFORE APPLYING THE METHOD

The Pearson correlation coefficient is 0.805, indicating a strong positive linear relationship between CAI and Accuracy. The p-value is 0.005, which is statistically significant (below the typical threshold of 0.05). This implies that the positive correlation between the CAI ratio and Accuracy before and after applying our method is not a random event, and higher CAI scores are associated with higher Accuracy. The p-value is 0.00093, which is statistically significant (below the typical threshold of 0.05) for the additional datasets. The p-value is 0.014399, which is statistically significant (below the typical threshold of 0.05) for Meta-Llama-3-8B-instruct on all datasets.

Table 4: Pearson correlation results for **CAI** and **accuracy**, **before** and **after** our proposed method on GPT-3.5-Turbo and GPT-4 Mini, **before** and **after** our proposed method.

| Metric | Pearson Correlation | P-value |
|--------|---------------------|---------|
| Before | 0.805 | 0.005 |
| After | 0.903 | 0.00035 |

Table 5: Pearson correlation results for **CAI** and **accuracy** for additional datasets on GPT-3.5-Turbo and GPT-4 Mini, **before** and **after** our proposed method.

| Metric | Pearson Correlation | P-value |
|--------|---------------------|---------|
| **Before** | 0.874 | 0.000937 |
| **After** | 0.852 | 0.00175 |

Table 6: Pearson correlation results for **CAI** and **accuracy** on Meta-Llama-3-8B-instruct, **before** and **after** our proposed method.

| Metric | Pearson Correlation | P-value |
|--------|---------------------|---------|
| **Before** | 0.812 | 0.0144 |
| **After** | 0.918 | 0.00129 |

## C.3    AFTER APPLYING THE METHOD

The Pearson correlation coefficient is 0.903, showing an even stronger positive correlation between CAI and Accuracy after applying the method. A larger CAI ratio and higher annotation produced by LLMs are extremely statistically significant, according to the p-value of 0.00035. This implies that the relationship between CAI and Accuracy is even more evident after using the approach, showing a more linear relationship where increases in CAI are more directly correlated with increases in Accuracy. In both stages (Before and After applying the method), the results display statistically significant correlations ($p < 0.05$), showing strong positive relationships between CAI scores and LLM accuracy. Tables 4, 5 and 6 show that all the P-values of the Pearson correlation are statistically significant.

## D    DATASET

| Task | Name | # data (Testing) | # data (Training) | # classes |
|------|------|------------------|-------------------|-----------|
| **Intent** | Bank77 | 3,080 | 10,003 | 77 |
| | CLINC (I) | 4,500 | 15,000 | 150 |
| | MTOP (I) | 4,386 | 15,638 | 102 |
| | Massive (I) | 2,974 | 11,510 | 59 |
| **Type** | FewRel | 4,480 | 40,320 | 64 |
| **Topic** | StackEx | 4,156 | 50,000 | 121 |
| | Reddit | 3,217 | 50,000 | 50 |
| **Domain** | Massive Scenario | 2,974 | 11,514 | 18 |

Table 7: Summary of Benchmark Datasets.

## D.1    BENCHMARK DATASETS

We evaluate our work on a series of open-source textual datasets spanning diverse domains, including *Bank77*, *CLINC* (Intent), *MTOP* (Intent), *Massive* (Intent), *StackExchange* and *Reddit* (Topic) (Geigle et al., 2021), and Few Rel Nat (Type). We also utilize the *Massive Intent* dataset, with annotation practice following (Zhang et al., 2023b). *Bank77* (Casanueva et al., 2020) is a banking dataset that focuses on fine-grained intent classification within a single domain. *CLINC* (I), *Massive* (I), and *MTOP* (I) are intent-based datasets (denoted as "I") (Larson et al., 2019b; FitzGerald et al., 2022; Li et al., 2020). Intent discovery (Zhang et al., 2021; 2022) explores unknown intents in semi-supervised utterance datasets. Each dataset is available in small-scale (testing) and large-scale (training) versions, and we use i.i.d. user preference samples from the training set. Since we are

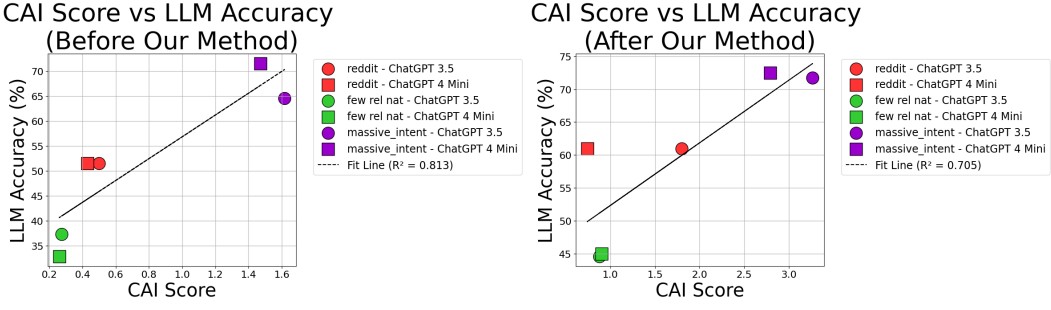

(a) GPT-3.5 Turbo and GPT-4o Mini      (b) GPT-3.5 Turbo and GPT-4o Mini

Figure 5: The above analysis shows the correlation between LLM annotation accuracy and the Consistent-and-Inconsistent (CAI) ratio. We also conducted statistical tests to assess the significance of this correlation. We collected the CAI ratios for (LLMs 3.5 Turbo and Specialised Model ) and (LLMs 4.0 Mini and Specialised Model ) across the datasets Reddit, few rel nat and massive intent. Using these data, we calculated the Pearson correlation coefficients between the LLM annotation accuracies and CAI ratios and computed the associated P-values (P value for After: 0.036) and (P value for Before: 0.014) to determine the statistical significance of the observed correlations.

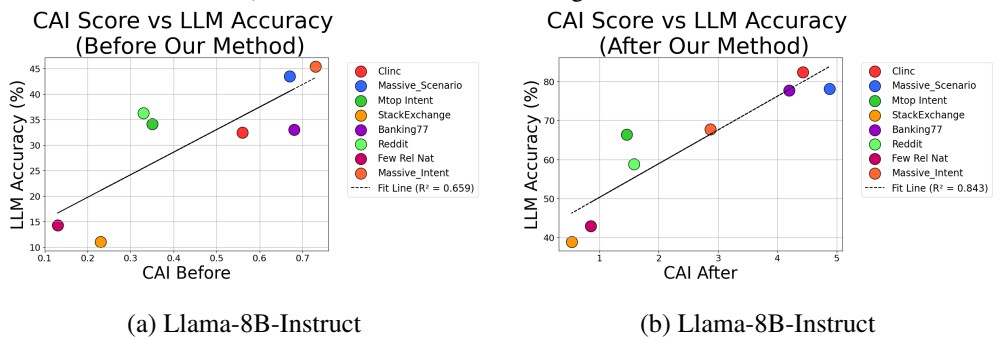

(a) Llama-8B-Instruct      (b) Llama-8B-Instruct

Figure 6: The above analysis shows the correlation between LLM annotation accuracy and the Consistent-and-Inconsistent (CAI) ratio. We also conducted statistical tests to assess the significance of this correlation. We collected the CAI ratios for (LLMs 3.5 Turbo and Specialised Model ) and (LLMs 4.0 Mini and Specialised Model ) across the datasets CLINC, Massive Scenario, MTOP Intent, Stack Exchange, and Banking77, Reddit, Few Rel Nat, Massive_Intent. Using these data, we calculated the Pearson correlation coefficients between the LLM annotation accuracies and CAI ratios and computed the associated P-values (P value for After: 0.014) and (P value for Before: 0.00129) to determine the statistical significance of the observed correlations.

working in a semi-supervised textual data setting, we directly use the small-scale set for testing. Detailed statistics of the datasets are shown in Table 7.

# E  BASELINES

To rigorously assess our proposed framework, we compare against a diverse set of strong baselines spanning specialized models, prompting methods, and clustering-based annotation strategies. Each baseline represents a distinct family of annotation approaches relevant to the semi-supervised setting.

**Specialised Model Only.**  A pre-trained task-specialised model assigns labels to unlabeled data using our preference-based annotation scheme and a small set of user-provided samples. This baseline measures the standalone utility of the specialised model without assistance from LLMs or consistency mechanisms. The specialised model relies on our proposed semantic clustering approach to organise examples into coherent intent-level groups and to assign each unlabeled instance a label that reflects the user-provided preference samples.

**LLMs Only (Annotation-Free Batch Prompting).**    We evaluate *GPT-3.5* and *GPT-4o mini* in a zero-shot setting, prompting them directly to generate category labels. While LLMs offer a scalable and cost-effective annotation mechanism, their outputs are prone to inconsistency, hallucination, or drift from user preferences in the absence of supervision or demonstrations. In the LLMs Only setting, the model is prompted using semantic batch clustering, where examples come from the same latent category but no explicit label is provided. This offers structure but not supervision.

**Specialised Model + LLMs (With Annotations Assigned by the Specialised Model as References).**    This variant employs our proposed *Majority Voting via Top-Nearest Embedding Similarity* method, using the task-specialised model to generate pseudo-labels for unlabeled samples. These pseudo-labeled instances are then used as reference annotations in the prompt to guide LLM annotation. This hybrid setup improves alignment while retaining annotation efficiency. In contrast, the Specialised Model + LLMs (Ours) setting augments the same clustered batch with an explicit intent label. This converts the prompt from a similarity-driven, label-free signal into a supervised alignment cue, enabling the LLM to ground its predictions in both semantic consistency and explicit task semantics.

**HCC w/o Co-Alignment.**    This ablation examines a simplified variant of our Heterogeneous Consistency Co-Alignment (HCC) framework. We exclude inconsistent samples and distill knowledge only from high-consistency examples into a pretrained BERT-based model. While this improves annotation precision, it lacks the full co-alignment loop present in HCC, making it less robust to preference drift.

**Clustering Baseline.**    We compare against Zhang et al. (2023b), a state-of-the-art clustering approach for few-shot annotation, where labels are propagated by embedding similarity. Unlike our method, it lacks preference modeling and relies heavily on embedding space separability. Our Top-Nearest Majority Voting introduces a new clustering method tailored to the semi-supervised regime.

**Prompt-Based Approaches.**    We evaluate several representative prompting methods, all using **KATE** (Liu et al., 2022) to select top-k similar demonstrations from user-labeled data:

- **Self-Consistency** (Wang et al., 2022) selects the most frequent response across multiple sampled reasoning paths, assuming the LLM is sufficiently competent. However, it incurs high compute cost and often reflects pretraining bias rather than user-specific intent.

- **Chain-of-Thought (CoT) Prompting** (Wei et al., 2022) guides the model to reason step-by-step via structured exemplars, improving interpretability and answer accuracy.

- **Few-Shot In-Context Learning** (Brown et al., 2020) provides a handful of labeled examples in the prompt to elicit more accurate model behavior, yet its effectiveness is sensitive to demonstration quality.

- **Self-Refine** (Madaan et al., 2024) iteratively improves model outputs through refinement steps and feedback, though its performance relies on LLM strength and auxiliary tool support.

## F    EVALUATION METRICS

We evaluate our method and the baselines based on two metrics: (1) *Annotation Accuracy*; (2) our proposed *CAI ratio*. Annotation Accuracy evaluates the correctness of LLM generated annotations, while the CAI Ratio measures the model's ability to correct inconsistencies. A higher CAI Ratio after applying our method indicates improved annotation and effective co-alignment .

### F.1    REFERENCE-FREE EVALUATION METRIC

Intuitively, inter-rater reliability (IRR) metrics such as Cohen's kappa and Krippendorff's alpha, which are widely used for assessing the reliability of crowdsourced annotations, may seem sufficient. However, these metrics assume that annotators are equally competent and operate under a fixed candidate label set, the condition that are hardly satisfied in reality. IRR becomes unstable

| Metric | Requires Ground Truth? | Captures Data Drift? | Tracks Alignment Over Time? |
|---|---|---|---|
| Accuracy | ✓ | ✗ | ✗ |
| Precision/Recall | ✓ | ✗ | ✗ |
| F1-score | ✓ | ✗ | ✗ |
| **CAI Ratio** | **Not Required** | ✓ | ✓ |

Table 8: **Comparison of traditional metrics and CAI Ratio.** Traditional metrics require ground-truth labels and therefore mainly support reference-based evaluation. In contrast, CAI Ratio does not require ground-truth labels and is designed to track annotation alignment under data drift and distributional changes.

when the class distribution is skewed, when annotators systematically omit certain classes, under marginal drift caused by heterogeneous annotator competence, or in open-set scenarios where no shared label set exists Wong et al. (2021). Consequently, IRR is less reliable for personalized or subjective tasks, where cultural or training differences amplify annotator variance and bias the results. By contrast, the Consistent-and-Inconsistent (CAI) ratio is label-set agnostic and represents the first reference-free metric designed to quantify annotation reliability by leveraging heterogeneous consistency signals (likelihood-derived and embedding agreement) without relying on references or oracle annotations.

### F.2 COMPARISON WITH TRADITIONAL EVALUATION METRICS

In Table 8 which shows a comparison with CAI ratio and Accuracy, Precision/Recall and F1-score in term of ground-truth label, robustness to Data Drift, and tracking annotation alignment over time.

## G EXPERIMENTAL DETAILS

The top-k selection and proportions of consistent and user-preference samples are as follows. For CLINC and Massive Scenario, 'top-k' is set to 5, with 'proportion' at 0.2. For MTOP Intent, 'proportion' is set to 0.8, and 'top-k' is updated to 15 after printing the current value. In StackExchange, 'top-k' is set to 5 and 'proportion' to 1, while in Banking77, 'top-k' is set to 3 and 'proportion' is 0.2. In massive intent, 'top-k' is 20 and 'proportion' is 0.5), proportion=0.2, and few real nat has top-k=30, and proportion is 1. In 'Reddit', 'top-k' is set to 7, and the proportion is 0.2. All tests are done with two random seeds with temperature parameters (0.5 and 1) for user preference samples, task-specialised model-assigned annotation, and LLMs with and without task-specialised model annotations. We have ran our methods and baselines with two random seeds.

### G.1 INVERSE CONSISTENT (IC) RATIO

The number of samples per class required for human annotation based on user preferences is determined by our Inverse Consistent (IC) ratio equation G.1. For user-preference samples. The $n$ denotes the total size of the consistent sample where $M = n$, and $k$ be the number of classes. The parameter $p$ represents the proportion of samples to be selected and is set to 5% (i.e., $p = 0.05$). In our experiment, we do not use all the identified consistent samples. The proportion of consistent samples used for co-alignment is determined by the IC ratio. Let $n_c$ be the number of consistent samples, so $M = n_c$ represents the size of the consistent sample selection. If the CAI ratio is greater than 0.5 (i.e., the number of consistent samples exceeds inconsistent ones), the value of $p$ will be reduced to use fewer consistent samples. If the CAI ratio is less than 0.4, $p$ is set to 1 (i.e., 100%) since more consistent samples are needed for co-alignment . The formula for the **Inverse Consistent (IC) ratio** is defined as IC $= \left( \frac{M \times p}{k} \right)$.

# H WHY GENERAL LLM AND TASK-SPECIFIC MODEL COLLABORATION IS ESSENTIAL FOR HETEROGENEOUS-CONSISTENCY CO-ALIGNMENT

In semi-supervised learning tasks that rely on user preferences, the key challenge lies in evaluating LLM-generated annotations and enabling mechanisms for co-alignment—especially when the competency of general LLMs is uncertain and no external knowledge is available. To address this, we propose **HCC**, a co-alignment framework that facilitates both self-assessment and refinement of LLM annotations through collaboration with a task-specialized model.

HCC introduces the *Consistent-and-Inconsistent (CAI) Ratio*, which quantifies agreements and disagreements between models to identify consistent and inconsistent samples. This iterative refinement process improves the performance of both the specialized model and the general LLM. To validate our approach, we experiment with *Meta-8B Instruct*, a lightweight LLM representing a low-competency model, and show that its collaboration with a task-specialized model enhances annotation robustness even under noisy conditions.

More broadly, given a semi-supervised learning task where LLM competency is unknown and no external knowledge exists, HCC addresses the fundamental question: *How can we evaluate and enable co-alignment for such datasets?* We answer this through **HCC**, a self-supervised strategy that allows self-correction and self-evaluation of LLM-generated annotations. By combining the CAI ratio with task-specialized model collaboration, HCC systematically improves annotation quality across both models.

### H.0.1 THE ROLE OF THE SPECIALISED MODEL IN HETEROGENEOUS-CONSISTENCY CO-ALIGNMENT

The specialised model plays a critical role in Heterogeneous-Consistency Co-Alignment (HCC) by providing a task-specific reference signal that safeguards the annotation process when the general-purpose LLM underperforms. Rather than relying solely on the LLM's predictions, HCC compares the outputs of the task-specialised model and the LLM to monitor annotation consistency and guide subsequent refinement. In this sense, the specialised model serves as a reference point for *course tracking*: it helps detect when the LLM deviates from task-relevant annotation patterns and provides a complementary signal for correcting such deviations.

This role is particularly important when the LLM has limited task competency. For example, in our experiments with Meta-Llama 3-8B Instruct, the model shows suboptimal performance on several datasets, accompanied by relatively low CAI scores. By incorporating the task-specialised model, HCC identifies heterogeneous consistency between the two models and iteratively refines the annotation process. This collaboration improves robustness even when the general-purpose LLM alone is unreliable for a specific task. As shown in Tables 9 and 10, HCC consistently improves annotation accuracy over baseline methods, including in settings where the LLM has relatively weak task-specific performance. These results demonstrate both the resilience of HCC and the importance of the specialised model in stabilising performance across diverse LLM configurations.

Moreover, recent studies show that LLMs can exhibit overconfidence and may fail to express uncertainty appropriately (Zhou et al., 2024; Xiong et al., 2023). These limitations make sole reliance on LLM-generated annotations risky, especially in low-resource or preference-sensitive settings. The task-specialised model mitigates this issue by introducing an additional task-grounded perspective, enabling HCC to verify, correct, and co-align annotations more reliably.

| Dataset | Specialised Model | LLM Only Llama-3-8B | Specialised + LLM | HCC w/o Corr. | HCC w/ Corr. | CAI Ratio ↑ Before → After |
|---|---|---|---|---|---|---|
| CLINC | $79.01_{\pm1.08}$ | $32.49_{\pm6.73}$ | $69.40_{\pm7.28}$ | $63.41_{\pm3.19}$ | $\mathbf{82.43}_{\pm0.20}$ | $0.56 \to 4.43$ |
| Massive Scenario | $75.55_{\pm1.76}$ | $43.52_{\pm1.85}$ | $66.74_{\pm0.98}$ | $70.06_{\pm1.12}$ | $\mathbf{78.13}_{\pm0.74}$ | $0.67 \to 4.88$ |
| MTOP Intent | $52.49_{\pm2.52}$ | $34.17_{\pm6.70}$ | $48.23_{\pm0.25}$ | $\mathbf{66.39}_{\pm0.70}$ | $63.39_{\pm1.47}$ | $0.35 \to 1.46$ |
| StackExchange | $32.27_{\pm0.65}$ | $11.02_{\pm2.78}$ | $26.26_{\pm2.16}$ | $16.03_{\pm0.13}$ | $\mathbf{38.88}_{\pm0.27}$ | $0.23 \to 0.53$ |
| Banking77 | $73.93_{\pm1.56}$ | $33.06_{\pm1.92}$ | $69.66_{\pm1.74}$ | $64.29_{\pm1.24}$ | $\mathbf{77.71}_{\pm0.25}$ | $0.68 \to 4.20$ |
| Reddit | $51.73_{\pm0.62}$ | $36.31_{\pm0.97}$ | $46.00_{\pm2.51}$ | $40.29_{\pm0.55}$ | $\mathbf{58.81}_{\pm0.28}$ | $0.33 \to 1.58$ |
| FewRel-Nat | $35.35_{\pm0.02}$ | $14.25_{\pm0.36}$ | $30.07_{\pm4.45}$ | $31.80_{\pm0.34}$ | $\mathbf{42.92}_{\pm0.06}$ | $0.13 \to 0.85$ |
| Massive Intent | $61.80_{\pm1.04}$ | $45.41_{\pm0.06}$ | $56.03_{\pm0.08}$ | $67.49_{\pm0.10}$ | $\mathbf{67.75}_{\pm0.43}$ | $0.73 \to 2.87$ |

Table 9: **Annotation accuracy comparison using Meta-Llama 3-8B Instruct.** We report accuracy (%) with standard deviations across eight datasets for the specialised model, zero-shot LLM, specialised-model-guided LLM, and HCC before and after correction. The best result for each dataset is bolded. Highlighted HCC columns emphasize the effect of correction, while the CAI column reports the consistency improvement from before to after correction.

| Dataset | Specialised Model | LLM Only Llama-3-8B | HCC w/ Corr. | Gain over Specialised | Gain over LLM |
|---|---|---|---|---|---|
| CLINC | $79.01_{\pm1.08}$ | $32.49_{\pm6.73}$ | $\mathbf{82.43}_{\pm0.20}$ | +3.42 | +49.94 |
| Massive Scenario | $75.55_{\pm1.76}$ | $43.52_{\pm1.85}$ | $\mathbf{78.13}_{\pm0.74}$ | +2.58 | +34.61 |
| MTOP Intent | $52.49_{\pm2.52}$ | $34.17_{\pm6.70}$ | $\mathbf{63.39}_{\pm1.47}$ | +10.90 | +29.22 |
| StackExchange | $32.27_{\pm0.65}$ | $11.02_{\pm2.78}$ | $\mathbf{38.88}_{\pm0.27}$ | +6.61 | +27.86 |
| Banking77 | $73.93_{\pm1.56}$ | $33.06_{\pm1.92}$ | $\mathbf{77.71}_{\pm0.25}$ | +3.78 | +44.65 |
| Reddit | $51.73_{\pm0.62}$ | $36.31_{\pm0.97}$ | $\mathbf{58.81}_{\pm0.28}$ | +7.08 | +22.50 |
| FewRel-Nat | $35.35_{\pm0.02}$ | $14.25_{\pm0.36}$ | $\mathbf{42.92}_{\pm0.06}$ | +7.57 | +28.67 |
| Massive Intent | $61.80_{\pm1.04}$ | $45.41_{\pm0.06}$ | $\mathbf{67.75}_{\pm0.43}$ | +5.95 | +22.34 |

Table 10: **Improvement of HCC over specialised-model and LLM-only baselines.** We compare corrected HCC against the specialised model and the zero-shot Meta-Llama 3-8B Instruct baseline. Green cells show absolute gains over the specialised model, while blue cells show absolute gains over the LLM-only baseline.

## I SENSITIVITY ANALYSIS OF $K$ ACROSS DIFFERENT SPECIALISED MODELS

To assess the robustness of **_HHC_**, we extend our ablations to three diverse sentence encoders (specialised models) and a small language model (serving as a weak model) collaborating with Llama-3-8B-Instruct (serving as a strong model). We have also included the parameter sizes of all specialised models in Table 11. We have conducted $K$ sensitivity analysis as shown on Table 12

| Backbone | Parameters | Role in Ablation |
|---|---|---|
| MiniLM-L6-v2 | $\sim$22.7M | Small general-purpose SBERT encoder |
| E5-base | $\sim$110M | Strong retrieval-focused sentence encoder |
| GTE-small | $\sim$33.4M | Small high-quality sentence encoder |
| BGE-small-en-v1.5 | $\sim$33.4M | Small retrieval-optimised sentence encoder |
| BERT-base-uncased | $\sim$110M | Small language-model backbone |

Table 11: **Backbones used in the robustness ablation.** We evaluate HCC with four sentence encoders, MiniLM-L6-v2, E5-base, GTE-small, and BGE-small-en-v1.5, together with BERT-base-uncased as a small language-model backbone.

with Specialised Models (We added ablations using **E5-base**, **GTE-small**, and **BGE-small** in addition to the standard BERT baseline.) **K Ablation.** We implemented a full sensitivity study with $K \in \{3, 5, 7, 10\}$ across 8 datasets. Results show **minimal deviation** across $K$, with $K = 3$ or $K = 5$ consistently performing best for fine-grained tasks.

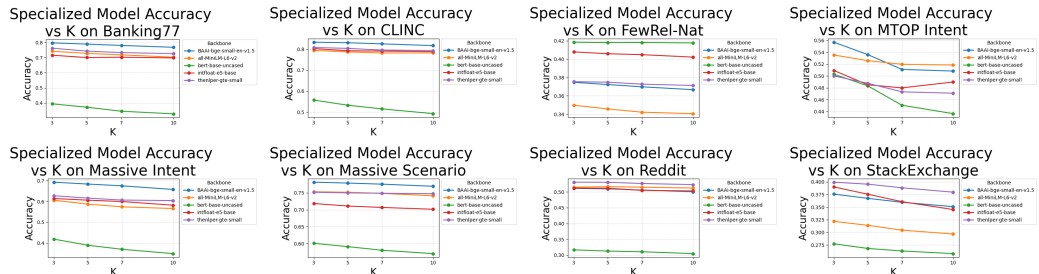

Figure 7: Accuracy of specialised models across datasets for different values of $K$. Performance is compared for five backbone encoders: BERT, BGE, MiniLM, E5, and GTE.

| Dataset | Backbone | K=3 | K=5 | K=7 | K=10 |
|---|---|---|---|---|---|
| **Banking77** | BAAI-bge-small-en-v1.5 | $0.796 \pm 0.002$ | $0.788 \pm 0.002$ | $0.780 \pm 0.006$ | $0.767 \pm 0.014$ |
| | all-MiniLM-L6-v2 | $0.741 \pm 0.006$ | $0.727 \pm 0.003$ | $0.718 \pm 0.004$ | $0.702 \pm 0.004$ |
| | bert-base-uncased | $0.394 \pm 0.010$ | $0.372 \pm 0.018$ | $0.346 \pm 0.025$ | $0.329 \pm 0.020$ |
| | intfloat-e5-base | $0.715 \pm 0.000$ | $0.700 \pm 0.000$ | $0.702 \pm 0.007$ | $0.697 \pm 0.004$ |
| | thenlper-gte-small | $0.761 \pm 0.007$ | $0.742 \pm 0.000$ | $0.732 \pm 0.006$ | $0.726 \pm 0.011$ |
| **CLINC** | BAAI-bge-small-en-v1.5 | $0.833 \pm 0.002$ | $0.830 \pm 0.005$ | $0.825 \pm 0.002$ | $0.817 \pm 0.004$ |
| | all-MiniLM-L6-v2 | $0.794 \pm 0.002$ | $0.787 \pm 0.007$ | $0.781 \pm 0.002$ | $0.782 \pm 0.001$ |
| | bert-base-uncased | $0.557 \pm 0.005$ | $0.533 \pm 0.008$ | $0.515 \pm 0.007$ | $0.493 \pm 0.002$ |
| | intfloat-e5-base | $0.802 \pm 0.007$ | $0.792 \pm 0.001$ | $0.790 \pm 0.010$ | $0.788 \pm 0.015$ |
| | thenlper-gte-small | $0.808 \pm 0.007$ | $0.803 \pm 0.009$ | $0.795 \pm 0.007$ | $0.792 \pm 0.002$ |
| **FewRel-Nat** | BAAI-bge-small-en-v1.5 | $0.375 \pm 0.003$ | $0.372 \pm 0.001$ | $0.370 \pm 0.000$ | $0.367 \pm 0.003$ |
| | all-MiniLM-L6-v2 | $0.350 \pm 0.003$ | $0.346 \pm 0.002$ | $0.342 \pm 0.003$ | $0.341 \pm 0.000$ |
| | bert-base-uncased | $0.419 \pm 0.009$ | $0.418 \pm 0.012$ | $0.418 \pm 0.011$ | $0.418 \pm 0.009$ |
| | intfloat-e5-base | $0.408 \pm 0.006$ | $0.406 \pm 0.009$ | $0.405 \pm 0.008$ | $0.402 \pm 0.009$ |
| | thenlper-gte-small | $0.375 \pm 0.012$ | $0.375 \pm 0.012$ | $0.373 \pm 0.014$ | $0.371 \pm 0.015$ |
| **MTOP Intent** | BAAI-bge-small-en-v1.5 | $0.557 \pm 0.012$ | $0.536 \pm 0.011$ | $0.511 \pm 0.020$ | $0.508 \pm 0.013$ |
| | all-MiniLM-L6-v2 | $0.535 \pm 0.017$ | $0.526 \pm 0.016$ | $0.520 \pm 0.019$ | $0.518 \pm 0.025$ |
| | bert-base-uncased | $0.503 \pm 0.015$ | $0.483 \pm 0.025$ | $0.451 \pm 0.020$ | $0.437 \pm 0.031$ |
| | intfloat-e5-base | $0.509 \pm 0.008$ | $0.485 \pm 0.006$ | $0.480 \pm 0.005$ | $0.490 \pm 0.012$ |
| | thenlper-gte-small | $0.500 \pm 0.014$ | $0.488 \pm 0.015$ | $0.473 \pm 0.023$ | $0.471 \pm 0.036$ |
| **Massive Intent** | BAAI-bge-small-en-v1.5 | $0.692 \pm 0.006$ | $0.683 \pm 0.013$ | $0.675 \pm 0.019$ | $0.657 \pm 0.035$ |
| | all-MiniLM-L6-v2 | $0.606 \pm 0.014$ | $0.587 \pm 0.011$ | $0.574 \pm 0.015$ | $0.566 \pm 0.011$ |
| | bert-base-uncased | $0.419 \pm 0.013$ | $0.390 \pm 0.025$ | $0.370 \pm 0.023$ | $0.350 \pm 0.028$ |
| | intfloat-e5-base | $0.614 \pm 0.002$ | $0.606 \pm 0.002$ | $0.599 \pm 0.001$ | $0.582 \pm 0.005$ |
| | thenlper-gte-small | $0.627 \pm 0.007$ | $0.616 \pm 0.006$ | $0.606 \pm 0.002$ | $0.604 \pm 0.001$ |
| **Massive Scenario** | BAAI-bge-small-en-v1.5 | $0.782 \pm 0.025$ | $0.779 \pm 0.026$ | $0.776 \pm 0.027$ | $0.770 \pm 0.030$ |
| | all-MiniLM-L6-v2 | $0.753 \pm 0.014$ | $0.752 \pm 0.013$ | $0.748 \pm 0.010$ | $0.741 \pm 0.009$ |
| | bert-base-uncased | $0.601 \pm 0.023$ | $0.591 \pm 0.029$ | $0.580 \pm 0.030$ | $0.570 \pm 0.032$ |
| | intfloat-e5-base | $0.718 \pm 0.027$ | $0.711 \pm 0.033$ | $0.707 \pm 0.029$ | $0.701 \pm 0.032$ |
| | thenlper-gte-small | $0.752 \pm 0.022$ | $0.750 \pm 0.015$ | $0.749 \pm 0.016$ | $0.747 \pm 0.015$ |
| **Reddit** | BAAI-bge-small-en-v1.5 | $0.512 \pm 0.005$ | $0.510 \pm 0.004$ | $0.506 \pm 0.006$ | $0.501 \pm 0.005$ |
| | all-MiniLM-L6-v2 | $0.515 \pm 0.002$ | $0.517 \pm 0.001$ | $0.515 \pm 0.003$ | $0.513 \pm 0.000$ |
| | bert-base-uncased | $0.316 \pm 0.006$ | $0.313 \pm 0.005$ | $0.310 \pm 0.009$ | $0.304 \pm 0.008$ |
| | intfloat-e5-base | $0.512 \pm 0.006$ | $0.512 \pm 0.001$ | $0.505 \pm 0.002$ | $0.503 \pm 0.001$ |
| | thenlper-gte-small | $0.531 \pm 0.003$ | $0.531 \pm 0.004$ | $0.527 \pm 0.000$ | $0.523 \pm 0.002$ |
| **StackExchange** | BAAI-bge-small-en-v1.5 | $0.376 \pm 0.006$ | $0.367 \pm 0.011$ | $0.359 \pm 0.009$ | $0.351 \pm 0.007$ |
| | all-MiniLM-L6-v2 | $0.322 \pm 0.006$ | $0.314 \pm 0.007$ | $0.304 \pm 0.009$ | $0.297 \pm 0.011$ |
| | bert-base-uncased | $0.277 \pm 0.005$ | $0.268 \pm 0.003$ | $0.263 \pm 0.006$ | $0.258 \pm 0.009$ |
| | intfloat-e5-base | $0.390 \pm 0.004$ | $0.375 \pm 0.006$ | $0.361 \pm 0.007$ | $0.345 \pm 0.006$ |
| | thenlper-gte-small | $0.399 \pm 0.013$ | $0.395 \pm 0.020$ | $0.388 \pm 0.019$ | $0.379 \pm 0.020$ |

Table 12: Mean $\pm$ standard deviation of specialised model accuracy for different values of $K$ across datasets and backbone models.

# J   ADDITIONAL ABLATION STUDIES BASED ON LLAMA-3-8B-INSTRUCT LARGE LANGUAGE MODEL

## J.1   HCC UNDER WEAKER (BERT-BASE-UNCASED) VS. STRONGER LANGUAGE MODEL (LLAMA-3-8B INSTRUCT) REGIMES

To demonstrate that Heterogeneous Consistency Co-Alignment (HCC) remains effective across both weaker and stronger language model regimes, we adopt `BERT-Base-Uncased` as a weak specialised model and `Llama-3-8B Instruct` as a strong LLM. As shown in Appendix Tables 13 and 14, HCC enables the open-source **Llama-3-8B Instruct**—even when paired with the weaker BERT backbone—to outperform **GPT-4o Mini on five datasets** and **GPT-3.5 Turbo on seven datasets**. This highlights HCC's ability to boost weaker specialised models while narrowing the performance gap between open-source and proprietary LLMs. In this setting, `bert-base-uncased` is used solely for the initial annotation assignment. However, using BERT for the divide–conquer co-alignment stage degrades performance due to its weaker semantic representations. To address this, we adopt a hybrid encoder strategy: during the co-alignment phase, BERT is replaced with **all-MiniLM-L6-v2**, a substantially stronger SBERT encoder.

This hybrid design produces more coherent semantic clusters and significantly improves HCC performance (Tables 13 and 14), often surpassing both GPT-4o Mini and GPT-3.5 Turbo. Thus, BERT is used to initialise label assignments, while MiniLM provides the semantic granularity required for reliable divide–conquer co-alignment.

Table 13: **Performance Comparison: HCC (R2 with BERT-Base-Uncased) vs. Closed-Source Models.** HCC uses only a weak BERT encoder, yet still outperforms GPT models on most datasets. (K=3)

| Dataset | HCC (BERT) | GPT-4o Mini | GPT-3.5 Turbo | Outcome |
|---|---|---|---|---|
| CLINC | **0.8240** | 0.8144 | 0.6658 | **Beats Both** |
| Massive Scenario | **0.7461** | 0.6683 | 0.6089 | **Beats Both** |
| Banking77 | **0.7162** | 0.6512 | 0.6512 | **Beats Both** |
| Reddit | **0.5813** | 0.5740 | 0.5112 | **Beats Both** |
| FewRel-Nat | **0.4205** | 0.3587 | 0.3287 | **Beats Both** |
| MTop Intent | 0.6562 | *0.7503* | 0.6495 | **Beats GPT-3.5 Only** |
| StackExchange | 0.3941 | *0.5190* | 0.3010 | **Beats GPT-3.5 Only** |
| Massive Intent | 0.6261 | *0.7693* | 0.7152 | *Lower than both* |

Table 14: **Performance Comparison: HCC (R2 with BERT-Base-Uncased) vs. Closed-Source Models.** HCC uses only a weak BERT encoder, yet outperforms GPT models on most datasets. (K=10)

| Dataset | HCC (BERT) | GPT-4o Mini | GPT-3.5 Turbo | Outcome |
|---|---|---|---|---|
| CLINC | **0.7958** | 0.8144 | 0.6658 | **Beats GPT-3.5** |
| Massive Scenario | **0.7576** | 0.6683 | 0.6089 | **Beats Both** |
| Banking77 | **0.7234** | 0.6512 | 0.6512 | **Beats Both** |
| Reddit | 0.4085 | 0.5740 | 0.5112 | *Lower than both* |
| FewRel-Nat | **0.4292** | 0.3587 | 0.3287 | **Beats Both** |
| MTop Intent | 0.5857 | *0.7503* | 0.6495 | **Beats GPT-3.5 Only** |
| StackExchange | 0.3898 | *0.5190* | 0.3010 | **Beats GPT-3.5 Only** |
| Massive Intent | 0.6187 | *0.7693* | 0.7152 | *Lower than both* |

KEY FINDINGS:

1. On **CLINC**, **Banking77**, **Massive Scenario**, **Reddit**, and **FewRel**, the HCC framework is so effective that it endows a (standard BERT model + Llama-3-8B) to surpass the annotation quality of GPT-3.5 and GPT-4o Mini.

## J.2   HCC ACROSS EMBEDDING BACKBONES AND SMALL LANGUAGE MODEL

The comparison of HCC across the best embedding backbones with ChatGPT-4o Mini and ChatGPT-3.5 Turbo is presented in Tables 18 and 17. Tables 19 and 20 further illustrate the performance of the HCC paradigm using different embedding backbones, where the best encoder is

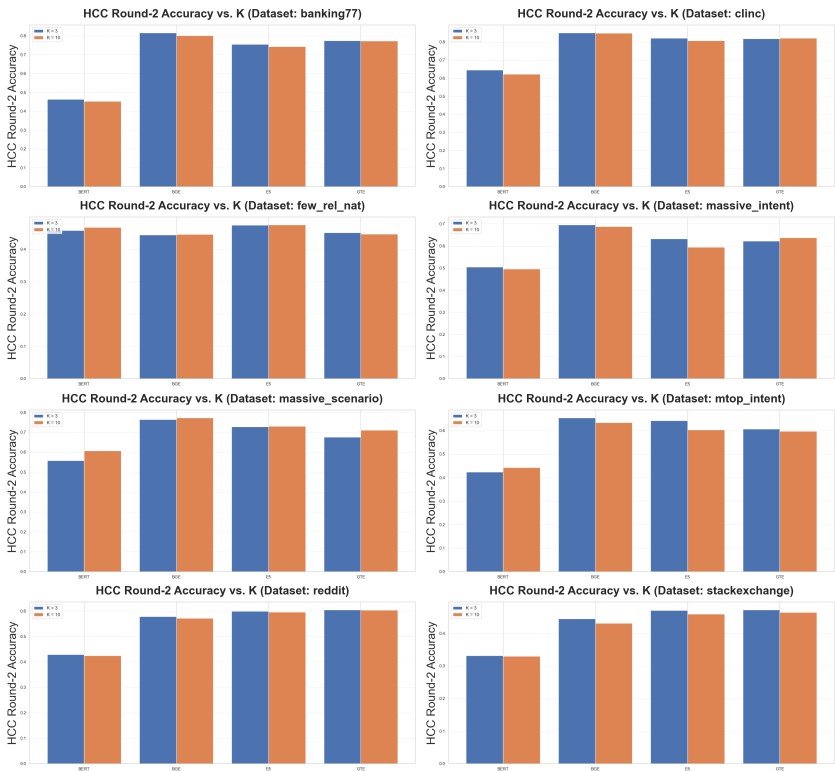

Figure 8: Accuracy of HCC (R2) across datasets for different values of $K$ (3,10) on Llama3-8 Instruct Model. Performance is compared for four backbone encoders: BERT, BGE, E5, and GTE.

selected for each dataset in conjunction with Llama-3-8B. We compare this configuration against GPT-4o Mini and GPT-3.5 Turbo to highlight its robustness and competitiveness. Our ablation studies in Section J.1 and Section J.2 show that HCC is **moderately robust** even when the specialised model is weak. For instance, using BERT as the specialised encoder, HCC still surpasses both standalone specialised models and standalone LLMs, regardless of whether clustering is performed with or without intent information. This indicates that HCC can compensate for imperfect cluster structures produced by weaker encoders. At the same time, the results show that HCC benefits consistently from **stronger embedding encoders**. Models such as E5, GTE, and BGE yield progressively higher accuracy, demonstrating that improvements in representation quality translate directly into more reliable clustering and, consequently, stronger co-alignment performance. Overall, under the standard K = 3 and k=10 settings, HCC remains stable across both weak and strong specialised models, confirming that it is broadly effective for categorical NLU annotation tasks while still offering additional gains when higher-quality encoders are available.

### J.3 CAI RATIO ACROSS EMBEDDING BACKBONES (BEFORE/AFTER HCC)

Table 19 and 20 shows how the **CAI Ratio** behaves when pairing **Llama-3-8B** with four specialized embedding backbones of varying capacities—**BERT** (weak), **GTE-small** (moderate), **E5-base** (strong), and **BGE-small-en-1.5** (strong). For each dataset–backbone configuration, we report the *Initial CAI* (before HCC) and *Final CAI* (after HCC), along with accuracy under Zero-Shot, Single-Shot, Specialized, and HCC (R2) settings. Two clear trends emerge from the results.

First, the **Final CAI** (after co-alignment) is consistently higher than the **Initial CAI** across all datasets and all backbones, mirroring the performance improvements achieved by HCC. In particular, the weakest model—**BERT**—shows the largest CAI gains, with improvements substantially higher than those of the stronger backbones. This reflects that HCC is effective across both weak and strong model settings, and that CAI accurately captures the improvement after co-alignment.

Table 15: Averaged performance over seeds $\{1, 42\}$ for $K \in \{3\}$, reported as mean $\pm$ standard deviation for each Dataset–Backbone pair.

| Dataset | Backbone | Zero-Shot | One-Shot | Initial CAI | Final CAI | R1 | R2 | Specialized |
|---|---|---|---|---|---|---|---|---|
| banking77 | BAAI-bge-small-en-v1.5 | 0.3365 ± 0.0039 | 0.6943 ± 0.0103 | 0.4950 ± 0.0071 | 4.6200 ± 0.0283 | **0.8130** ± 0.0005 | 0.8161 ± 0.0039 | 0.7964 ± 0.0023 |
| banking77 | bert-base-uncased | 0.1804 ± 0.0034 | 0.3610 ± 0.0202 | 0.1700 ± 0.0283 | 0.6950 ± 0.0212 | 0.4373 ± 0.0101 | **0.4628** ± 0.0057 | 0.3945 ± 0.0096 |
| banking77 | intfloat-e5-base | 0.3218 ± 0.0174 | 0.6273 ± 0.0051 | 0.4600 ± 0.0424 | 2.9100 ± 0.1838 | 0.5235 ± 0.0305 | **0.7544** ± 0.0158 | 0.7154 ± 0.0002 |
| banking77 | thenlper-gte-small | 0.3185 ± 0.0087 | 0.6619 ± 0.0140 | 0.4700 ± 0.0141 | 3.8150 ± 0.1768 | **0.7763** ± 0.0046 | 0.7747 ± 0.0041 | 0.7614 ± 0.0073 |
| clinc | BAAI-bge-small-en-v1.5 | 0.2823 ± 0.0184 | 0.7168 ± 0.0055 | 0.3650 ± 0.0354 | 5.3350 ± 0.5020 | **0.8559** ± 0.0024 | 0.8511 ± 0.0013 | 0.8329 ± 0.0022 |
| clinc | bert-base-uncased | 0.2026 ± 0.0306 | 0.4829 ± 0.0079 | 0.1700 ± 0.0141 | 1.1750 ± 0.0071 | 0.6264 ± 0.0066 | **0.6454** ± 0.0077 | 0.5572 ± 0.0046 |
| clinc | intfloat-e5-base | 0.3023 ± 0.0360 | 0.6870 ± 0.0036 | 0.4000 ± 0.0990 | 4.3700 ± 0.3253 | 0.8144 ± 0.0041 | **0.8211** ± 0.0041 | 0.8021 ± 0.0071 |
| clinc | thenlper-gte-small | 0.2761 ± 0.0071 | 0.7018 ± 0.0195 | 0.3350 ± 0.0071 | 4.1600 ± 0.0566 | **0.8242** ± 0.0031 | 0.8182 ± 0.0091 | 0.8076 ± 0.0066 |
| few_rel_nat | BAAI-bge-small-en-v1.5 | 0.1269 ± 0.0039 | 0.3522 ± 0.0060 | 0.1350 ± 0.0071 | 0.8850 ± 0.0212 | 0.4443 ± 0.0106 | **0.4445** ± 0.0008 | 0.3748 ± 0.0028 |
| few_rel_nat | bert-base-uncased | 0.1254 ± 0.0098 | 0.3922 ± 0.0047 | 0.1550 ± 0.0212 | 1.0150 ± 0.0071 | 0.4589 ± 0.0071 | **0.4590** ± 0.0074 | 0.4185 ± 0.0092 |
| few_rel_nat | intfloat-e5-base | 0.1260 ± 0.0024 | 0.3879 ± 0.0022 | 0.1450 ± 0.0071 | 0.9900 ± 0.0141 | 0.4694 ± 0.0161 | **0.4749** ± 0.0125 | 0.4079 ± 0.0062 |
| few_rel_nat | thenlper-gte-small | 0.1252 ± 0.0104 | 0.3499 ± 0.0118 | 0.1350 ± 0.0071 | 0.8750 ± 0.0212 | 0.4496 ± 0.0025 | **0.4516** ± 0.0003 | 0.3754 ± 0.0117 |
| massive_intent | BAAI-bge-small-en-v1.5 | 0.3653 ± 0.0026 | 0.6685 ± 0.0100 | 0.6200 ± 0.0000 | 3.5800 ± 0.2970 | **0.6964** ± 0.0005 | 0.6957 ± 0.0124 | 0.6923 ± 0.0057 |
| massive_intent | bert-base-uncased | 0.2270 ± 0.0266 | 0.4080 ± 0.0097 | 0.2400 ± 0.0424 | 1.0050 ± 0.0495 | **0.5108** ± 0.0124 | 0.5047 ± 0.0052 | 0.4191 ± 0.0126 |
| massive_intent | intfloat-e5-base | 0.3245 ± 0.0119 | 0.5876 ± 0.0207 | 0.5150 ± 0.0212 | 2.4300 ± 0.0566 | 0.6138 ± 0.0121 | **0.6323** ± 0.0155 | 0.6140 ± 0.0019 |
| massive_intent | thenlper-gte-small | 0.3265 ± 0.0223 | 0.6083 ± 0.0162 | 0.4950 ± 0.0354 | 2.3450 ± 0.1626 | 0.5752 ± 0.0801 | 0.6226 ± 0.0188 | **0.6271** ± 0.0071 |
| massive_scenario | BAAI-bge-small-en-v1.5 | 0.4469 ± 0.0019 | 0.7076 ± 0.0007 | 0.8050 ± 0.0212 | 5.2000 ± 0.6364 | **0.7932** ± 0.0138 | 0.7634 ± 0.0140 | 0.7816 ± 0.0250 |
| massive_scenario | BAAI-bge-small-en-v1.5 | 0.4469 ± 0.0019 | 0.7076 ± 0.0007 | 0.8050 ± 0.0212 | 5.2000 ± 0.6364 | **0.7932** ± 0.0138 | 0.7634 ± 0.0140 | 0.7816 ± 0.0250 |
| massive_scenario | bert-base-uncased | 0.4144 ± 0.0136 | 0.5521 ± 0.0266 | 0.5500 ± 0.0424 | 1.6350 ± 0.8556 | **0.6686** ± 0.0017 | 0.5585 ± 0.1393 | 0.6007 ± 0.0231 |
| massive_scenario | intfloat-e5-base | 0.4412 ± 0.0024 | 0.6500 ± 0.0204 | 0.7600 ± 0.0141 | 3.9500 ± 0.7354 | **0.7406** ± 0.0216 | 0.7280 ± 0.0100 | 0.7184 ± 0.0273 |
| massive_scenario | thenlper-gte-small | 0.4438 ± 0.0043 | 0.6765 ± 0.0252 | 0.8000 ± 0.0141 | 3.4350 ± 1.6051 | **0.7618** ± 0.0088 | 0.6750 ± 0.0934 | 0.7518 ± 0.0223 |
| mtop_intent | BAAI-bge-small-en-v1.5 | 0.2707 ± 0.0179 | 0.5618 ± 0.0100 | 0.2350 ± 0.0071 | 1.6100 ± 0.0990 | **0.6567** ± 0.0221 | 0.6545 ± 0.0318 | 0.5569 ± 0.0124 |
| mtop_intent | bert-base-uncased | 0.2441 ± 0.0015 | 0.5008 ± 0.0047 | 0.2100 ± 0.0000 | 0.7050 ± 0.5869 | **0.5878** ± 0.0406 | 0.4235 ± 0.2304 | 0.5030 ± 0.0155 |
| mtop_intent | intfloat-e5-base | 0.2513 ± 0.0123 | 0.5165 ± 0.0095 | 0.1950 ± 0.0071 | 1.2800 ± 0.1414 | 0.6306 ± 0.0300 | **0.6420** ± 0.0345 | 0.5092 ± 0.0079 |
| mtop_intent | thenlper-gte-small | 0.2738 ± 0.0058 | 0.5131 ± 0.0005 | 0.2150 ± 0.0354 | 1.2400 ± 0.1697 | 0.6009 ± 0.0166 | **0.6072** ± 0.0048 | 0.5001 ± 0.0137 |
| reddit | BAAI-bge-small-en-v1.5 | 0.3004 ± 0.0022 | 0.4997 ± 0.0007 | 0.3250 ± 0.0071 | 1.5800 ± 0.0990 | 0.4254 ± 0.0015 | **0.5779** ± 0.0092 | 0.5115 ± 0.0051 |
| reddit | bert-base-uncased | 0.2016 ± 0.0033 | 0.3202 ± 0.0084 | 0.1050 ± 0.0071 | 0.6300 ± 0.0000 | 0.2850 ± 0.0141 | **0.4288** ± 0.0125 | 0.3164 ± 0.0057 |
| reddit | intfloat-e5-base | 0.2955 ± 0.0011 | 0.5009 ± 0.0152 | 0.3000 ± 0.0141 | 1.4800 ± 0.0707 | 0.4304 ± 0.0029 | **0.5985** ± 0.0024 | 0.5121 ± 0.0059 |
| reddit | thenlper-gte-small | 0.3043 ± 0.0233 | 0.5219 ± 0.0114 | 0.3300 ± 0.0283 | 1.6600 ± 0.0707 | 0.4405 ± 0.0053 | **0.6048** ± 0.0002 | 0.5311 ± 0.0033 |
| stackexchange | BAAI-bge-small-en-v1.5 | 0.1109 ± 0.0082 | 0.3307 ± 0.0111 | 0.1250 ± 0.0071 | 0.6650 ± 0.0071 | 0.4430 ± 0.0051 | **0.4449** ± 0.0014 | 0.3745 ± 0.0077 |
| stackexchange | bert-base-uncased | 0.0929 ± 0.0003 | 0.2489 ± 0.0090 | 0.0700 ± 0.0000 | 0.4150 ± 0.0212 | 0.3294 ± 0.0017 | **0.3314** ± 0.0097 | 0.2774 ± 0.0048 |
| stackexchange | intfloat-e5-base | 0.1193 ± 0.0088 | 0.3405 ± 0.0102 | 0.1400 ± 0.0141 | 0.6950 ± 0.0071 | 0.4635 ± 0.0036 | **0.4706** ± 0.0099 | 0.3898 ± 0.0044 |
| stackexchange | thenlper-gte-small | 0.1263 ± 0.0054 | 0.3584 ± 0.0083 | 0.1500 ± 0.0141 | 0.7500 ± 0.0424 | 0.4669 ± 0.0100 | **0.4723** ± 0.0034 | 0.3992 ± 0.0133 |

Table 16: Averaged performance over seeds $\{1, 42\}$ for $K = 10$, reported as mean $\pm$ standard deviation for each Dataset–Backbone pair.

| Dataset | Backbone | Zero-Shot | One-Shot | Initial CAI | Final CAI | R1 | R2 | Specialized |
|---|---|---|---|---|---|---|---|---|
| banking77 | BAAI-bge-small-en-v1.5 | 0.3435 ± 0.0078 | 0.6687 ± 0.0370 | 0.5250 ± 0.0495 | 4.3350 ± 0.2333 | 0.7976 ± 0.0090 | **0.8016** ± 0.0005 | 0.7674 ± 0.0135 |
| banking77 | bert-base-uncased | 0.1758 ± 0.0090 | 0.2977 ± 0.0248 | 0.1550 ± 0.0495 | 0.5250 ± 0.0495 | 0.4487 ± 0.0005 | **0.4534** ± 0.0002 | 0.3286 ± 0.0202 |
| banking77 | intfloat-e5-base | 0.2919 ± 0.0138 | 0.6008 ± 0.0191 | 0.4000 ± 0.0141 | 2.5950 ± 0.0919 | 0.7404 ± 0.0044 | **0.7433** ± 0.0021 | 0.6974 ± 0.0041 |
| banking77 | thenlper-gte-small | 0.3179 ± 0.0239 | 0.6407 ± 0.0090 | 0.4500 ± 0.0283 | 3.1900 ± 0.1131 | 0.7677 ± 0.0009 | **0.7734** ± 0.0179 | 0.7256 ± 0.0106 |
| clinc | BAAI-bge-small-en-v1.5 | 0.2747 ± 0.0066 | 0.7048 ± 0.0036 | 0.3450 ± 0.0212 | 4.3050 ± 0.1344 | 0.8381 ± 0.0046 | **0.8490** ± 0.0042 | 0.8166 ± 0.0036 |
| clinc | bert-base-uncased | 0.1586 ± 0.0156 | 0.4353 ± 0.0038 | 0.1250 ± 0.0071 | 0.9250 ± 0.0212 | 0.6151 ± 0.0044 | **0.6221** ± 0.0112 | 0.4927 ± 0.0019 |
| clinc | intfloat-e5-base | 0.2430 ± 0.0014 | 0.6889 ± 0.0189 | 0.2900 ± 0.0141 | 3.4700 ± 0.0000 | **0.8231** ± 0.0116 | 0.8083 ± 0.0011 | 0.7883 ± 0.0146 |
| clinc | thenlper-gte-small | 0.2602 ± 0.0053 | 0.6838 ± 0.0094 | 0.3300 ± 0.0283 | 3.8250 ± 0.1061 | 0.8213 ± 0.0148 | **0.8214** ± 0.0002 | 0.7920 ± 0.0019 |
| few_rel_nat | BAAI-bge-small-en-v1.5 | 0.1201 ± 0.0057 | 0.3408 ± 0.0022 | 0.1250 ± 0.0071 | 0.8900 ± 0.0566 | 0.4324 ± 0.0073 | **0.4467** ± 0.0095 | 0.3667 ± 0.0028 |
| few_rel_nat | bert-base-uncased | 0.1269 ± 0.0024 | 0.3919 ± 0.0077 | 0.1450 ± 0.0071 | 1.0550 ± 0.0071 | **0.4706** ± 0.0046 | 0.4684 ± 0.0065 | 0.4177 ± 0.0087 |
| few_rel_nat | intfloat-e5-base | 0.1285 ± 0.0002 | 0.3732 ± 0.0003 | 0.1450 ± 0.0212 | 0.9950 ± 0.0495 | **0.4763** ± 0.0057 | 0.4761 ± 0.0025 | 0.4022 ± 0.0092 |
| few_rel_nat | thenlper-gte-small | 0.1251 ± 0.0046 | 0.3504 ± 0.0066 | 0.1350 ± 0.0071 | 0.9100 ± 0.0283 | **0.4541** ± 0.0014 | 0.4477 ± 0.0043 | 0.3713 ± 0.0150 |
| massive_intent | BAAI-bge-small-en-v1.5 | 0.3741 ± 0.0112 | 0.6180 ± 0.0457 | 0.6450 ± 0.0354 | 3.4400 ± 0.1273 | 0.6720 ± 0.0121 | **0.6881** ± 0.0040 | 0.6574 ± 0.0352 |
| massive_intent | bert-base-uncased | 0.2095 ± 0.0152 | 0.3497 ± 0.0342 | 0.1950 ± 0.0495 | 0.7900 ± 0.1414 | **0.5089** ± 0.0021 | 0.4955 ± 0.0292 | 0.3495 ± 0.0278 |
| massive_intent | intfloat-e5-base | 0.2914 ± 0.0050 | 0.5496 ± 0.0140 | 0.4050 ± 0.0212 | 1.8850 ± 0.4596 | **0.6374** ± 0.0078 | 0.5947 ± 0.0411 | 0.5819 ± 0.0045 |
| massive_intent | thenlper-gte-small | 0.3267 ± 0.0050 | 0.5856 ± 0.0055 | 0.4950 ± 0.0354 | 2.5100 ± 0.0141 | **0.6478** ± 0.0078 | 0.6370 ± 0.0017 | 0.6042 ± 0.0014 |
| massive_scenario | BAAI-bge-small-en-v1.5 | 0.4509 ± 0.0043 | 0.6843 ± 0.0347 | 0.7900 ± 0.0000 | 5.0650 ± 0.0778 | **0.7860** ± 0.0036 | 0.7727 ± 0.0043 | 0.7695 ± 0.0302 |
| massive_scenario | bert-base-uncased | 0.4333 ± 0.0036 | 0.5311 ± 0.0231 | 0.5650 ± 0.0495 | 1.7700 ± 0.4384 | **0.6407** ± 0.0117 | 0.6071 ± 0.0621 | 0.5701 ± 0.0316 |
| massive_scenario | intfloat-e5-base | 0.4544 ± 0.0078 | 0.6316 ± 0.0226 | 0.7950 ± 0.0071 | 4.0550 ± 0.7000 | **0.7416** ± 0.0197 | 0.7300 ± 0.0138 | 0.7014 ± 0.0319 |
| massive_scenario | thenlper-gte-small | 0.4457 ± 0.0316 | 0.6675 ± 0.0038 | 0.8200 ± 0.1131 | 3.7700 ± 0.7495 | 0.7461 ± 0.0014 | 0.7103 ± 0.0178 | **0.7473** ± 0.0150 |
| mtop_intent | BAAI-bge-small-en-v1.5 | 0.2681 ± 0.0164 | 0.5136 ± 0.0300 | 0.2100 ± 0.0424 | 1.3250 ± 0.2333 | 0.6317 ± 0.0330 | **0.6335** ± 0.0685 | 0.5082 ± 0.0129 |
| mtop_intent | bert-base-uncased | 0.2329 ± 0.0289 | 0.4477 ± 0.0324 | 0.1600 ± 0.0424 | 0.6000 ± 0.4525 | **0.5595** ± 0.0409 | 0.4424 ± 0.2384 | 0.4365 ± 0.0314 |
| mtop_intent | intfloat-e5-base | 0.2749 ± 0.0015 | 0.4895 ± 0.0081 | 0.1950 ± 0.0212 | 1.2050 ± 0.0636 | 0.6008 ± 0.0148 | **0.6037** ± 0.0245 | 0.4897 ± 0.0123 |
| mtop_intent | thenlper-gte-small | 0.2621 ± 0.0176 | 0.4930 ± 0.0392 | 0.2000 ± 0.0283 | 1.2050 ± 0.1344 | 0.5920 ± 0.0263 | **0.5975** ± 0.0311 | 0.4709 ± 0.0360 |
| reddit | BAAI-bge-small-en-v1.5 | 0.2902 ± 0.0090 | 0.4916 ± 0.0037 | 0.3100 ± 0.0000 | 1.6400 ± 0.1273 | 0.4210 ± 0.0024 | **0.5712** ± 0.0081 | 0.5008 ± 0.0048 |
| reddit | bert-base-uncased | 0.1862 ± 0.0075 | 0.3113 ± 0.0112 | 0.1050 ± 0.0000 | 0.6150 ± 0.0212 | 0.2824 ± 0.0143 | **0.4248** ± 0.0125 | 0.3045 ± 0.0077 |
| reddit | intfloat-e5-base | 0.2976 ± 0.0121 | 0.4969 ± 0.0011 | 0.3050 ± 0.0212 | 1.4700 ± 0.0424 | 0.4287 ± 0.0057 | **0.5962** ± 0.0013 | 0.5033 ± 0.0013 |
| reddit | thenlper-gte-small | 0.3129 ± 0.0112 | 0.5151 ± 0.0053 | 0.3500 ± 0.0141 | 1.7150 ± 0.0354 | 0.4399 ± 0.0031 | **0.6032** ± 0.0015 | 0.5233 ± 0.0020 |
| stackexchange | BAAI-bge-small-en-v1.5 | 0.1079 ± 0.0090 | 0.3130 ± 0.0136 | 0.1100 ± 0.0283 | 0.6100 ± 0.0424 | **0.4397** ± 0.0012 | 0.4311 ± 0.0049 | 0.3496 ± 0.0088 |
| stackexchange | bert-base-uncased | 0.0929 ± 0.0102 | 0.2336 ± 0.0160 | 0.0600 ± 0.0141 | 0.3800 ± 0.0141 | 0.3277 ± 0.0054 | **0.3301** ± 0.0054 | 0.2579 ± 0.0092 |
| stackexchange | intfloat-e5-base | 0.1079 ± 0.0124 | 0.3021 ± 0.0039 | 0.1200 ± 0.0141 | 0.5850 ± 0.0071 | **0.4706** ± 0.0065 | 0.4590 ± 0.0141 | 0.3450 ± 0.0065 |
| stackexchange | thenlper-gte-small | 0.1216 ± 0.0032 | 0.3328 ± 0.0228 | 0.1300 ± 0.0000 | 0.7050 ± 0.0495 | 0.4635 ± 0.0053 | **0.4647** ± 0.0053 | 0.3795 ± 0.0197 |

Second, after applying HCC, the **Final CAI exhibits a strong and stable correlation** with the resulting co-aligned accuracy. This indicates that HCC not only improves the quality of the assigned annotations, but also enhances the discriminative power of CAI. Consequently, CAI serves as a **robust, backbone-agnostic reliability signal** for assessing LLM annotation quality at scale. Likewise, the **Initial CAI** accurately reflects baseline LLM performance across different backbone–dataset combinations, demonstrating that CAI is sensitive to both model quality and task difficulty.

In terms of effectiveness, HCC achieves the highest accuracy among all three settings, consistently outperforming the Zero-Shot, Single-Shot, and Specialized baselines. Overall, this ablation demonstrates that the **CAI Ratio is both embedding-agnostic and model-agnostic**: when Llama-3-8B serves as the primary annotator, HCC reliably improves both CAI and annotation accuracy across

Table 17: **Performance Comparison: HCC (R2-Best Among All Specialised Model Based on Table 19) vs. Closed-Source Models. We compare the best HCC configuration (using the optimal backbone per dataset) against GPT-4o Mini and GPT-3.5 Turbo. HCC outperforms both closed-source models on the majority of datasets. (K=3).**

| Dataset | HCC (R2) | 4o Mini | 3.5 Turbo | HCC > 4o mini | HCC > 3.5 Turbo |
|---|---|---|---|---|---|
| **CLINC** | **83.98** | 81.44 | 66.58 | **Yes** | **Yes** |
| **Massive Scenario** | **79.49** | 66.83 | 60.89 | **Yes** | **Yes** |
| **MTop Intent** | 67.76 | **75.03** | 64.95 | No | Slightly |
| **StackExchange** | 47.74 | **51.90** | 30.10 | No | **Yes** |
| **Banking77** | **80.68** | 65.12 | 65.12 | **Yes** | **Yes** |
| **Reddit** | **60.62** | 57.40 | 51.12 | **Yes** | **Yes** |
| **FewRel-Nat** | **47.46** | 35.87 | 32.87 | **Yes** | **Yes** |
| **Massive Intent** | 66.88 | **76.93** | 71.52 | No | No |

Table 18: **Performance Comparison: HCC (R2-Best Among All Specialised Models Based on Table 20) vs. Closed-Source Models.We compare the best HCC configuration (using the optimal backbone per dataset) against GPT-4o Mini and GPT-3.5 Turbo. HCC outperforms both closed-source models on the majority of datasets.(K=10).**

| Dataset | HCC (R2) | 4o Mini | 3.5 Turbo | HCC > 4o mini | HCC > 3.5 Turbo |
|---|---|---|---|---|---|
| **CLINC** | **85.02** | 81.44 | 66.58 | **Yes** | **Yes** |
| **Massive Scenario** | **79.62** | 66.83 | 60.89 | **Yes** | **Yes** |
| **MTop Intent** | 66.71 | **75.03** | 64.95 | No | **Yes** |
| **StackExchange** | 47.28 | **51.90** | 30.10 | No | **Yes** |
| **Banking77** | **82.18** | 65.12 | 65.12 | **Yes** | **Yes** |
| **Reddit** | 43.43 | **57.40** | **51.12** | No | No |
| **FewRel-Nat** | **47.86** | 35.87 | 32.87 | **Yes** | **Yes** |
| **Massive Intent** | 69.33 | **76.93** | **71.52** | No | No |

most embedding backbones. This confirms the robustness of our verification (CAI) and refinement (HCC) pipeline, and establishes CAI as a stable, reference-free reliability signal for large-scale annotation.

### J.4 STATISTICAL VALIDATION OF CAI EFFECTIVENESS ON ACROSS SENTENCE ENCODER AND LANGUAGE MODEL

To further validate the reliability of the CAI Ratio as a reference-free quality signal, we conducted a Pearson correlation test between (1) *Initial CAI* and the LLM's *Zero-Shot* and *Single-Shot* accuracies, and (2) *Final CAI* and the *HCC Co-Aligned* accuracy. As show in the Table 21 and Table 22 across all embedding backbones (BERT, GTE-small, E5-base, BGE-small-en-1.5) and all eight datasets, **Initial CAI shows a moderate positive correlation** with both Zero-Shot and Single-Shot Llama-3-8B accuracy, indicating that raw agreement between the encoders and the LLM reflects the baseline reliability of the model. After applying HCC, **Final CAI exhibits a strong and consistently higher correlation** with the resulting co-aligned accuracy, demonstrating that the HCC refinement process both increases CAI and strengthens its alignment with true correctness. These results confirm that CAI becomes a **more discriminative and robust predictor of LLM output quality** after co-alignment, supporting its use as a backbone-agnostic reliability indicator for large-scale annotation.

### J.5 RUNNING TIME AND TOKEN COST

We measured the runtime and token cost of each configuration, as summarized in Table 23, which reports the LLM running time and token usage per 10,000 examples under both **zero-shot** and **single-shot (group prompting)** settings using Llama-3-8B Instruct.

### J.6 HCC RUNTIME FOR CO-ALIGNMENT

To evaluate the efficiency of HCC, as shown in Table 24, we measured the end-to-end latency of the **Refinement Phase** (DCCA) using **Llama-3-8B Instruct**. The reported runtime values are averaged across **four sentence encoders** (E5-base, GTE-small, BGE-small, and BERT-base).

It is important to note:

| Dataset | Backbone | Initial CAI | Final CAI | LLM (Zero-Shot) | LLM (Single-Shot) | Specialized | HCC (R2) Acc. |
|---|---|---|---|---|---|---|---|
| CLINC | BERT | 0.14 | 1.17 | 18.58 | 42.80 | 56.04 | 82.40 |
| | E5 | 0.30 | 4.31 | 26.18 | 61.51 | 79.71 | 83.11 |
| | GTE | 0.28 | 4.10 | 27.31 | 60.18 | 80.29 | 81.87 |
| | BGE | 0.28 | 4.67 | 29.27 | 60.27 | 83.44 | **83.98** |
| Massive Scenario | BERT | 0.55 | 1.90 | 43.31 | 53.67 | 61.70 | 74.61 |
| | E5 | 0.64 | 4.08 | 45.16 | 57.73 | 73.77 | 75.66 |
| | GTE | 0.75 | 4.80 | 45.93 | 62.78 | 76.77 | 77.14 |
| | BGE | 0.72 | 5.59 | 46.57 | 62.58 | **79.93** | 79.49 |
| Mtop Intent | BERT | 0.26 | 0.99 | 32.15 | 45.55 | 49.20 | **65.32** |
| | E5 | 0.26 | 1.21 | 33.58 | 47.04 | 50.36 | 58.30 |
| | GTE | 0.29 | 1.26 | 37.28 | 46.58 | 49.04 | 62.47 |
| | BGE | 0.30 | 1.70 | 35.59 | 47.61 | 54.81 | **67.76** |
| StackExchange | BERT | 0.07 | 0.30 | 10.18 | 26.59 | 28.08 | 39.41 |
| | E5 | 0.13 | 0.72 | 11.21 | 36.93 | 39.29 | **47.74** |
| | GTE | 0.15 | 0.77 | 12.22 | 38.93 | 40.86 | 46.10 |
| | BGE | 0.11 | 0.66 | 10.68 | 36.55 | 37.99 | 43.31 |
| Banking77 | BERT | 0.16 | 0.70 | 17.99 | 34.68 | 38.77 | 71.62 |
| | E5 | 0.46 | 2.74 | 32.53 | 63.08 | 71.53 | 74.58 |
| | GTE | 0.47 | 3.62 | 31.95 | 67.18 | 76.66 | 77.08 |
| | BGE | 0.50 | 4.78 | 34.55 | 68.70 | 79.81 | **80.68** |
| Reddit | BERT | 0.13 | 0.56 | 21.17 | 32.36 | 31.24 | 58.13 |
| | E5 | 0.30 | 1.44 | 30.34 | 50.23 | 50.79 | 60.12 |
| | GTE | 0.32 | 1.64 | 29.87 | 52.60 | 52.88 | **60.62** |
| | BGE | 0.33 | 1.51 | 30.15 | 50.73 | 50.79 | 56.79 |
| FewRel-Nat | BERT | 0.12 | 0.77 | 12.21 | 34.96 | 41.21 | 42.05 |
| | E5 | 0.13 | 0.99 | 12.77 | 35.31 | 40.36 | **47.46** |
| | GTE | 0.12 | 0.85 | 11.99 | 31.23 | 36.72 | 44.31 |
| | BGE | 0.14 | 0.88 | 13.33 | 32.99 | 37.28 | 44.84 |
| Massive Intent | BERT | 0.21 | 0.86 | 20.07 | 40.85 | 41.02 | 62.61 |
| | E5 | 0.54 | 2.45 | 33.62 | 58.14 | 61.53 | 63.99 |
| | GTE | 0.52 | 2.56 | 32.41 | 60.05 | 62.21 | 63.25 |
| | BGE | 0.55 | 3.07 | 34.10 | 67.05 | **69.64** | 66.88 |

Table 19: **Ablation Study across Embedding Backbones and a Small Language Model (Llama8-3 Instruct).** We compare the performance of HCC using three sentence encoders (E5, GTE, BGE) and a small language model (BERT) across eight datasets. *Initial CAI* and *Final CAI* show how the verification signal strengthens as model quality improves. *HCC (R2)* consistently outperforms the Zero-Shot, Single-Shot, and Specialized baselines in most configurations, with the exception of the BGE backbone on Massive Intent and Massive Scenario. (K=3)

| Dataset | Backbone | Initial CAI | Final CAI | Zero-Shot | Single-Shot | Specialized | HCC (R2) |
|---|---|---|---|---|---|---|---|
| CLINC | BERT | 0.19 | 0.97 | 24.73 | 37.22 | 49.13 | 79.58 |
| | E5 | 0.39 | 3.73 | 35.31 | 56.78 | 77.80 | 81.91 |
| | GTE | 0.41 | 4.15 | 35.49 | 58.31 | 79.07 | 82.24 |
| | BGE | 0.47 | 4.67 | 38.96 | 60.20 | 81.40 | **85.02** |
| Massive Scenario | BERT | 0.53 | 1.82 | 44.52 | 48.79 | 59.25 | 75.76 |
| | E5 | 0.69 | 4.02 | 46.33 | 58.04 | 72.39 | 74.45 |
| | GTE | 0.73 | 4.58 | 46.07 | 58.94 | 75.79 | 75.79 |
| | BGE | 0.72 | 5.87 | 46.37 | 64.59 | 79.09 | **79.62** |
| MTop Intent | BERT | 0.26 | 0.99 | 32.15 | 45.55 | 49.20 | 65.32 |
| | E5 | 0.26 | 1.21 | 33.58 | 47.04 | 50.36 | 58.30 |
| | GTE | 0.29 | 1.26 | 37.28 | 46.58 | 49.04 | 62.47 |
| | BGE | 0.30 | 1.70 | 35.59 | 47.61 | 54.81 | **67.76** |
| StackExchange | BERT | 0.08 | 0.29 | 10.27 | 25.77 | 26.44 | 38.98 |
| | E5 | 0.14 | 0.62 | 12.95 | 32.84 | 34.96 | **47.28** |
| | GTE | 0.16 | 0.77 | 13.69 | 37.32 | 39.34 | 46.27 |
| | BGE | 0.15 | 0.66 | 13.04 | 33.93 | 35.59 | 43.67 |
| Banking77 | BERT | 0.12 | 0.51 | 18.51 | 28.02 | 31.43 | 72.34 |
| | E5 | 0.46 | 2.68 | 31.27 | 61.43 | 70.03 | 72.44 |
| | GTE | 0.51 | 2.98 | 33.96 | 63.44 | 71.82 | 76.98 |
| | BGE | 0.67 | 4.68 | 39.90 | 69.48 | 77.69 | **82.18** |
| Reddit | BERT | 0.12 | 0.52 | 19.40 | 31.43 | 29.90 | 40.85 |
| | E5 | 0.32 | 1.53 | 29.59 | 50.45 | 50.42 | 42.99 |
| | GTE | 0.36 | 1.71 | 32.95 | 51.85 | 52.19 | 43.43 |
| | BGE | 0.30 | 1.56 | 28.91 | 49.70 | 49.74 | 41.16 |
| FewRel-Nat | BERT | 0.12 | 0.77 | 12.61 | 35.18 | 41.16 | 42.92 |
| | E5 | 0.15 | 1.00 | 13.68 | 34.24 | 39.58 | **47.86** |
| | GTE | 0.11 | 0.87 | 11.47 | 32.19 | 36.07 | 45.45 |
| | BGE | 0.12 | 0.87 | 11.61 | 32.43 | 36.47 | 44.29 |
| Massive Intent | BERT | 0.22 | 0.68 | 23.91 | 32.92 | 32.99 | 61.87 |
| | E5 | 0.63 | 2.39 | 38.40 | 53.73 | 57.87 | 63.35 |
| | GTE | 0.67 | 1.43 | 39.04 | 57.83 | 60.32 | 47.51 |
| | BGE | 0.82 | 4.08 | 42.70 | 64.86 | 68.22 | **69.33** |

Table 20: **Ablation Study across Embedding Backbones and a Small Language Model (Llama3-8B Instruct).** We compare the performance of HCC using three sentence encoders (E5,GTE,BGE) and a small language model (BERT) across eight datasets. *Initial CAI* and *Final CAI* show how the verification signal strengthens as model quality improves. *HCC (R2)* consistently outperforms the Zero-Shot, Single-Shot, and Specialized baselines in most configurations, with the exception of the **Reddit** (K=10).

- These measurements **exclude** one-time preprocessing operations—namely, semantic clustering performed by the task-specific specialized model and general LLM annotation (both zero-shot and single-shot prompting)—which are lightweight and standard.
- The **Refinement Phase** constitutes the main active computational cost of HCC, representing the core alignment process where consistency-based correction and re-verification occur.

| Comparison Stage (K=3) | Correlation ($r$) | P-value ($p$) | Significance |
|---|---|---|---|
| **Baseline State** *(Initial CAI vs. Zero-Shot Acc.)* | 0.5729 | $6.11 \times 10^{-4}$ | Yes ($p < 0.001$) |
| **Prompted State** *(Initial CAI vs. Single-Shot Acc.)* | 0.6605 | $3.89 \times 10^{-5}$ | Yes ($p < 0.001$) |
| **Refined State** *(Final CAI vs. HCC Final Acc.)* | **0.7791** | $\mathbf{1.50 \times 10^{-7}}$ | **Yes ($p < 10^{-7}$)** |

Table 21: **Statistical Validation of CAI Effectiveness (N=32).** Pearson correlation tests across all 8 datasets and 4 backbones (3 Sentence Encoder and a Language Model) show that the CAI ratio depicts as reliable metric of accuracy as the co-alignment process proceeds, culminating in a highly significant correlation in the final co-alignment state.(K=3)

| Comparison Stage (K=10) | Correlation ($r$) | P-value ($p$) | Significance |
|---|---|---|---|
| **Baseline State** *(Initial CAI vs. Zero-Shot Acc.)* | 0.5731 | $6.07 \times 10^{-4}$ | Yes ($p < 0.001$) |
| **Prompted State** *(Initial CAI vs. Single-Shot Acc.)* | 0.6614 | $3.75 \times 10^{-5}$ | Yes ($p < 0.001$) |
| **Refined State** *(Final CAI vs. HCC Final Acc.)* | **0.7789** | $\mathbf{1.47 \times 10^{-7}}$ | **Yes ($p < 10^{-7}$)** |

Table 22: **Statistical Validation of CAI Effectiveness (N=32).** Pearson correlation tests across all 8 datasets and 4 backbones show that the CAI signal becomes increasingly predictive of accuracy as co-alignment progresses, culminating in a highly significant correlation at the refined state.(K=10)

| Dataset | Specialised Models | Llama 3 8B Instruct (One-Shot) | | Llama 3 8B Instruct(Zero-Shot) | |
|---|---|---|---|---|---|
| | | Tok/10k (M) | Time/10k (min) | Tok/10k (M) | Time/10k (min) |
| Banking77 | bert-base-uncased | 1.55 | 83.1 | 1.24 | 106.7 |
| | intfloat-e5-base | 1.54 | 85.4 | 1.26 | 116.8 |
| | thenlper-gte-small | 1.44 | 77.1 | 1.21 | 106.9 |
| | BAAI-bge-small-en-v1.5 | 1.52 | 83.8 | 1.22 | 106.1 |
| Clinc | bert-base-uncased | 1.42 | 75.9 | 1.74 | 211.9 |
| | intfloat-e5-base | 1.17 | 37.0 | 1.74 | 239.7 |
| | thenlper-gte-small | 1.16 | 36.0 | 1.69 | 199.6 |
| | BAAI-bge-small-en-v1.5 | 1.16 | 36.2 | 1.68 | 197.4 |
| few_rel_nat | bert-base-uncased | 1.17 | 35.9 | 1.02 | 46.4 |
| | intfloat-e5-base | 1.17 | 37.0 | 1.03 | 48.7 |
| | thenlper-gte-small | 1.16 | 36.0 | 1.03 | 47.8 |
| | BAAI-bge-small-en-v1.5 | 1.16 | 36.2 | 1.03 | 47.5 |
| massive_intent | bert-base-uncased | 0.98 | 67.7 | 0.92 | 104.6 |
| | intfloat-e5-base | 0.99 | 64.9 | 0.90 | 96.1 |
| | thenlper-gte-small | 0.99 | 63.2 | 0.93 | 102.6 |
| | BAAI-bge-small-en-v1.5 | 1.04 | 71.9 | 0.95 | 109.0 |
| massive_scenario | bert-base-uncased | 0.58 | 32.0 | 0.47 | 47.5 |
| | intfloat-e5-base | 0.58 | 34.4 | 0.48 | 48.8 |
| | thenlper-gte-small | 0.57 | 34.6 | 0.46 | 47.4 |
| | BAAI-bge-small-en-v1.5 | 0.56 | 33.5 | 0.48 | 50.4 |
| mtop_intent | bert-base-uncased | 1.30 | 63.8 | 1.32 | 121.8 |
| | intfloat-e5-base | 1.30 | 65.2 | 1.37 | 134.2 |
| | thenlper-gte-small | 1.30 | 69.5 | 1.37 | 127.9 |
| | BAAI-bge-small-en-v1.5 | 1.33 | 68.9 | 1.40 | 140.3 |
| Reddit | bert-base-uncased | 0.95 | 46.7 | 0.79 | 69.3 |
| | intfloat-e5-base | 0.94 | 44.0 | 0.77 | 67.6 |
| | thenlper-gte-small | 0.96 | 45.6 | 0.80 | 74.5 |
| | BAAI-bge-small-en-v1.5 | 0.95 | 44.9 | 0.79 | 70.5 |
| Stackexchange | bert-base-uncased | 2.13 | 87.4 | 2.49 | 312.6 |
| | intfloat-e5-base | 2.04 | 85.5 | 2.36 | 285.5 |
| | thenlper-gte-small | 2.09 | 86.7 | 2.41 | 296.1 |
| | BAAI-bge-small-en-v1.5 | 2.10 | 91.0 | 2.46 | 316.2 |

Table 23: **Marginal Estimated Cost per 10k Examples: Comparison of Teacher Models (Llama 3 8B). M denotes millions and Min denotes minutes**

**Observation:** Across nearly all datasets, **single-shot prompting is substantially faster and more token-efficient** than zero-shot prompting. Thus, HCC does not introduce **significant additional inference cost**; rather, it leverages the specialised model to perform semantic clustering and provide compact in-context examples for *group-based* annotation, followed by a lightweight refinement stage. This design keeps the overall pipeline relatively efficient while improving annotation quality. At the same time, the *zero-shot* predictions preserve output diversity by exposing the full token-distribution behaviour of the LLM, which complements the specialised model and mitigates the overconfidence commonly observed in LLM-only predictions.

| Dataset | Total Runtime All Backbones | Average Runtime per Backbone |
|---|---|---|
| Banking77 | 2 min 32.0 sec | 38.0 sec |
| CLINC | 3 min 45.3 sec | 56.3 sec |
| Reddit | 3 min 05.1 sec | 46.3 sec |
| MTOP Intent | 3 min 52.5 sec | 58.1 sec |
| Massive Scenario | 2 min 41.9 sec | 40.5 sec |
| Massive Intent | 2 min 13.7 sec | 33.4 sec |
| FewRel-Nat | 4 min 31.0 sec | 1 min 7.8 sec |
| StackExchange | 4 min 55.4 sec | 1 min 13.9 sec |
| **Average** | $\approx$ **3 min 27 sec** | $\approx$ **51.8 sec** |

Table 24: **Runtime analysis of HCC co-alignment with Llama-3-8B.** We report the total runtime required to process each dataset across all specialised backbones, together with the average runtime per backbone. The results show that HCC incurs limited computational overhead, with an average per-backbone runtime of approximately 51.8 seconds.

### J.7 MORE DETAILS ON TWO-ROUND CO-ALIGNMENT AND CAI IMPROVEMENT (LLAMA 3-8 INSTRUCT)

Table 25 and Table 26 reports the accuracy changes from Round 1 to Round 2 across all embedding backbones (E5-base, GTE-small, BGE-small-en-v1.5, and BERT-base-uncased) on the Llama3-8 Instruct Model. The results show that two rounds consistently provide the best balance between improvement and stability. Round 2 captures the remaining correctable cases, while additional rounds would propagate noise and degrade annotation quality.

**Round 1.** The initial consistent set ($\mathcal{C}$) resolves the easier or more obvious annotation errors, but many harder cases in the inconsistent set ($\mathcal{I}$) remain incorrect. This is because Round 1 relies only on $\mathcal{C} \cup H$, so its coverage is limited.

**Round 2.** The samples corrected in Round 1 expand the reference base, enabling HCC to resolve the remaining inconsistencies. This often yields further improvements (e.g., **+17.7%** on Reddit with BERT).

**Convergence and Signal Exhaustion (Why Stop at Round 2 ?).** Our results (Table 25 and Table 26) show that performance reaches a plateau by Round 2. At this point, the reference set has expanded to include all samples that can be reliably aligned ($\mathcal{C} \cup \mathcal{CI}$). The remaining inconsistent samples ($\mathcal{II}$) are mostly "difficult" cases where the LLM and the embedding model genuinely disagree and cannot be corrected further. Importantly, going beyond Round 2 risks performance regression rather than simply yielding diminishing returns.

Across several tasks, Round 2 already reaches a performance plateau. Therefore, additional rounds tend to yield diminishing returns or even reduce accuracy.For example, Banking77 (BERT) shows a drop of -1.79% and StackExchange (BERT) drops by -0.56%.

These patterns indicate that the remaining inconsistencies correspond to difficult samples, making Round 2 the appropriate and stable stopping point.

**Vote Weighting.** Our voting procedure is already similarity-aware because the top-$K$ neighbors are selected using cosine similarity, ensuring that only the most semantically similar examples contribute to the decision. Within this top-$K$ set, however, the final vote is taken using uniform weighting (simple majority). More specifically, for each inconsistent instance $x \in \mathcal{I}$, we compute its similarity with every intent group.

| Dataset (K=3) | Backbone | HCC (R1) Acc. | HCC (R2) Acc. | Δ | Observation |
|---|---|---|---|---|---|
| Banking77 | BERT | 73.41 | 71.62 | -1.79 | Saturation |
| | E5 | 75.42 | 74.58 | -0.84 | Saturation |
| | GTE | 76.85 | 77.08 | +0.23 | Convergence |
| | BGE | 81.36 | 80.68 | -0.68 | Saturation |
| CLINC | BERT | 80.00 | 82.40 | +2.40 | Refinement |
| | E5 | 81.40 | 83.11 | +1.71 | Refinement |
| | GTE | 80.44 | 81.87 | +1.43 | Refinement |
| | BGE | 84.38 | 83.98 | -0.40 | Saturation |
| Reddit | BERT | 40.41 | 58.13 | +17.72 | Correction |
| | E5 | 43.39 | 60.12 | +16.73 | Correction |
| | GTE | 43.61 | 60.62 | +17.01 | Correction |
| | BGE | 42.65 | 56.79 | +14.14 | Correction |
| MTop Intent | BERT | 0.5819 | 0.6532 | +0.0713 | Refinement |
| | E5 | 0.6192 | 0.5830 | -0.0362 | Regression |
| | GTE | 0.4466 | 0.6247 | +0.1781 | Refinement |
| | BGE | 0.6553 | 0.6776 | +0.0223 | Refinement |
| Massive Scenario | BERT | 74.82 | 74.61 | -0.21 | Saturation |
| | E5 | 74.68 | 75.66 | +0.98 | Refinement |
| | GTE | 78.14 | 77.14 | -1.00 | Saturation |
| | BGE | 80.87 | 79.49 | -1.38 | Saturation |
| Massive Intent | BERT | 61.53 | 62.61 | +1.08 | Refinement |
| | E5 | 59.78 | 63.99 | +4.21 | Refinement |
| | GTE | 62.58 | 63.25 | +0.67 | Refinement |
| | BGE | 65.47 | 66.88 | +1.41 | Refinement |
| FewRel-Nat | BERT | 42.08 | 42.05 | -0.03 | Saturation |
| | E5 | 46.56 | 47.46 | +0.90 | Refinement |
| | GTE | 42.12 | 44.31 | +2.19 | Refinement |
| | BGE | 44.73 | 44.84 | +0.11 | Convergence |
| StackExchange | BERT | 39.97 | 39.41 | -0.56 | Saturation |
| | E5 | 46.90 | 47.74 | +0.84 | Refinement |
| | GTE | 45.84 | 46.10 | +0.26 | Convergence |
| | BGE | 44.20 | 43.31 | -0.89 | Saturation |

Table 25: **Two-Round Correction Summary (Llama3-8 Instruct) Across Datasets and All Specialised Models.** Comparing **Round 1** and **Round 2** accuracy across all backbones shows that while Round 2 provides critical corrections for complex tasks (e.g., Reddit), it represents a saturation point for simpler tasks (K=3).

| Dataset (K=10) | Backbone | HCC (R1) Acc. | HCC (R2) Acc. | Δ | Observation |
|---|---|---|---|---|---|
| Banking77 | BERT | 0.7331 | 0.7234 | -0.0097 | Regression |
| | E5 | 0.7406 | 0.7244 | -0.0162 | Regression |
| | GTE | 0.7633 | 0.7698 | +0.0065 | Refinement |
| | BGE | 0.8185 | 0.8218 | +0.0033 | Refinement |
| CLINC | BERT | 0.7958 | 0.7958 | 0.0000 | Convergence |
| | E5 | 0.8258 | 0.8191 | -0.0067 | Regression |
| | GTE | 0.8162 | 0.8224 | +0.0062 | Refinement |
| | BGE | 0.8427 | 0.8502 | +0.0075 | Refinement |
| Reddit | BERT | 0.4016 | 0.4085 | +0.0069 | Refinement |
| | E5 | 0.4308 | 0.4299 | -0.0009 | Regression |
| | GTE | 0.4302 | 0.4343 | +0.0041 | Refinement |
| | BGE | 0.4224 | 0.4116 | -0.0108 | Regression |
| MTop Intent | BERT | 0.5819 | 0.6532 | +0.0713 | Refinement |
| | E5 | 0.6256 | 0.5830 | -0.0426 | Regression |
| | GTE | 0.6211 | 0.6247 | +0.0036 | Refinement |
| | BGE | 0.6623 | 0.6776 | +0.0153 | Refinement |
| Massive Scenario | BERT | 0.7515 | 0.7576 | +0.0061 | Refinement |
| | E5 | 0.7364 | 0.7445 | +0.0081 | Refinement |
| | GTE | 0.7720 | 0.7579 | -0.0141 | Regression |
| | BGE | 0.8013 | 0.7962 | -0.0051 | Regression |
| FewRel-Nat | BERT | 0.4250 | 0.4292 | +0.0042 | Refinement |
| | E5 | 0.4638 | 0.4786 | +0.0148 | Refinement |
| | GTE | 0.4580 | 0.4545 | -0.0035 | Regression |
| | BGE | 0.4453 | 0.4429 | -0.0024 | Regression |
| Massive Intent | BERT | 0.6110 | 0.6187 | +0.0077 | Refinement |
| | E5 | 0.6533 | 0.6335 | -0.0198 | Regression |
| | GTE | 0.6261 | 0.4751 | -0.1510 | Regression |
| | BGE | 0.6944 | 0.6933 | -0.0011 | Regression |
| StackExchange | BERT | 0.3891 | 0.3898 | +0.0007 | Refinement |
| | E5 | 0.4704 | 0.4728 | +0.0024 | Refinement |
| | GTE | 0.4581 | 0.4627 | +0.0046 | Refinement |
| | BGE | 0.4389 | 0.4367 | -0.0022 | Regression |

Table 26: **Two-Round Correction Summary (Llama3-8 Instruct) Across Datasets and All Specialised Models (K=10).**

# K ADDITIONAL ABLATION STUDIES BASED ON CHATGPT4O MINI LARGE LANGUAGE MODEL

## K.1 CASE ANALYSIS OF CAI ON CHATGPT 4O MINI

**Failure Case: MASSIVE-Intent.** While our co-alignment pipeline achieves consistent gains on Banking77, CLINC, MTOP, and Reddit, **MASSIVE-Intent emerges as a clear failure case**, revealing important limitations of our approach. MASSIVE-Intent is uniquely challenging due to its **dense, fine-grained intent space**, where many labels differ only by minor slot variations (e.g., inform_city, inform_city_other, inform_place_detail). As a result, semantically similar queries receive nearly identical embeddings, causing **nearest-neighbor voting to collapse** and producing unreliable correction candidates.

Furthermore, the dataset's **multilingual and translationese style** produces short, syntactically uniform queries, which substantially reduces the discriminative power of English-centric embedding models. **In addition, ChatGPT-4o-mini is primarily specialised for reasoning tasks and does not exhibit equally strong multilingual capabilities**, further limiting its ability to exploit subtle cross-lingual intent distinctions. This is reflected in its **low consistent-sample ratio ($\sim$0.40–0.45)** and a **weak CAI signal ($<$1.0)**, indicating that disagreements between the LLM and the embedder are largely uninformative. When the consistent set is small and noisy, DCCA cannot provide meaningful guidance and instead amplifies uncertainty, resulting in limited or negative correction effects.Overall, MASSIVE-Intent highlights a boundary condition of our method: when intent spaces are extremely fine-grained and embedding separability is low, consistency-driven repair becomes unreliable. This failure case indicates future extensions, such as density-adaptive clustering, slot-aware representations, and multilingual embedding backbones.

## K.2 RUNNING TIME AND TOTAL TOKEN COST (CHATGPT 4O-MINI)

We report the estimated token cost and runtime of ChatGPT-4o-mini across all tasks in Table 27. For each dataset and specialised encoder backbone, we provide the estimated token usage per 10k examples and the corresponding estimated runtime under both **single-shot** (with intent) and **zero-shot** (without intent) prompting.

| Dataset/ChatGPT 4o mini | Model | Est. Tokens/10k (Millions) | | Est. Time/10k (Minutes) | |
|---|---|---|---|---|---|
| | | Single-Shot | Zero-Shot | Single-Shot | Zero-Shot |
| Banking77 | bert-base-uncased | 1.02 | 0.92 | 16.1 | 18.2 |
| | intfloat-e5-base | 0.99 | 0.90 | 15.9 | 17.7 |
| | thenlper-gte-small | 1.01 | 0.88 | 16.2 | 17.2 |
| | BAAI-bge-small-en-v1.5 | 1.02 | 0.86 | 16.3 | 16.9 |
| Clinc | bert-base-uncased | 1.11 | 0.98 | 13.9 | 14.9 |
| | intfloat-e5-base | 1.12 | 0.97 | 14.1 | 14.3 |
| | thenlper-gte-small | 1.13 | 0.98 | 14.0 | 14.5 |
| | BAAI-bge-small-en-v1.5 | 1.11 | 0.98 | 14.3 | 15.0 |
| few_rel_nat | bert-base-uncased | 1.15 | 1.06 | 14.7 | 15.7 |
| | intfloat-e5-base | 1.13 | 1.07 | 14.5 | 15.8 |
| | thenlper-gte-small | 1.13 | 1.10 | 14.7 | 16.1 |
| | BAAI-bge-small-en-v1.5 | 1.15 | 1.12 | 14.3 | 16.2 |
| massive_intent | bert-base-uncased | 0.78 | 0.75 | 14.7 | 19.1 |
| | intfloat-e5-base | 0.77 | 0.71 | 15.0 | 17.7 |
| | thenlper-gte-small | 0.78 | 0.70 | 14.9 | 17.2 |
| | BAAI-bge-small-en-v1.5 | 0.77 | 0.72 | 14.9 | 17.3 |
| massive_scenario | bert-base-uncased | 0.52 | 0.37 | 12.6 | 13.6 |
| | intfloat-e5-base | 0.51 | 0.38 | 12.5 | 12.6 |
| | thenlper-gte-small | 0.52 | 0.37 | 12.7 | 13.1 |
| | BAAI-bge-small-en-v1.5 | 0.52 | 0.38 | 12.4 | 12.8 |
| mtop_intent | bert-base-uncased | 1.07 | 1.63 | 15.7 | 30.3 |
| | intfloat-e5-base | 1.06 | 1.62 | 16.3 | 29.0 |
| | thenlper-gte-small | 1.05 | 1.69 | 16.1 | 30.8 |
| | BAAI-bge-small-en-v1.5 | 1.05 | 1.71 | 15.7 | 31.0 |
| Reddit | bert-base-uncased | 0.83 | 0.67 | 15.0 | 15.6 |
| | intfloat-e5-base | 0.82 | 0.66 | 14.9 | 15.0 |
| | thenlper-gte-small | 0.83 | 0.67 | 14.9 | 15.6 |
| | BAAI-bge-small-en-v1.5 | 0.83 | 0.67 | 14.9 | 15.2 |
| Stackexchange | bert-base-uncased | 1.61 | 1.45 | 19.8 | 21.2 |
| | intfloat-e5-base | 1.61 | 1.43 | 19.7 | 21.1 |
| | thenlper-gte-small | 1.61 | 1.43 | 19.9 | 20.2 |
| | BAAI-bge-small-en-v1.5 | 1.61 | 1.44 | 20.1 | 20.3 |

Table 27: Marginal Estimated Cost per 10k Examples (GPT-4o-mini): Single-Shot (With Intent) vs Zero-Shot (Without Intent)

### K.3 RUNNING TIME OF HCC FOR CO-ALIGNMENT (CHATGPT 4O-MINI)

To demonstrate the computational efficiency of Heterogeneous Consistency Co-alignment (HCC), we report the actual wall-clock execution time for the complete two-round verification procedure on all datasets in Table 28. The results show that HCC completes co-alignment within **0.36–1.11 minutes per dataset**, highlighting its extremely low computational overhead.

| Dataset / ChatGPT-4o-mini | Specialised Model | Time (s) | Time (min) |
|---|---|---|---|
| Clinc | bert-base-uncased | 33.46 | 0.56 |
| | intfloat-e5-base | 37.76 | 0.63 |
| | thenlper-gte-small | 35.63 | 0.59 |
| | BAAI-bge-small-en-v1.5 | 35.42 | 0.59 |
| massive_scenario | bert-base-uncased | 21.79 | 0.36 |
| | intfloat-e5-base | 28.95 | 0.48 |
| | thenlper-gte-small | 26.01 | 0.43 |
| | BAAI-bge-small-en-v1.5 | 26.44 | 0.44 |
| Stackexchange | bert-base-uncased | 47.42 | 0.79 |
| | intfloat-e5-base | 63.09 | 1.05 |
| | thenlper-gte-small | 60.33 | 1.01 |
| | BAAI-bge-small-en-v1.5 | 59.43 | 0.99 |
| Banking77 | bert-base-uncased | 26.25 | 0.44 |
| | intfloat-e5-base | 27.75 | 0.46 |
| | thenlper-gte-small | 26.54 | 0.44 |
| | BAAI-bge-small-en-v1.5 | 24.31 | 0.41 |
| massive_intent | bert-base-uncased | 24.72 | 0.41 |
| | intfloat-e5-base | 27.49 | 0.46 |
| | thenlper-gte-small | 24.92 | 0.42 |
| | BAAI-bge-small-en-v1.5 | 22.18 | 0.37 |
| few_rel_nat | bert-base-uncased | 45.91 | 0.77 |
| | intfloat-e5-base | 66.86 | 1.11 |
| | thenlper-gte-small | 63.53 | 1.06 |
| | BAAI-bge-small-en-v1.5 | 63.22 | 1.05 |
| Reddit | bert-base-uncased | 35.48 | 0.59 |
| | intfloat-e5-base | 48.39 | 0.81 |
| | thenlper-gte-small | 43.38 | 0.72 |
| | BAAI-bge-small-en-v1.5 | 42.57 | 0.71 |
| mtop_intent | bert-base-uncased | 31.22 | 0.52 |
| | intfloat-e5-base | 47.50 | 0.79 |
| | thenlper-gte-small | 44.53 | 0.74 |
| | BAAI-bge-small-en-v1.5 | 41.39 | 0.69 |

Table 28: Actual Execution Time per Dataset for HCC Co-Alignment (ChatGPT 4o mini).

### K.4 TWO-ROUND CO-ALIGNMENT AND CAI IMPROVEMENT (CHATGPT 4O-MINI

) Table 30 summarises the two-round HCC correction results for `ChatGPT-4o-mini` across all datasets. We report (1) the initial and final CAI ratios, (2) Round-1 and Round-2 correction accuracies, and (3) the accuracy gain from R1 to R2. These results explicitly show how HCC improves annotation alignment.

| Dataset | Backbone | Initial CAI | Final CAI | LLM (Zero-Shot) | LLM (Single-Shot) | Specialized | HCC (R2) |
|---------|----------|-------------|-----------|-----------------|-------------------|-------------|----------|
| **CLINC** | BERT | 0.70 | 1.34 | 76.82 | 56.89 | 55.40 | 85.91 |
| | E5 | 1.42 | 5.23 | 77.33 | 71.87 | 80.71 | 87.67 |
| | GTE | 1.34 | 5.51 | 77.16 | 70.93 | 81.22 | **88.04** |
| | BGE | 1.59 | 6.51 | 78.38 | 72.64 | 83.13 | 88.02 |
| **Massive Scenario** | BERT | 0.73 | 1.51 | 64.29 | 55.78 | 58.44 | 77.91 |
| | E5 | 1.02 | 3.32 | 63.21 | 63.52 | 69.91 | 76.06 |
| | GTE | 1.20 | 4.63 | 64.36 | 67.72 | 73.60 | 78.21 |
| | BGE | 1.10 | 3.27 | 61.87 | 69.87 | **76.40** | 73.67 |
| **MTOP Intent** | BERT | 0.74 | 1.30 | 71.82 | 54.77 | 51.39 | 74.76 |
| | E5 | 0.68 | 1.29 | 70.75 | 55.77 | 51.48 | 67.74 |
| | GTE | 0.65 | 1.59 | 71.32 | 54.65 | 50.98 | 71.45 |
| | BGE | 0.82 | 2.20 | 71.77 | 58.50 | 56.57 | **74.94** |
| **StackExchange** | BERT | 0.22 | 0.42 | 35.90 | 39.03 | 38.67 | 44.18 |
| | E5 | 0.41 | 0.87 | 41.34 | 41.03 | 38.67 | **51.15** |
| | GTE | 0.40 | 0.89 | 41.46 | 42.11 | 38.98 | 51.79 |
| | BGE | 0.38 | 0.83 | 40.59 | 41.24 | 37.13 | 49.15 |
| **Banking77** | BERT | 0.47 | 0.78 | 58.44 | 47.76 | 40.13 | 79.09 |
| | E5 | 1.41 | 3.74 | 62.60 | 71.66 | 71.56 | 80.19 |
| | GTE | 1.51 | 5.02 | 62.89 | 75.29 | 75.62 | **82.14** |
| | BGE | 1.82 | 5.23 | 64.68 | 79.42 | 79.48 | 81.59 |
| **Reddit** | BERT | 0.19 | 0.54 | 45.63 | 38.42 | 32.05 | 59.56 |
| | E5 | 0.44 | 1.47 | 48.80 | 52.44 | 51.63 | 61.36 |
| | GTE | 0.50 | 1.66 | 49.95 | 53.93 | 53.34 | **63.38** |
| | BGE | 0.47 | 1.58 | 50.20 | 51.76 | 51.51 | 60.24 |
| **FewRel-Nat** | BERT | 0.33 | 0.93 | 35.40 | 41.29 | 42.50 | 45.67 |
| | E5 | 0.33 | 1.03 | 34.84 | 40.49 | 41.23 | **49.01** |
| | GTE | 0.31 | 1.01 | 36.29 | 38.13 | 38.37 | 47.68 |
| | BGE | 0.30 | 0.98 | 35.58 | 36.99 | 37.68 | 45.96 |
| **Massive Intent** | BERT | 0.69 | 0.95 | 72.76 | 48.45 | 42.80 | 70.14 |
| | E5 | 1.58 | 2.86 | 74.01 | 62.51 | 61.26 | 69.77 |
| | GTE | 2.86 | 2.98 | 74.54 | 64.22 | 63.21 | 68.99 |
| | BGE | 2.13 | 4.52 | **75.45** | 69.10 | 68.83 | 75.22 |

Table 29: **Aggregated Backbone Comparison across 8 datasets.** Initial CAI = Final CAI = model's Consistent–Inconsistent Ratio Count. Zero-Shot = without showing intent, Single-Shot = with showing intent, Specialized = correction rate, HCC (R2) = two-round majority voting accuracy (GPT-4o mini).

| Dataset | Backbone | Initial CAI | Final CAI | HCC (R1) Acc. | HCC (R2) Acc. | $\Delta$ Gain | Observation |
|---------|----------|-------------|-----------|---------------|---------------|---------------|-------------|
| **CLINC** | BERT | 0.70 | 1.34 | 85.93 | 85.91 | -0.02 | Convergence |
| | E5 | 1.42 | 5.23 | 87.33 | 87.67 | +0.34 | Convergence |
| | GTE | 1.34 | 5.51 | 86.87 | 87.04 | +0.17 | Convergence |
| | BGE | 1.59 | 6.51 | 89.24 | 88.02 | -1.22 | Convergence |
| **Massive Scenario** | BERT | 0.73 | 1.51 | 78.41 | 77.91 | -0.50 | Saturation |
| | E5 | 1.02 | 3.32 | 72.73 | 76.06 | +3.33 | Convergence |
| | GTE | 1.20 | 4.63 | 77.71 | 78.21 | +0.50 | Saturation |
| | BGE | 1.10 | 3.27 | 71.12 | 73.67 | +2.55 | Convergence |
| **MTop Intent** | BERT | 0.74 | 1.30 | 73.26 | 74.76 | +1.50 | Refinement |
| | E5 | 0.68 | 1.29 | 67.83 | 67.74 | -0.09 | Saturation |
| | GTE | 0.65 | 1.59 | 69.81 | 71.45 | +1.64 | Refinement |
| | BGE | 0.82 | 2.20 | 73.00 | 74.94 | +1.94 | Refinement |
| **StackExchange** | BERT | 0.22 | 0.42 | 43.53 | 44.18 | +0.65 | Convergence |
| | E5 | 0.87 | 0.87 | 50.94 | 51.15 | +0.21 | Convergence |
| | GTE | 0.40 | 0.89 | 51.32 | 51.49 | +0.17 | Saturation |
| | BGE | 0.38 | 0.83 | 49.83 | 49.16 | -0.67 | Convergence |
| **Banking77** | BERT | 0.47 | 0.78 | 75.97 | 79.09 | +3.12 | Refinement |
| | E5 | 1.41 | 3.74 | 71.66 | 80.19 | +8.53 | Correction |
| | GTE | 1.51 | 5.02 | 75.29 | 82.14 | +6.85 | Refinement |
| | BGE | 1.82 | 5.23 | 81.62 | 81.59 | -0.03 | Convergence |
| **FewRel-Nat** | BERT | 0.33 | 0.93 | 26.90 | 45.67 | +18.77 | Correction |
| | E5 | 0.33 | 1.03 | 50.29 | 49.00 | -1.29 | Convergence |
| | GTE | 0.31 | 1.01 | 47.88 | 47.70 | -0.18 | Saturation |
| | BGE | 0.30 | 0.98 | 46.96 | 45.96 | -1.00 | Convergence |
| **Reddit** | BERT | 0.19 | 0.54 | 42.40 | 59.56 | +17.16 | Correction |
| | E5 | 0.44 | 1.47 | 44.82 | 61.36 | +16.54 | Correction |
| | GTE | 0.50 | 1.66 | 45.79 | 63.38 | +17.59 | Correction |
| | BGE | 0.47 | 1.58 | 43.33 | 60.24 | +16.91 | Correction |
| **Massive Intent** | BERT | 0.69 | 0.95 | 70.40 | 70.14 | -0.26 | Saturation |
| | E5 | 1.58 | 2.86 | 69.83 | 69.77 | -0.06 | Convergence |
| | GTE | 1.62 | 2.98 | 69.24 | 69.00 | -0.24 | Convergence |
| | BGE | 2.13 | 4.52 | 74.45 | 75.22 | +0.77 | Refinement |

Table 30: **Two-Round Correction / GPT-4o mini & CAI Score Summary.**

# L ADDITIONAL ABLATION STUDIES BASED ON CHATGPT3.5 TURBO LARGE LANGUAGE MODEL

## L.1 FAILURE CASE OF CAI ON CHATGPT 3.5 TURBO

**No Failure Case:** Our experiments show that CAI is extremely effective on ChatGPT-3.5 Turbo compared to other LLMs such as GPT-4o-Mini and open-source LLM Llama3-8 Instruct. CAI im-

provements correlate reliably with downstream gains, showing that ChatGPT-3.5 aligns well with embedding-driven cluster structure. **Even in extremely fine-grained settings such as MASSIVE-Intent, where cluster separability is weak, CAI correctly identifies that only marginal improvements are possible. The small CAI gain (0.69→0.95) aligns proportionally with the small accuracy gain from Zero-Shot to HCC(R2) (+0.47%), demonstrating that CAI remains a reliable indicator even under challenging semantic conditions.**

## L.2 TOTAL TOKEN USAGE AND LLM RUNNING TIME (CHATGPT-3.5 TURBO)

We report the total token usage and LLM (ChatGPT-3.5 Turbo) running time across all tasks in Table 31. For each dataset and specialised encoder backbone, we provide the estimated token consumption per 10k examples and the corresponding estimated runtime for **single-shot** (with intent) and **zero-shot** (without intent) prompting. These results show that HCC does not introduce significant additional token usage or latency beyond the baseline prompting scheme.

| Dataset/ChatGPT 3.5 Turbo | Model | Est. Tokens/10k (Millions) | | Est. Time/10k (Minutes) | |
|---|---|---|---|---|---|
| | | Single-Shot | Zero-Shot | Single-Shot | Zero-Shot |
| Banking77 | bert-base-uncased | 1.02 | 0.92 | 16.1 | 18.2 |
| | intfloat-e5-base | 0.99 | 0.90 | 15.9 | 17.7 |
| | thenlper-gte-small | 1.01 | 0.88 | 16.2 | 17.2 |
| | BAAI-bge-small-en-v1.5 | 1.02 | 0.86 | 16.3 | 16.9 |
| Clinc | bert-base-uncased | 1.11 | 0.98 | 13.9 | 14.9 |
| | intfloat-e5-base | 1.12 | 0.97 | 14.1 | 14.3 |
| | thenlper-gte-small | 1.13 | 0.98 | 14.0 | 14.5 |
| | BAAI-bge-small-en-v1.5 | 1.11 | 0.98 | 14.3 | 15.0 |
| few_rel_nat | bert-base-uncased | 1.15 | 1.06 | 14.7 | 15.7 |
| | intfloat-e5-base | 1.14 | 1.07 | 14.5 | 15.8 |
| | thenlper-gte-small | 1.13 | 1.10 | 14.7 | 16.1 |
| | BAAI-bge-small-en-v1.5 | 1.15 | 1.12 | 14.3 | 16.2 |
| massive_intent | bert-base-uncased | 0.78 | 0.75 | 14.7 | 19.1 |
| | intfloat-e5-base | 0.78 | 0.71 | 15.0 | 17.7 |
| | thenlper-gte-small | 0.77 | 0.70 | 14.9 | 17.2 |
| | BAAI-bge-small-en-v1.5 | 0.77 | 0.72 | 14.9 | 17.3 |
| massive_scenario | bert-base-uncased | 0.52 | 0.37 | 12.6 | 13.6 |
| | intfloat-e5-base | 0.52 | 0.37 | 12.5 | 12.6 |
| | thenlper-gte-small | 0.52 | 0.37 | 12.7 | 13.1 |
| | BAAI-bge-small-en-v1.5 | 0.52 | 0.37 | 12.4 | 12.8 |
| mtop_intent | bert-base-uncased | 1.07 | 1.63 | 15.7 | 30.3 |
| | intfloat-e5-base | 1.06 | 1.62 | 16.3 | 29.0 |
| | thenlper-gte-small | 1.05 | 1.69 | 16.1 | 30.8 |
| | BAAI-bge-small-en-v1.5 | 1.04 | 1.71 | 15.7 | 31.0 |
| Reddit | bert-base-uncased | 0.83 | 0.67 | 15.0 | 15.6 |
| | intfloat-e5-base | 0.83 | 0.67 | 14.9 | 15.0 |
| | thenlper-gte-small | 0.83 | 0.67 | 14.9 | 15.6 |
| | BAAI-bge-small-en-v1.5 | 0.83 | 0.67 | 14.9 | 15.2 |
| Stackexchange | bert-base-uncased | 1.60 | 1.45 | 19.8 | 21.2 |
| | intfloat-e5-base | 1.59 | 1.43 | 19.7 | 21.1 |
| | thenlper-gte-small | 1.60 | 1.44 | 19.9 | 20.2 |
| | BAAI-bge-small-en-v1.5 | 1.59 | 1.43 | 20.1 | 20.3 |

Table 31: Marginal estimated cost per 10k examples (ChatGPT-3.5 Turbo): single-shot (with intent) vs. zero-shot (without intent).

### L.3 RUNNING TIME OF HCC FOR CO-ALIGNMENT (CHATGPT-3.5 TURBO)

We further report the wall-clock running time of HCC for co-alignment on all tasks in Table 32. For each dataset and embedding model, we list the per-dataset execution time in seconds and minutes. These measurements confirm that HCC operates within **0.36–1.15 minutes per dataset**, demonstrating that the verification and correction procedure is highly efficient in practice.

| Dataset | Embedding Model | Time (s) | Time (min) |
|---|---|---|---|
| **Clinc** | bert-base-uncased | 32.53 | 0.54 |
| | intfloat-e5-base | 33.55 | 0.56 |
| | thenlper-gte-small | 32.38 | 0.54 |
| | BAAI-bge-small-en-v1.5 | 32.06 | 0.53 |
| **Massive Scenario** | bert-base-uncased | 22.32 | 0.37 |
| | intfloat-e5-base | 29.06 | 0.48 |
| | thenlper-gte-small | 24.82 | 0.41 |
| | BAAI-bge-small-en-v1.5 | 24.98 | 0.42 |
| **StackExchange** | bert-base-uncased | 47.76 | 0.80 |
| | intfloat-e5-base | 58.51 | 0.98 |
| | thenlper-gte-small | 53.74 | 0.90 |
| | BAAI-bge-small-en-v1.5 | 55.59 | 0.93 |
| **Banking77** | bert-base-uncased | 25.88 | 0.43 |
| | intfloat-e5-base | 24.55 | 0.41 |
| | thenlper-gte-small | 21.65 | 0.36 |
| | BAAI-bge-small-en-v1.5 | 21.61 | 0.36 |
| **Massive Intent** | bert-base-uncased | 23.93 | 0.40 |
| | intfloat-e5-base | 25.66 | 0.43 |
| | thenlper-gte-small | 25.12 | 0.42 |
| | BAAI-bge-small-en-v1.5 | 22.20 | 0.37 |
| **FewRel-Nat** | bert-base-uncased | 46.35 | 0.77 |
| | intfloat-e5-base | 67.00 | 1.12 |
| | thenlper-gte-small | 65.12 | 1.09 |
| | BAAI-bge-small-en-v1.5 | 68.96 | 1.15 |
| **Reddit** | bert-base-uncased | 35.28 | 0.59 |
| | intfloat-e5-base | 44.34 | 0.74 |
| | thenlper-gte-small | 38.55 | 0.64 |
| | BAAI-bge-small-en-v1.5 | 39.20 | 0.65 |
| **Mtop Intent** | bert-base-uncased | 33.70 | 0.56 |
| | intfloat-e5-base | 51.86 | 0.86 |
| | thenlper-gte-small | 48.40 | 0.81 |
| | BAAI-bge-small-en-v1.5 | 45.46 | 0.76 |

Table 32: **Execution Time per Dataset for HCC Co-alignment (ChatGPT-3.5 Turbo).**

### L.4 CO-ALIGNMENT AND CAI IMPROVEMENT (CHATGPT-3.5 TURBO)

Finally, Table L.4 reports the two-round HCC correction results for `ChatGPT-3.5 Turbo` across all datasets. We list the initial and final CAI ratios, the Round-1 (R1) and Round-2 (R2) accuracies, and the resulting accuracy change $\Delta = R2 - R1$. This comparison highlights where HCC leads to refinement, convergence, saturation, or regression when applied on top of ChatGPT-3.5 Turbo.

| Dataset | Backbone | Initial CAI | Final CAI | Zero-Shot | Single-Shot | Specialized | HCC (R2) Acc. |
|---|---|---|---|---|---|---|---|
| **CLINC** | BERT | 0.89 | 1.49 | 67.78 | 56.22 | 55.40 | 82.76 |
| | E5 | 1.93 | 6.29 | 68.89 | 80.04 | 80.71 | 86.80 |
| | GTE | 1.84 | 6.50 | 67.71 | 80.49 | 81.22 | 85.84 |
| | BGE | 1.94 | 6.51 | 68.80 | 82.00 | 83.13 | 87.67 |
| **Massive Scenario** | BERT | 0.88 | 1.62 | 58.81 | 60.52 | 58.44 | 74.98 |
| | E5 | 1.31 | 3.21 | 59.65 | 67.25 | 69.91 | 70.28 |
| | GTE | 1.60 | 4.53 | 60.86 | 71.32 | 73.60 | 73.71 |
| | BGE | 1.56 | 5.73 | 60.66 | 73.97 | 76.40 | 77.30 |
| **MTOP Intent** | BERT | 0.59 | 1.28 | 56.79 | 53.67 | 51.39 | 71.71 |
| | E5 | 0.59 | 1.28 | 57.84 | 52.49 | 51.48 | 71.71 |
| | GTE | 0.55 | 1.70 | 58.96 | 53.90 | 50.98 | 68.40 |
| | BGE | 0.67 | 2.11 | 60.37 | 57.73 | 56.57 | 69.81 |
| **StackExchange** | BERT | 0.39 | 0.58 | 31.47 | 27.45 | 27.41 | 41.58 |
| | E5 | 0.74 | 1.25 | 36.33 | 38.57 | 38.67 | 48.44 |
| | GTE | 0.74 | 1.26 | 35.68 | 38.76 | 38.98 | 47.16 |
| | BGE | 0.61 | 1.11 | 34.53 | 37.05 | 37.13 | 45.43 |
| **Banking77** | BERT | 0.58 | 0.91 | 51.46 | 42.08 | 40.13 | 74.81 |
| | E5 | 2.03 | 4.88 | 62.31 | 71.56 | 71.56 | 77.44 |
| | GTE | 2.59 | 5.15 | 64.81 | 75.49 | 75.62 | 73.96 |
| | BGE | 2.62 | 4.08 | 66.72 | 79.41 | 79.48 | 72.63 |
| **Reddit** | BERT | 0.24 | 0.59 | 42.56 | 34.19 | 32.05 | 58.38 |
| | E5 | 0.70 | 1.88 | 51.48 | 51.91 | 51.63 | 60.58 |
| | GTE | 0.84 | 2.24 | 52.75 | 53.93 | 53.34 | 60.71 |
| | BGE | 0.82 | 2.19 | 52.10 | 51.79 | 51.51 | 58.19 |
| **FewRel-Nat** | BERT | 0.33 | 0.95 | 31.14 | 43.46 | 42.50 | 44.91 |
| | E5 | 0.36 | 1.10 | 31.00 | 40.49 | 41.23 | 48.59 |
| | GTE | 0.32 | 1.01 | 31.65 | 38.77 | 38.37 | 47.32 |
| | BGE | 0.29 | 0.98 | 30.76 | 38.68 | 37.68 | 45.85 |
| **Massive Intent** | BERT | 0.69 | 0.95 | 67.22 | 48.15 | 42.80 | 67.69 |
| | E5 | 1.89 | 3.25 | 71.08 | 63.25 | 61.26 | 68.86 |
| | GTE | 1.71 | 3.21 | 68.59 | 64.93 | 63.21 | 68.66 |
| | BGE | 2.43 | 4.90 | 69.94 | 69.33 | 68.83 | 72.73 |

Table 33: **Aggregated Backbone Comparison across 8 datasets.** Initial CAI and Final CAI represent Consistent–Inconsistent ratios. Zero-Shot = accuracy without showing intent, Single-Shot = accuracy with showing intent, Specialized = correction rate, HCC (R2) = two-round majority vote accuracy.

| Dataset | Backbone | Initial CAI | Final CAI | HCC(R1) Acc. | HCC(R2) Acc. | Δ Gain | Observation |
|---|---|---|---|---|---|---|---|
| Clinc | BERT | 0.89 | 1.49 | 82.33 | 82.76 | +0.43 | Convergence |
| | E5 | 1.93 | 6.29 | 86.69 | 86.80 | +0.11 | Convergence |
| | GTE | 1.84 | 6.50 | 85.64 | 85.84 | +0.20 | Convergence |
| | BGE | 1.94 | 6.51 | 87.89 | 87.67 | -0.22 | Saturation |
| Massive Scenario | BERT | 0.88 | 1.62 | 71.15 | 74.98 | +3.83 | Correction |
| | E5 | 1.31 | 3.21 | 68.22 | 70.28 | +2.06 | Correction |
| | GTE | 1.60 | 4.53 | 75.55 | 73.71 | -1.84 | Saturation |
| | BGE | 1.56 | 5.73 | 80.36 | 77.30 | -3.06 | Saturation |
| MTOP Intent | BERT | 0.59 | 1.28 | 70.73 | 71.71 | +0.98 | Refinement |
| | E5 | 0.59 | 1.28 | 71.48 | 71.71 | +0.23 | Minor Gain |
| | GTE | 0.55 | 1.70 | 65.32 | 68.40 | +3.08 | Correction |
| | BGE | 0.67 | 2.11 | 69.40 | 69.81 | +0.41 | Refinement |
| StackExchange | BERT | 0.39 | 0.58 | 42.16 | 41.58 | -0.58 | Saturation |
| | E5 | 0.74 | 1.25 | 48.32 | 48.44 | +0.12 | Convergence |
| | GTE | 0.74 | 1.26 | 47.88 | 47.16 | -0.72 | Saturation |
| | BGE | 0.61 | 1.11 | 45.28 | 45.43 | +0.15 | Convergence |
| Banking77 | BERT | 0.58 | 0.91 | 72.40 | 74.81 | +2.41 | Refinement |
| | E5 | 2.03 | 4.88 | 75.88 | 77.44 | +1.56 | Refinement |
| | GTE | 2.59 | 5.15 | 72.05 | 73.96 | +1.91 | Refinement |
| | BGE | 2.62 | 4.08 | 82.11 | 72.63 | -9.48 | Overshoot |
| Reddit | BERT | 0.24 | 0.59 | 41.19 | 58.38 | +17.19 | Strong Correction |
| | E5 | 0.70 | 1.88 | 43.36 | 60.58 | +17.22 | Strong Correction |
| | GTE | 0.84 | 2.24 | 43.99 | 60.71 | +16.72 | Strong Correction |
| | BGE | 0.82 | 2.19 | 42.24 | 58.19 | +15.95 | Strong Correction |
| FewRel-Nat | BERT | 0.33 | 0.95 | 45.51 | 44.91 | -0.60 | Saturation |
| | E5 | 0.36 | 1.10 | 49.96 | 48.59 | -1.37 | Saturation |
| | GTE | 0.32 | 1.01 | 47.30 | 47.32 | +0.02 | Convergence |
| | BGE | 0.29 | 0.98 | 46.41 | 45.85 | -0.56 | Saturation |
| Massive Intent | BERT | 0.72 | 1.02 | 66.64 | 67.69 | +1.05 | Correction |
| | E5 | 1.89 | 3.25 | 69.97 | 68.86 | -1.11 | Saturation |
| | GTE | 1.71 | 3.21 | 68.90 | 68.66 | -0.24 | Saturation |
| | BGE | 2.43 | 4.90 | 68.80 | 72.73 | +3.93 | Correction |

Table 34: **HCC Correction Summary: Initial vs Final CAI, R1/R2 Accuracy (in %), and Gain.**

# M  ADDITIONAL ABLATION STUDIES ON LLAMA-3-7B-INSTRUCT WITH 1%, 5%, AND 10% USER PREFERENCE

We further investigate the robustness of HCC under different levels of user-preference supervision, namely 1%, 5%, and 10%, across multiple random seeds using `Llama-3-7B-Instruct`. To broaden the evaluation, we additionally include two challenging datasets, `FEW_NERD_NAT` and `GO_EMOTION`.

Table 35: **Llama 3-8B Instruct**: Accuracy comparison across datasets with 1%, 5%, and 10% user preference samples. Results are reported as mean $\pm$ standard deviation (%).

| Dataset | Prop. | LLM Zero-Shot | Specialised + LLM | Specialised Model | HCC (Ours) |
|---|---|---|---|---|---|
| CLINC | 1% | $29.26 \pm 1.04$ | $54.68 \pm 1.24$ | $64.85 \pm 0.13$ | $\mathbf{71.18 \pm 0.13}$ |
| | 5% | $31.51 \pm 1.82$ | $65.08 \pm 1.72$ | $79.31 \pm 0.47$ | $\mathbf{81.36 \pm 0.60}$ |
| | 10% | $36.87 \pm 1.00$ | $70.80 \pm 0.02$ | $84.38 \pm 0.40$ | $\mathbf{86.84 \pm 0.06}$ |
| MASSIVE-Scenario | 1% | $43.99 \pm 0.37$ | $55.56 \pm 1.06$ | $60.42 \pm 0.67$ | $\mathbf{66.21 \pm 0.24}$ |
| | 5% | $46.48 \pm 1.14$ | $67.33 \pm 0.52$ | $72.83 \pm 0.27$ | $\mathbf{74.45 \pm 0.74}$ |
| | 10% | $46.92 \pm 0.50$ | $71.33 \pm 0.63$ | $78.53 \pm 0.22$ | $\mathbf{79.22 \pm 0.90}$ |
| MTOP-Intent | 1% | $28.10 \pm 0.22$ | $\mathbf{45.30 \pm 0.57}$ | $41.14 \pm 0.90$ | $44.95 \pm 1.79$ |
| | 5% | $31.22 \pm 0.74$ | $56.68 \pm 0.87$ | $53.44 \pm 0.50$ | $\mathbf{61.87 \pm 0.70}$ |
| | 10% | $32.08 \pm 0.77$ | $59.32 \pm 1.56$ | $56.48 \pm 0.25$ | $\mathbf{61.67 \pm 4.87}$ |
| StackExchange | 1% | $11.39 \pm 0.36$ | $20.96 \pm 0.17$ | $21.15 \pm 0.14$ | $\mathbf{29.36 \pm 0.51}$ |
| | 5% | $12.15 \pm 0.25$ | $30.48 \pm 0.43$ | $31.61 \pm 0.20$ | $\mathbf{38.85 \pm 0.66}$ |
| | 10% | $14.26 \pm 1.09$ | $34.16 \pm 0.23$ | $36.92 \pm 0.59$ | $\mathbf{44.74 \pm 0.05}$ |
| BANKING77 | 1% | $29.23 \pm 1.09$ | $50.18 \pm 3.39$ | $53.73 \pm 3.71$ | $\mathbf{60.28 \pm 1.49}$ |
| | 5% | $38.07 \pm 1.09$ | $69.79 \pm 1.57$ | $73.71 \pm 1.86$ | $\mathbf{78.08 \pm 1.23}$ |
| | 10% | $39.75 \pm 0.38$ | $76.38 \pm 0.02$ | $81.50 \pm 0.16$ | $\mathbf{84.38 \pm 0.60}$ |
| GoEmotion | 1% | $\mathbf{19.61 \pm 0.40}$ | $12.52 \pm 0.68$ | $12.50 \pm 0.55$ | $18.80 \pm 0.92$ |
| | 5% | $20.15 \pm 0.20$ | $14.34 \pm 0.58$ | $14.31 \pm 0.67$ | $\mathbf{21.11 \pm 0.02}$ |
| | 10% | $19.68 \pm 0.10$ | $18.05 \pm 0.29$ | $17.90 \pm 0.27$ | $\mathbf{26.10 \pm 1.15}$ |
| FewRel-Natural | 1% | $11.59 \pm 0.22$ | $23.68 \pm 1.07$ | $25.52 \pm 0.73$ | $\mathbf{34.58 \pm 0.92}$ |
| | 5% | $12.81 \pm 0.64$ | $31.68 \pm 0.02$ | $34.02 \pm 0.34$ | $\mathbf{42.75 \pm 0.09}$ |
| | 10% | $13.20 \pm 0.24$ | $33.84 \pm 0.18$ | $36.68 \pm 0.78$ | $\mathbf{45.61 \pm 1.37}$ |
| Reddit | 1% | $22.70 \pm 0.47$ | $37.17 \pm 0.02$ | $37.88 \pm 0.27$ | $\mathbf{51.06 \pm 0.51}$ |
| | 5% | $28.31 \pm 0.94$ | $49.27 \pm 0.12$ | $50.99 \pm 0.26$ | $\mathbf{58.80 \pm 0.33}$ |
| | 10% | $31.84 \pm 0.43$ | $52.08 \pm 0.20$ | $54.18 \pm 0.41$ | $\mathbf{60.79 \pm 0.02}$ |
| FewNERD-Natural | 1% | $\mathbf{36.05 \pm 0.10}$ | $25.48 \pm 1.07$ | $20.76 \pm 0.81$ | $29.53 \pm 0.19$ |
| | 5% | $\mathbf{36.41 \pm 0.07}$ | $29.66 \pm 0.24$ | $27.62 \pm 0.01$ | $31.31 \pm 0.12$ |
| | 10% | $\mathbf{35.82 \pm 1.65}$ | $31.68 \pm 0.93$ | $29.89 \pm 0.91$ | $29.97 \pm 0.95$ |
| MASSIVE-Intent | 1% | $29.66 \pm 1.98$ | $44.57 \pm 0.05$ | $46.42 \pm 0.59$ | $\mathbf{54.37 \pm 0.33}$ |
| | 5% | $35.13 \pm 1.37$ | $55.83 \pm 0.76$ | $\mathbf{61.72 \pm 1.06}$ | $60.15 \pm 5.35$ |
| | 10% | $35.86 \pm 1.23$ | $58.56 \pm 0.92$ | $65.03 \pm 0.77$ | $\mathbf{66.85 \pm 0.67}$ |

Our procedure first uses the specialised model to produce preliminary annotations, which serve as the basis for constructing semantic clusters. We then apply group prompting, where each LLM query contains 10 instances selected from the same predicted annotation cluster. When the specialised model produces higher-quality annotations, the resulting clusters become more semantically coherent, allowing the LLM to process groups of closely related samples rather than heterogeneous instances.

This cluster-based prompting strategy improves both response consistency and annotation accuracy. In particular, we find that the top 10% of LLM outputs, ranked by alignment confidence, achieve substantially higher accuracy than the bottom 1%. This observation indicates that the quality of annotation alignment is a strong indicator of downstream LLM reliability.

The full results are reported in Table 35. Overall, HCC consistently improves over the standalone LLM and the specialised model on most datasets and supervision levels. As the available user-preference supervision increases from 1% to 10%, HCC generally exhibits steady performance improvements, showing that it can effectively exploit limited supervision to enhance annotation reliability. In addition, the relatively small standard deviations across random seeds suggest that HCC remains stable under different initialisations.

Table 36: **Llama 3-8B Instruct**: Accuracy comparison across datasets under imbalanced user preference samples with 1%, 5%, and 10% label budgets. Results are reported as mean $\pm$ standard deviation (%).

| Dataset | Budget | LLM Zero-Shot | Specialised + LLM | Specialised Model | HCC (Ours) |
|---|---|---|---|---|---|
| CLINC | 1% | 27.28 ± 0.06 | 54.39 ± 0.06 | 62.86 ± 0.94 | **63.36 ± 1.91** |
| | 5% | 34.01 ± 2.43 | 68.28 ± 1.34 | 78.98 ± 0.24 | **82.11 ± 0.18** |
| | 10% | 36.54 ± 1.46 | 71.79 ± 1.48 | 83.98 ± 1.49 | **85.61 ± 0.72** |
| MASSIVE-Scenario | 1% | 45.15 ± 0.49 | 57.30 ± 2.10 | 60.73 ± 2.76 | **65.59 ± 1.03** |
| | 5% | 46.22 ± 0.29 | 66.80 ± 1.02 | 75.94 ± 1.16 | **77.67 ± 1.04** |
| | 10% | 46.36 ± 0.86 | 70.10 ± 0.57 | **78.50 ± 0.08** | 75.03 ± 2.94 |
| MTOP-Intent | 1% | 27.82 ± 1.31 | 44.93 ± 1.90 | 39.47 ± 3.83 | **54.00 ± 0.63** |
| | 5% | 30.11 ± 0.50 | 51.64 ± 1.85 | 49.66 ± 1.09 | **60.80 ± 2.50** |
| | 10% | 32.03 ± 0.57 | 57.37 ± 0.83 | 55.97 ± 2.30 | **58.40 ± 3.59** |
| StackExchange | 1% | 11.49 ± 0.70 | 21.11 ± 0.96 | 21.03 ± 1.49 | **29.38 ± 0.36** |
| | 5% | 12.27 ± 0.37 | 29.86 ± 0.48 | 31.44 ± 0.06 | **38.29 ± 0.42** |
| | 10% | 13.16 ± 0.28 | 33.20 ± 0.67 | 35.73 ± 0.38 | **42.82 ± 0.42** |
| BANKING77 | 1% | 31.49 ± 0.19 | 58.72 ± 0.83 | 62.03 ± 0.41 | **63.99 ± 1.30** |
| | 5% | 37.08 ± 0.06 | 69.71 ± 1.43 | 74.17 ± 0.76 | **74.32 ± 5.29** |
| | 10% | 40.39 ± 1.07 | 77.55 ± 1.25 | 81.38 ± 1.02 | **84.63 ± 0.44** |
| Reddit | 1% | 22.58 ± 0.22 | 38.25 ± 0.89 | 38.67 ± 1.03 | **50.44 ± 0.20** |
| | 5% | 28.88 ± 0.12 | 49.19 ± 0.84 | 50.23 ± 0.81 | **58.70 ± 0.30** |
| | 10% | 29.49 ± 1.60 | 52.34 ± 0.36 | 53.48 ± 0.39 | **61.11 ± 0.16** |
| FewRel-Natural | 1% | 11.45 ± 0.27 | 23.36 ± 0.12 | 24.70 ± 0.23 | **31.67 ± 0.49** |
| | 5% | 12.46 ± 0.25 | 31.46 ± 1.22 | 33.69 ± 0.84 | **42.40 ± 0.28** |
| | 10% | 12.91 ± 0.46 | 33.34 ± 0.35 | 36.10 ± 0.23 | **42.25 ± 2.17** |
| MASSIVE-Intent | 1% | 31.61 ± 1.01 | 45.76 ± 0.39 | **49.34 ± 1.43** | 48.08 ± 8.41 |
| | 5% | 34.56 ± 0.44 | 54.14 ± 0.29 | 58.15 ± 0.22 | **61.40 ± 0.44** |
| | 10% | 35.92 ± 0.08 | 59.18 ± 0.49 | 63.23 ± 0.18 | **63.32 ± 5.14** |

# N   OVERALL REFLECTION OF CAI

**Interpretation.** A low CAI value indicates that the *Consistent* subset (LLM–encoder agreement) remains small relative to the *Inconsistent* subset. Importantly, this does *not* mean that HCC has failed; rather, CAI is correctly signalling **low semantic confidence due to weak embedding separation**. In such cases, the encoder struggles to form meaningful clusters, and the consistency signal becomes naturally noisier.

**Implications for Practice.** To enhance the robustness of CAI under low-CAI regimes, we recommend:

(i) increasing the proportion of human-in-the-loop or user-preference annotations to stabilise cluster boundaries, and

(ii) adopting a stronger encoder backbone (e.g., E5 or BGE) to improve representation discriminability.

**Role as a Diagnostic Signal.** Crucially, even when CAI is small (e.g., on StackExchange), HCC can still produce measurable accuracy gains. In such settings, CAI serves its intended diagnostic role: it **signals that improvements are likely to be moderate and that downstream predictions**

**should be interpreted conservatively**. This behaviour is consistent with the broader empirical findings in Section 5.2 and Table 19, where smaller CAI deltas systematically correspond to weaker post-alignment improvements. For clarity, we summarise practical CAI operating regimes in Table 37.

Table 37: **CAI Regime Guide (Empirically Grounded Interpretation).** CAI measures cross-model agreement between embedding-level structural signals and LLM token-level predictions. The regimes provide diagnostic interpretation based on observed experimental behaviour.

| CAI Range | Before HCC (Diagnostic Meaning) | After HCC (Observed Behaviour) |
|---|---|---|
| CAI $< 0.5$ | Very weak cross-model agreement. Embedding structure provides minimal support for LLM predictions. Accuracy is typically unstable or low. | If CAI remains in this regime, improvements are limited. Structural reinforcement is weak and gains are inconsistent. |
| CAI $0.5 \sim 1.0$ | Weak-to-moderate agreement. Some structural consistency exists, but disagreement still dominates. | Moderate CAI increases may improve stability, but accuracy gains are dataset-dependent. |
| CAI $1.0 \sim 2.0$ | Balanced regime: agreement begins to exceed disagreement. Indicates meaningful structural support. | When CAI moves into or above this regime, accuracy improvements are frequently observed, especially for closed-source LLMs. |
| CAI $2.0 \sim 4.0$ | Strong agreement. Embedding similarity and token-level predictions are mutually reinforcing. | Typically associated with stable and substantial accuracy gains across most datasets in our experiments. |
| CAI $> 4.0$ | High-consistency regime. Strong structural alignment between models. | Corresponds to the largest observed performance gains (e.g., `Clinc`, `Banking77`, `Massive Scenario`). |

## O  CROSS-DOMAIN GENERALIZATION.

Our evaluation already spans **four distinct domain families**, demonstrating that HCC generalizes across radically different linguistic styles, noise levels, and task structures **without any modification to the method**. The domains included are:

- **Technical and Specialized**:
  - **Banking77** — fine-grained financial queries (HCC (R2): 71–80%)
  - **StackExchange** — programming and troubleshooting (HCC (R2): 39–47%)
- **Social Media (Noisy, Informal)**:
  - **Reddit** — subjective, noisy user discussions (HCC (R2): 58–60%)
- **Fact-Based / Structured**:
  - **FewRel-Nat** — relation classification with structured logical semantics (HCC (R2): 42–47%)
- **General Assistant and Everyday Queries**:
  - **CLINC, Massive Intent, Massive Scenario, MTOP** — broad user-assistant intent spaces (HCC (R2): 62–83%)

Across all domains, HCC achieves moderate improvement over baselines. These consistent gains demonstrate **strong cross-domain generalization**.

## P  CAN HCC HANDLE CONFLICTING USER PREFERENCES?

**Conflicting Preferences.** HCC naturally accommodates conflicting user preferences because the propagation step is *conditioned on each user's seed examples* ($H$). Our method assumes that each

Table 38: **Model selection using the CAI Ratio.** We compare the model selected by the highest CAI Ratio against the model with the highest annotation accuracy. The accuracy gap is computed as the CAI-selected model accuracy minus the best accuracy.

| Dataset | CAI-Selected Model | | Best-Accuracy Model | | Match | Gap (%) |
|---|---|---|---|---|---|---|
| | Model | Acc. (%) | Model | Acc. (%) | | |
| CLINC | Gemini 1.5 Flash | 87.24 | Gemini 1.5 Flash | 87.24 | ✓ | 0.00 |
| MTOP-Intent | Gemini 1.5 Flash | 75.85 | Gemini 1.5 Flash | 75.85 | ✓ | 0.00 |
| StackExchange | Gemini 1.5 Flash | 57.31 | Gemini 1.5 Flash | 57.31 | ✓ | 0.00 |
| BANKING77 | Gemini 1.5 Flash | 73.76 | GPT-3.5 Turbo | 73.93 | ✗ | -0.17 |
| MASSIVE-Scenario | Gemini 1.5 Flash | 67.72 | GPT-3.5 Turbo | 75.55 | ✗ | -7.83 |
| Reddit | Gemini 1.5 Flash | 56.23 | GPT-4o Mini | 57.39 | ✗ | -1.16 |
| GoEmotion | Gemini 1.5 Flash | 29.44 | GPT-4o Mini | 33.82 | ✗ | -4.38 |
| FewRel-Natural | Gemini 1.5 Flash | 52.74 | Gemini 1.5 Flash | 52.74 | ✓ | 0.00 |
| FewNERD-Natural | Gemini 1.5 Flash | 75.48 | Gemini 1.5 Flash | 75.48 | ✓ | 0.00 |
| MASSIVE-Intent | Gemini 1.5 Flash | 77.03 | Gemini 1.5 Flash | 77.03 | ✓ | 0.00 |
| **Match Rate** | **6 / 10** | | | | **60%** | – |

user provides a *small labeled set per class*, which captures that user's individual labeling preferences. These seed examples act as the **personalized anchors** that guide consistency propagation and alignment. Thus, HCC generates **per-user, preference-aligned annotations** for large unlabeled corpora, rather than learning a global reward model shared across users.

The goal of HCC is therefore **not general RLHF-style preference modeling**, but **Personalized Natural Language Understanding**, reflecting each user's subjective preferences to unlabeled data in a reliable, scalable, and training-free manner.

**Pairwise Data.** Our framework addresses *categorical annotation* rather than *pairwise preference ranking*. While CAI could, in principle, be extended to pairwise comparisons for RLHF-style data, this falls outside the scope of our cold-start classification setting. Nevertheless, we view this as a promising direction for future work.

## Q    MODEL SELECTION STUDIES

**Model Selection with CAI Ratio.**    We further examine whether the CAI Ratio can be used as a reference-free criterion for model selection. Specifically, we compare the model selected by the highest CAI Ratio with the model that achieves the highest annotation accuracy. The candidate LLM set includes GPT-3.5 Turbo, GPT-4o Mini, Gemini 1.5 Flash, and Llama-3-8B Instruct.

As shown in Table 38, CAI-based selection correctly identifies the best-performing model in 6 out of 10 datasets. In the remaining cases, the selected model still achieves competitive accuracy, with relatively small gaps on Banking77 and Reddit, but larger gaps on MASSIVE-Scenario and GoEmotion. These results suggest that although the CAI Ratio is not a perfect proxy for accuracy, it provides a useful reference-free heuristic for selecting strong LLMs in semi-supervised annotation settings.

# R    ALGORITHM TABLE

We present the algorithmic procedures for Divide-and-Conquer Co-Alignment (DCCA) with MV-VTES (Algorithm 1) and Majority Voting via the Top-Nearest Embedding Scheme (MV-VTES) (Algorithm 2).

---

**Algorithm 1:** Divide-and-Conquer Co-Alignment (DCCA)

---

**Input:** Consistent set $\mathcal{C} = \{(x_i, \bar{y}_i^c)\}_{i=1}^{|\mathcal{C}|}$,

Inconsistent set $\mathcal{I} = \{(x_i, \bar{y}_i^I)\}_{i=1}^{|\mathcal{I}|}$,
User-preference set $H$,
Number of neighbours $k$,
Embedding function $\mathcal{S}(\cdot)$ (used in MV-VTES).
**Output:** Self-corrected dataset $D^{(\text{final})}$.

1   **Round 1: Co-aligning $\mathcal{I}$ to obtain $\mathcal{I}^{(1)}$**

2   $\mathcal{I}^{(1)} \leftarrow \emptyset$

3   **foreach** $(x, \bar{y}^I) \in \mathcal{I}$ **do**

4      $\hat{y} \leftarrow \text{MV-VTES}(x, \mathcal{C} \cup H, k, \mathcal{S})$        // Eq. equation 3, equation 4

5      $\mathcal{I}^{(1)} \leftarrow \mathcal{I}^{(1)} \cup \{(x, \hat{y})\}$

6   **CAI-based partition of $\mathcal{I}$**

7   $\mathcal{CI} \leftarrow \emptyset$,

8   $\mathcal{II} \leftarrow \emptyset$

9   **foreach** $(x, \bar{y}^I) \in \mathcal{I}$ **do**

10     Retrieve $(x, \hat{y}) \in \mathcal{I}^{(1)}$

11     **if** $\hat{y} = \bar{y}^I$ **then**

12      $\mathcal{CI} \leftarrow \mathcal{CI} \cup \{(x, \hat{y})\}$        // Becomes consistent

13     **else**

14      $\mathcal{II} \leftarrow \mathcal{II} \cup \{(x, \bar{y}^I)\}$        // Remains inconsistent

15   **Round 2: Co-aligning $\mathcal{II}$ to obtain $\mathcal{II}^{(1)}$**

16   $\mathcal{II}^{(1)} \leftarrow \emptyset$

17   **foreach** $(x, \bar{y}^I) \in \mathcal{II}$ **do**

18     $\tilde{y} \leftarrow \text{MV-VTES}(x, \mathcal{C} \cup \mathcal{CI} \cup H, k, \mathcal{S})$

19     $\mathcal{II}^{(1)} \leftarrow \mathcal{II}^{(1)} \cup \{(x, \tilde{y})\}$

20   **Final dataset**

21   $D^{(\text{final})} \leftarrow \mathcal{C} \cup \mathcal{CI} \cup \mathcal{II}^{(1)}$

22   **return** $D^{(\text{final})}$

---

---

**Algorithm 2:** Majority Voting via Top-Nearest Embedding Scheme (MV-VTES)

---

**Input:** Query sample $x$,
Reference set $\mathcal{D}_e = \mathcal{C} \cup H \subset \mathcal{X} \times \mathcal{Y}$ where $(a, \bar{y}) \in \mathcal{D}_e$,
Number of neighbours $k$,
Embedding function $\mathcal{S}(\cdot)$.
**Output:** Refined annotation $\hat{y}$ for $x$.

1   $\mathcal{A}_e \leftarrow \{a \in \mathcal{X} \mid (a, \bar{y}) \in \mathcal{D}_e\}$
2   **foreach** $(a, \bar{y}) \in \mathcal{D}_e$ **do**
3     $\left| \quad s(a, x) \leftarrow \dfrac{\mathcal{S}(a) \cdot \mathcal{S}(x)}{\|\mathcal{S}(a)\| \, \|\mathcal{S}(x)\|} \right.$          `// Cosine similarity`
4   Select $\{a_i\}_{i=1}^k = \text{TopK}_{a \in \mathcal{A}_e} \, s(a, x)$ as in Eq. equation 3
5   Let $\bar{y}_i$ be the label associated with $a_i$ in $\mathcal{D}_e$
6   Let $A \leftarrow \{\bar{y}_1, \ldots, \bar{y}_k\}$ be the set of unique retrieved labels
7   **foreach** $a \in A$ **do**
8     $\left| \quad n_a \leftarrow \sum_{i=1}^k \mathbb{I}[\bar{y}_i = a] \right.$          `// Label frequency`
9   $\hat{y} \leftarrow \arg\max_{a \in A} n_a$          `// Eq. equation 4`
10
11   **return** $\hat{y}$

---

# S  PROMPT INSTRUCTION

## S.1  PROMPT INSTRUCTION (WITHOUT INTENT)

Listing 1: Prompt construction for batched intent labelling

```python
def Prompt(prompts, specialised_model_labels, intention_set,
    temperature):
    """
    Build the LLM prompt for batched intent labelling.
    prompts: list of input sentences
    specialised_model_labels: labels from the specialised model
    intention_set: candidate intent set
    temperature: decoding temperature for the LLM
    """

    # 1. Build sentence-prefixed inputs
    combination = []
    for prompt, label in zip(prompts, specialised_model_labels):
        prompt1 = f'For the sentence: "{prompt}"'
        combination.append(prompt1)

    # 2. Assemble instruction block
    respon = ""
    respon += f"Please return exactly {len(prompts)} responses.\n"
    respon += (
        f"Identify the intention for each sentence "
        f"using the set: {intention_set}.\n"
    )
    respon += "Follow the specified output format strictly.\n"
    respon += "Make sure the number of outputs matches the number of
        inputs.\n"

    # 3. Query the LLM
    output = openai.ChatCompletion.create(
        model="gpt-4o-mini",
        messages=[{"role": "user", "content": respon}],
        temperature=temperature,
        max_tokens=2048,
    )

    return output
```

## S.2 PROMPT INSTRUCTION (WITH INTENT)

Listing 2: Prompt construction for batched intent labelling

```python
def Prompt(prompts, specialised_model_labels, intention_set,
        temperature):
    """
    Build the LLM prompt for batched intent labelling.
    prompts: list of input sentences
    specialised_model_labels: labels from the specialised model
    intention_set: candidate intent set
    temperature: decoding temperature for the LLM
    """

    # 1. Build sentence-prefixed inputs
    combination = []
    for prompt, label in zip(prompts, specialised_model_labels):
        prompt1 = f'For the sentence: "{prompt}" and its predicted
                intent: "{specialised_model_labels}". Use this predicted
                intent as guidance.'
        combination.append(prompt1)

    # 2. Assemble instruction block
    respon = ""
    respon += f"Please return exactly {len(prompts)} responses.\n"
    respon += (
        f"Identify the intention for each sentence "
        f"using the set: {intention_set}.\n"
    )
    respon += "Follow the specified output format strictly.\n"
    respon += "Make sure the number of outputs matches the number of
        inputs.\n"

    # 3. Query the LLM
    output = openai.ChatCompletion.create(
        model="gpt-4o-mini",
        messages=[{"role": "user", "content": respon}],
        temperature=temperature,
        max_tokens=2048,
    )

    return output
```

# T   T-SNE VISUALIZATION FOR CLUSTERING ON ALL DATASETS FOR CHATGPT-4O MINI

In the following t-SNE visualization, the LLM-generated annotations of the identified inconsistent samples exhibit substantial deviation from their true annotations. Conversely, for the identified consistent samples, the LLM-generated annotations show strong agreement with the corresponding ground-truth labels.

# U   USAGE OF LARGE LANGUAGE MODEL (LLM)

ChatGPT (OpenAI) was used solely for language editing to improve clarity and grammar. It did not contribute to the scientific content, analysis, or conclusions of this work.

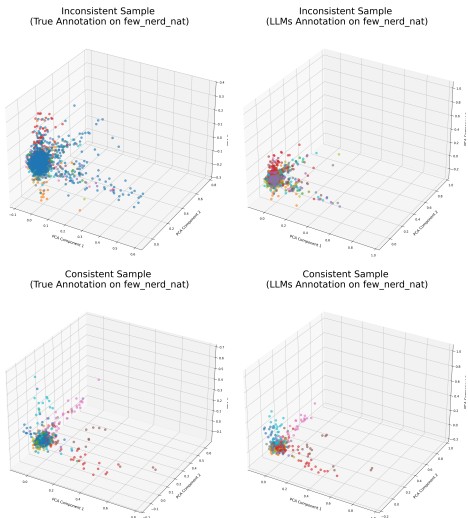

Figure 9: **Visualization of t-SNE Clustering** for LLM vs True Annotations on *Few_Nerd_Nat* Dataset. LLM outputs exhibit *high similarity* with ground-truth labels on **consistent** samples, while showing *significant divergence* on **inconsistent** samples.

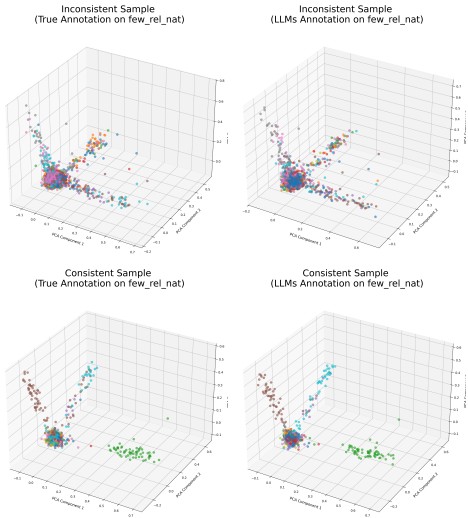

Figure 10: **Visualization of t-SNE Clustering** for LLM vs True Annotations on *Few_Rel_Nat* Dataset. LLM outputs exhibit *high similarity* with ground-truth labels on **consistent** samples, while showing *significant divergence* on **inconsistent** samples.

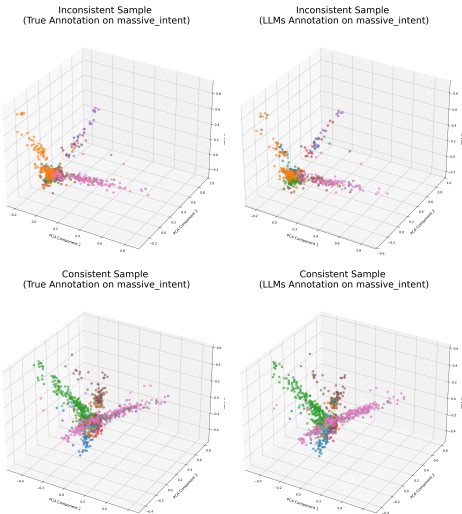

Figure 11: **Visualization of t-SNE Clustering** for LLM vs True Annotations on *Massive_Intent* Dataset. LLM outputs exhibit ***high similarity*** with ground-truth labels on **consistent** samples, while showing ***significant divergence*** on **inconsistent** samples.

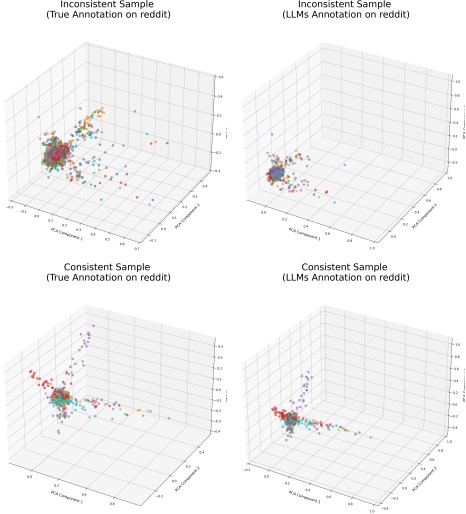

Figure 12: **Visualization of t-SNE Clustering** for LLM vs True Annotations on *Reddit* Dataset. LLM outputs exhibit ***high similarity*** with ground-truth labels on **consistent** samples, while showing ***significant divergence*** on **inconsistent** samples.

