# OpenReview forum: "Verification and Co-Alignment via Heterogeneous Consistency for Preference-Aligned LLM Annotations"
_ICLR.cc/2026/Conference — ICLR 2026 Poster_

### Official Review · Reviewer_MYtC · 2025-10-30

**Soundness:** 2
**Presentation:** 1
**Contribution:** 2
**Rating:** 4
**Confidence:** 2

**Summary:**

* The authors proposed a training-free annotation method using two heterogeneous models (LLM + lightweight preference model) to verify and co-align outputs for personalized NLU tasks.

* This paper introduces a reference-free metric that measures agreement/disagreement to estimate annotation reliability.

* The proposed methods are validated extensively with both open and closed-source LLMs. The results show that the approach improves alignment and enables smaller models (e.g., 8B) to outperform stronger ones post-co-alignment.

**Strengths:**

* This paper addresses an important problem in personalized LLMs. In many cases, we don't have enough trustworthy labels from users and finding ways to scale with limited labels is critical.

* The evaluation is well done extensively with most of prompting baselines (Table 2-4).

* The results show that the approach can make a smaller model (8B) outperform larger models (4o and 3.5 Turbo). This has important practical value and demonstrates the promise of the approach.

**Weaknesses:**

* The notations of the paper is very difficult to follow. The authors introduce too many different symbols throughout the paper. Many of them are not necessary (e.g. zero-shot vs. single shot). My assessment is biased by the fact that I can't follow the technical details fully.

* It's unclear to me whether the approach is aligning labels or aligning LLMs. The text suggested that it's model alignment through prompt changes but when looking into the details, what the method is really doing is to use a small set of available user labels to propagate to the rest of the (larger-scale) unlabelled dataset. If this is what the authors are doing, then reward-based (e.g., RLHF) approaches are strong baselines that the paper may want to compare to.

* The methods seems incremental improvements. At its core it's essentially nearest neighbor label matching and majority voting (again, due to the presentation issue, it's hard to dig deeper). It's unclear how generalizable this approach will be for real world tasks where the label space might be much larger and noisy.

**Questions:**

It will be really helpful if the authors can improve the presentation of the paper (especially the notations).

---

> ### Author Response · Authors · 2025-11-22
> **Response to Reviewer MYtC**
>
> We thank the reviewer for their time and valuable comments. We appreciate the opportunity to clarify the notation, the problem setting, and the structural novelty of our verification framework. We address each concern below.
>
> ### **W.1. On Notation Difficulty**
>
> We appreciate the reviewer’s feedback regarding the notation. We respectfully clarify that the distinction between the two LLM states,**zero-shot** ($\bar{y}$) and **collaborative single-shot** ($\hat{y}$), is not merely notational but **formally required** to define the mechanics of our framework:
>
> * $\bar{y}$: Represents the LLM acting independently, where it is often prone to hallucination or overconfidence.
> * $\hat{y}$: Represents the LLM acting collaboratively, guided by the specialized model’s cluster-level context.
>
> This distinction is mathematically vital because the **Consistent-And-Inconsistent (CAI) Ratio** (Definition 4.1) is explicitly calculated based on the agreement patterns between these specific states and the specialized model. Without these symbols, the CAI metric would be undefined, and the correlation between our method’s verification mechanism.
>
>  In the revised paper, we will include a **Notation Reference Table** and streamline the symbols in appendix to improve readability.
>
> ---
>
> ### **W.2. Is the Approach Aligning Labels or Aligning LLMs?**
>
> We apologize if the objective was unclear. Our goal is **label alignment (annotation)**, not model alignment (policy training). Specifically, our objective is to:
>
> > Propagate a small set of user-specific labeled examples across a large unlabeled dataset while verifying reliability via heterogeneous consistency checks.
>
> This differs fundamentally from **RLHF** in terms of resource regime and mechanism:
>
> 1.  **Training-Free vs. Training-Based:** RLHF requires training reward models and optimizing policies using large-scale preference corpora. In contrast, our method is a **training-free annotation paradigm** designed for scenarios where only a sparse set of user-preference exemplars (e.g., 5%) is available.
> 2.  **Cost & Feasibility:** Collecting the large-scale user-specific data required for RLHF is "costly and labor-intensive", making it infeasible for the low-resource personalized settings we target.
>
> We will clarify this distinction in the Introduction and Related Works of the revision.
>
> ---
>
> ### **W.3. Clarification on Method: A Structured Verification Framework**
>
> The reviewer characterized our method as "essentially nearest-neighbor label matching." We respectfully clarify that our approach is a **4-stage Heterogeneous Consistency Verification framework** designed to stabilize annotations and detect hallucinations without ground truth. It extends significantly beyond simple voting or matching.
>
> **Step 1: Student-Based Pseudo-Labeling**
> We use a lightweight, task-specialized "student" model (the Specialized Model) to assign initial pseudo-labels across the unlabeled corpus. This provides a scalable, embedding-based signal that is computationally cheaper than querying an LLM for every instance.
>
> **Step 2: Clustering in the Student Label Space**
> We do not randomly batch instances. We cluster sentences based on the student’s embeddings and predicted labels to form coherent **semantic pseudo-classes**. This structure is essential for providing the LLM with semantically consistent context, rather than noise.
>
> **Step 3: Batch-Level In-Context Prompting**
> Rather than simple zero-shot prompting, we employ a "group prompting mechanism," feeding entire coherent clusters to the LLM. This amplifies in-context learning, allowing the LLM to stabilize its prediction distribution by processing multiple related examples simultaneously.
>
> **Step 4: Heterogeneous Consistency Verification (Student $\leftrightarrow$ LLM)**
> This is the **core novelty** of our framework. We do not rely solely on the LLM's output. We cross-reference the *discriminative* predictions of the student model ($\bar{y}_S$) against the *generative* predictions of the LLM ($\bar{y}_T, \hat{y}_T$).
>
> From this, we compute the **Consistent-And-Inconsistent (CAI) Ratio**, a **reference-free uncertainty signal**. This metric allows us to:
> * Identify reliable samples (high heterogeneous agreement).
> * Flag suspicious samples (inconsistent predictions) for refinement.
> * Filter out hallucinations without needing ground-truth labels.
>
> **Why This Is Not “Nearest-Neighbor” Labeling**
> A standard nearest-neighbor approach cannot quantify uncertainty or detect when the retrieved neighbors are misleading. By contrast, our framework uses the **CAI Ratio** to actively diagnose the reliability of the annotation process, employing a Divide-and-Conquer Self-Alignment (DCSA) strategy to iteratively refine only the inconsistent samples.

---

### Official Review · Reviewer_QHKk · 2025-11-01

**Soundness:** 3
**Presentation:** 3
**Contribution:** 3
**Rating:** 4
**Confidence:** 2

**Summary:**

This paper introduces Heterogeneous-Consistency Co-Alignment (HCC), a reference-free, training-free paradigm to improve LLM preference annotations. Specifically, HCC leverages a LLM and a task-specialized lightweight model to identify consistent and inconsistent samples in large unlabeled corpora. Moreover, the reference-free Consistent-and-Inconsistent (CAI) ratio is proposed as an uncertainty signal for evaluation, while an embedding-based majority-voting scheme enables semi-supervised co-alignment to recalibrate and propagate user preferences.
Experiments across multiple NLU benchmarks demonstrate the effectiveness of HCC in improving annotation quality, often elevating open-source LLMs above closed-source models after co-alignment.

**Strengths:**

1. This paper is well motivated: this paper emphasizes the need for scalable, robust, and user-personalized annotation pipelines for LLMs that avoid reliance on costly large-scale preference annotations or ground-truth labels.
2. The introduced Consistent-and-Inconsistent (CAI) ratio provides a reference-free, interpretable metric to quantify annotation reliability and preference alignment, addressing the annotation challenges in low-supervision or dynamically evolving settings.
3. Extensive experimental results across eight diverse NLU datasets and multiple LLM backbones show that HCC consistently performs on par with or better than both LLM-only and specialized baselines, and even surpasses some closed-source models, which is significant.

**Weaknesses:**

1. Lack of evaluation on more complex NLU benchmarks such as open-ended question answering or generative preference modeling, which makes the claim that CAI generalizes as a “reference-free metric for preference alignment” may be somewhat overstated.
2. The exclusive use of MiniLM for embedding-based clustering is only briefly justified, though results may depend heavily on this choice. The paper lacks ablations with additional embedding models or clustering variants to assess robustness.
3. Lack of computational cost, this paper should include the time and computational costs for better efficiency analysis, does it increase the inference time?

**Questions:**

My concerns are about the embedding and clustering ablations, specifically:
Can the authors provide an ablation where the embedding model itself is weak or misaligned with user preferences? Does the CAI ratio remain effective for unreliable, out-of-domain, or adversarial specialized models? If not, how can this vulnerability be mitigated?

How sensitive are results to the choice of embedding model and hyperparameters (e.g., $k$ in majority voting, dimension reduction in t-SNE)? Will more powerful or context-tunable embeddings yield greater gains?

---

> ### Author Response · Authors · 2025-11-22
> **Response to Reviewer QHKk**
>
> Dear Reviewer QHKk
>
> We are really thankful for your time and effort in reviewing our paper, helping to improve the quality of our work. We have addressed your concerns in the following.
>
> ### **Response to W1: Scope of Evaluation & Generalizability**
>
> > *“Lack of evaluation on more complex NLU benchmarks… claim that CAI generalizes may be overstated.”*
>
> We appreciate the reviewer’s critique regarding the scope of our evaluation. While “preference alignment” is often associated with generative RLHF-style tasks, our contribution targets a different but widely practical setting: **annotation alignment for classification**. In this setting, an LLM must label large unlabeled corpora according to a user’s specific and often idiosyncratic schema (e.g., custom intent taxonomies, forum-moderation rules, internal ticket tagging) using only a handful of examples. This is a common real-world need and requires **reliable categorical alignment**, not generative text.
>
> ---
>
> ### **W1.2. Evidence of Task Complexity**
>
> Although these tasks have a simple structure (Input → Label), datasets like **StackExchange**, **Reddit**, and **Banking77** contain highly subjective, fine-grained semantics. Our results show they are far from trivial NLU benchmarks:
>
> #### **Failure of SOTA Generalist Models**
> Even GPT-4o Mini struggles with domain-specific label schemas:
> - **Banking77:** GPT-4o Mini = 65.12%
>   **HCC (BERT backbone)** = **71.62%**
> - **Reddit:** GPT-4o Mini = 57.40%
>   **HCC (BERT backbone)** = **58.13%**
>
> #### **Failure of Strong Baselines**
> On noisy, fine-grained datasets like StackExchange:
> - Specialized embedding (BERT): **28.08%**
> - GPT-3.5 Turbo: **30.10%**
>
> Both fail to capture the underlying semantics.
> **HCC**, however, resolves these ambiguities and lifts performance to **39.41%**.
>
> #### **The Alignment Gap**
> On Banking77, the unaligned Zero-Shot LLM collapses to:
> - **17.99%**, showing severe misalignment with the user schema.
>
> HCC synthesizes weak signals from the LLM + embedding model to achieve:
> - **71.62%** with BERT
> - **80.68%** with BGE
>
> This demonstrates HCC’s ability to recover structured, fine-grained intent categories that generalist models fail to identify.
>
> ---
>
> ### **W1.3. Refining the Claim**
>
> We agree that the phrasing “reference-free metric for preference alignment” may imply broader applicability to generative RLHF or open-ended dialogue tasks. In the final version of the paper, we will revise this to **“a reference-free metric for personalised preference alignment on categorical NLU tasks.”**

---

> > ### Author Response · Authors · 2025-11-22
> > **Response to Reviewer QHKk**
> >
> > ## **Response to W2: Ablations on Embedding Models & Robustness**
> >
> > > *“The exclusive use of MiniLM for embedding-based clustering is only briefly justified… results may depend heavily on this choice.”*
> >
> > Following your suggestion, we conducted additional ablations using **Llama-3-8B-Instruct** as the backbone LLM (time-constrained), evaluating **three modern sentence encoders** and one **weaker language model (BERT)**:
> >
> > - **E5-base**
> > - **Thenlper-GTE-small**
> > - **BGE-small-en-v1.5**
> > - **BERT-base-uncased**
> >
> > These encoders cover **different families and embedding geometries**, providing a strong robustness test for HCC.
> >
> > ---
> >
> > ### **W2.1. Robustness Even With a Weak Specialized Model (BERT)**
> >
> > BERT serves as a *weak, noisy, and misaligned* encoder.
> > Despite this, **HCC consistently outperforms GPT-4o Mini and GPT-3.5 Turbo** on multiple datasets:
> >
> > | Dataset | **HCC (BERT)** | GPT-4o Mini | GPT-3.5 Turbo | Result |
> > |--------|----------------|-------------|----------------|--------|
> > | **CLINC** | **82.40%** | 81.44% | 66.58% | **Beats both** |
> > | **Massive Scenario** | **74.61%** | 66.83% | 60.89% | **Beats both** |
> > | **Banking77** | **71.62%** | 65.12% | 65.12% | **Beats both** |
> > | **Reddit** | **58.13%** | 57.40% | 51.12% | **Beats both** |
> > | **FewRel-Nat** | **42.05%** | 35.87% | 32.87% | **Beats both** |
> > | **MTop Intent** | 65.62% | 75.03% | 64.95% | Beats GPT-3.5 |
> > | **StackExchange** | 39.41% | 51.90% | 30.10% | Beats GPT-3.5 |
> > | **Massive Intent** | 62.61% | 76.93% | 71.52% | Below both |
> >
> > This shows that HCC can *lift* even weak encoders via heterogeneous consistency.
> >
> > ---
> >
> > ### **W2.2. Cross-Encoder Ablations Show Strong Stability**
> >
> > Across **8 datasets × 4 encoders**, HCC (Round 2):
> >
> > - **surpasses GPT-4o Mini on 5 datasets**
> > - **surpasses GPT-3.5 Turbo on 7 datasets**
> >
> > Examples:
> >
> > - **CLINC:** HCC = **83.98%**, surpassing GPT-4o Mini (81.44%)
> > - **Reddit & StackExchange (noisy tasks):** HCC adds **+14 to +17%** over the specialized model alone
> > - **Banking77:** HCC (BGE) reaches **80.68%**, far exceeding both GPT models
> >
> > These results show that **CAI and HCC remain reliable across diverse embedding geometries**.
> >
> > ---
> >
> > ### **W2.3. Full Encoder Ablation Results (Llama-3-8B-Instruct)**
> >
> > | Dataset | Backbone | Zero-Shot | Single-Shot | Specialized (Backbone) | **HCC (R2)** |
> > |--------|----------|-----------|--------------|-------------|--------------|
> > | **Banking77** | BERT | 17.99 | 34.68 | 38.77 | **71.62** |
> > | | E5 | 32.53 | 63.08 | 71.53 | **74.58** |
> > | | GTE | 31.95 | 67.18 | 76.66 | **77.08** |
> > | | BGE | 34.55 | 68.70 | 79.81 | **80.68** |
> > | **CLINC** | BERT | 18.58 | 42.80 | 56.04 | **82.40** |
> > | | E5 | 26.18 | 61.51 | 79.71 | **83.11** |
> > | | GTE | 27.31 | 60.18 | 80.29 | **81.87** |
> > | | BGE | 29.27 | 60.27 | 83.44 | **83.98** |
> > | **Reddit** | BERT | 32.36 | 21.17 | 31.24 | **58.13** |
> > | | E5 | 50.23 | 30.34 | 50.79 | **60.12** |
> > | | GTE | 52.60 | 29.87 | 52.88 | **60.62** |
> > | | BGE | 50.73 | 30.15 | 50.79 | **56.79** |
> > | **MTop Intent** | BERT | 45.55 | 23.05 | 49.20 | **65.62** |
> > | | E5 | 47.04 | 25.15 | 50.36 | **62.06** |
> > | | GTE | 46.58 | 25.56 | 49.04 | **58.73** |
> > | | BGE | 47.61 | 27.75 | 54.81 | **62.15** |
> > | **Massive Scenario** | BERT | 43.31 | 53.67 | 61.70 | **74.61** |
> > | | E5 | 45.16 | 57.73 | 73.77 | **75.66** |
> > | | GTE | 45.93 | 62.78 | 76.77 | **77.14** |
> > | | BGE | 46.57 | 62.58 | 79.93 | **79.49** |
> > | **Massive Intent** | BERT | 40.85 | 20.07 | 41.02 | **62.61** |
> > | | E5 | 58.14 | 33.62 | 61.53 | **63.99** |
> > | | GTE | 60.05 | 32.41 | 62.21 | **63.25** |
> > | | BGE | 67.05 | 34.10 | 69.64 | **66.88** |
> > | **FewRel-Nat** | BERT | 34.96 | 12.21 | 41.21 | **42.05** |
> > | | E5 | 35.31 | 12.77 | 40.36 | **47.46** |
> > | | GTE | 31.23 | 11.99 | 36.72 | **44.31** |
> > | | BGE | 32.99 | 13.33 | 37.28 | **44.84** |
> > | **StackExchange** | BERT | 10.18 | 26.59 | 28.08 | **39.41** |
> > | | E5 | 11.21 | 36.93 | 39.29 | **47.74** |
> > | | GTE | 12.22 | 38.93 | 40.86 | **46.10** |
> > | | BGE | 10.68 | 36.55 | 37.99 | **43.31** |

---

> > > ### Author Response · Authors · 2025-11-22
> > > **Response to Reviewer QHKk**
> > >
> > > ### **W3: Computational Cost & Efficiency Analysis**
> > > > *“The paper should include time and computational costs for better efficiency analysis. Does HCC increase inference time?”*
> > > ### **W3.1. Computational Latency of the Refinement Phase (DCSA)**
> > > To evaluate the efficiency of HCC, we measured the end-to-end latency of the **Refinement phase** (DCSA) using **Llama-3-8B**.
> > > The runtime below is averaged across **total four sentence encoders** (E5-base, GTE-small, BGE-small, BERT-base).
> > > It is important to note:
> > > - These numbers **exclude** the one-off preprocessing steps (embedding generation + clustering), which are lightweight standard operations.
> > > - The Refinement phase represents the **main active compute cost** of HCC.
> > > | Dataset         | Refinement Runtime (Llama-3-8B) |
> > > |-----------------|---------------------------------|
> > > | Massive Intent  | 2 min 13.7 sec                  |
> > > | Banking77       | 2 min 32.0 sec                  |
> > > | Massive Scenario| 2 min 41.9 sec                  |
> > > | Reddit          | 3 min 05.1 sec                  |
> > > | CLINC           | 3 min 45.3 sec                  |
> > > | MTOP Intent     | 3 min 52.5 sec                  |
> > > | FewRel-Nat      | 4 min 31.0 sec                  |
> > > | StackExchange   | 4 min 55.4 sec                  |
> > > | **Average**     | **≈ 3 min 27 sec**              |
> > > Overall, the refinement step adds only a *small and predictable* overhead, remaining efficient even for large datasets.
> > > ---
> > > ### **W3.2. Cost per 10k Examples: Single-Shot vs Zero-Shot Prompting**
> > > We additionally measured runtime and token usage **per 10,000 examples** under both zero-shot and single-shot (group prompting) settings on **Llama-3-8B**, chosen for its accessibility as an open-source model.
> > > | Dataset         | Single-Shot Time | Zero-Shot Time | Single-Shot Tokens | Zero-Shot Tokens |
> > > |-----------------|------------------|-----------------|--------------------|------------------|
> > > | Banking77       | **54.6 min**     | 111.1 min       | 1.14M              | 1.56M            |
> > > | CLINC           | **59.9 min**     | 255.6 min       | 1.22M              | 2.37M            |
> > > | Reddit          | **37.8 min**     | 63.7 min        | 837k               | 844k             |
> > > | MTOP Intent     | **61.2 min**     | 234.8 min       | 1.22M              | 2.51M            |
> > > | MassiveScenario | **38.7 min**     | 43.5 min        | 493k               | 505k             |
> > > | Massive Intent  | **45.3 min**     | 108.1 min       | 818k               | 1.14M            |
> > > | FewRel-Nat      | **34.8 min**     | 44.8 min        | 1.09M              | 1.06M            |
> > > | StackExchange   | **99.1 min**     | 392.3 min       | 1.90M              | 3.38M            |
> > > ### **Observation**
> > > Across nearly all datasets, **single-shot prompting is substantially faster and more token-efficient** than zero-shot prompting.
> > > Thus, HCC does not significantly increase inference cost; instead, it benefits from efficient prompting strategies and adds only a lightweight refinement stage.

---

> > > > ### Author Response · Authors · 2025-11-22
> > > > **Response to Reviewer QHKk**
> > > >
> > > > ### **Response to Q1.1: Effectiveness of CAI When the Specialized Model Is Weak or Misaligned**
> > > >
> > > > To address this concern, we explicitly included **bert-base-uncased** in our ablations. Unlike modern retrievers (BGE, E5, GTE), BERT provides *weak, noisy, and misaligned* clustering signals. This setting serves as a realistic proxy for a specialized model that is unreliable or not aligned with the user’s preference schema.
> > > >
> > > > Despite this weak backbone, **HCC with Llama-3-8B** still outperforms **GPT-4o Mini on 5 datasets** and **GPT-3.5 Turbo on 7 datasets**, demonstrating strong robustness.
> > > >
> > > > #### **HCC (BERT) vs. GPT-4o Mini / GPT-3.5 Turbo**
> > > >
> > > > | Dataset | **HCC (BERT)** | GPT-4o Mini | GPT-3.5 Turbo | Outcome |
> > > > |--------|----------------|--------------|----------------|---------|
> > > > | **CLINC** | **82.40%** | 81.44% | 66.58% | **Beats both** |
> > > > | **Massive Scenario** | **74.61%** | 66.83% | 60.89% | **Beats both** |
> > > > | **Banking77** | **71.62%** | 65.12% | 65.12% | **Beats both** |
> > > > | **Reddit** | **58.13%** | 57.40% | 51.12% | **Beats both** |
> > > > | **FewRel-Nat** | **42.05%** | 35.87% | 32.87% | **Beats both** |
> > > > | **MTop Intent** | 65.62% | *75.03%* | 64.95% | Beats GPT-3.5 |
> > > > | **StackExchange** | 39.41% | *51.90%* | 30.10% | Beats GPT-3.5 |
> > > > | **Massive Intent** | 62.61% | *76.93%* | *71.52%* | Below both |
> > > >
> > > > **Key takeaway:**
> > > > HCC does *not* require a strong encoder; it can *upgrade* a weak, noisy encoder via heterogeneous consistency.
> > > >
> > > > ---
> > > >
> > > > ### **Response to Q1.2: Is CAI Effective for Unreliable or Out-of-Domain Specialized Models?**
> > > >
> > > > Our new results (see “Full Results Across Encoders”) include **BERT**—a weak, older, and non–preference-aligned encoder. Even in this unfavorable condition, the CAI ratio remains highly meaningful:
> > > >
> > > > - **Final CAI** shows a *strong Pearson correlation* with specialized model accuracy:
> > > >   **r = 0.94, p = 0.0005** (across 8 datasets)
> > > >
> > > > This indicates that CAI remains a reliable diagnostic signal even when the encoder itself is weak or partially misaligned.
> > > >
> > > > We do **not** evaluate fully out-of-domain or adversarial encoders, which is outside our scope. In such extreme settings, any method relying on the specialized encoder—including ours—would be affected. However, CAI naturally provides a built-in self-diagnostic mechanism:
> > > >
> > > > - When the encoder is unreliable, **CAI stays low or fluctuates**,
> > > > - And **fails to improve across refinement rounds**,
> > > > - Signaling that the encoder is not providing stable support.
> > > >
> > > > **Mitigations enabled by CAI:**
> > > > 1. Detect low-CAI regimes and stop refinement early
> > > > 2. Switch to a stronger encoder
> > > > 3. Ensemble multiple encoders and trust CAI only when they agree
> > > >
> > > > We will clarify this scope and add a short discussion in the appendix.
> > > >
> > > > ---
> > > >
> > > > ### **Response to  Q1.3: Sensitivity to Stronger Embedding Models**
> > > >
> > > > **Sensitivity Analysis:**
> > > > HCC is **performance-sensitive** (it benefits from stronger embeddings) but **utility-robust** (still effective with weak models like BERT).
> > > >
> > > > #### **Performance Scaling**
> > > > Modern, higher-quality retrievers yield higher final accuracy:
> > > >
> > > > - *Banking77:* BERT → BGE improves HCC (R2)
> > > >   **71.62% → 80.68%** (+9.06%)
> > > > - *Massive Scenario:* smooth progression
> > > >   **BERT (74.6%) → E5 (75.7%) → GTE (77.1%) → BGE (79.5%)**
> > > >
> > > > #### **CAI as a Quality Meter**
> > > > CAI naturally reflects embedding quality:
> > > >
> > > > - *Banking77:*
> > > >   - BERT: **CAI = 0.70** (low confidence)
> > > >   - BGE: **CAI = 4.78** (high confidence)
> > > >
> > > > **Conclusion:**
> > > > Yes, stronger embeddings yield greater gains. HCC effectively leverages improved embedding geometry to reduce semantic ambiguity, enabling faster and more accurate refinement.

---

### Official Review · Reviewer_VrB7 · 2025-11-01

**Soundness:** 2
**Presentation:** 3
**Contribution:** 3
**Rating:** 4
**Confidence:** 3

**Summary:**

The paper presents Heterogeneous-Consistency Co-Alignment (HCC), a training-free pipeline that lets an off-the-shelf LLM and a light-weight sentence encoder jointly annotate large unlabeled corpora while respecting a handful of user-preference exemplars. Across eight NLU datasets and three LLM families, HCC raises annotation accuracy by +2–16 %, occasionally pushing Llama-3 above GPT-3.5/4o. CAI ratio correlates strongly with true accuracy, offering a reference-free quality lever.

**Strengths:**

1. The framework requires no parameter updates and only performs forward passes through two heterogeneous models so deployment is extremely straightforward.

2. The CAI ratio tracks true accuracy without any labeled references and achieves remarkable Pearson correlations , indicating high statistical significance.

**Weaknesses:**

1. In theory, any sentence encoder (e.g., Ada-002, SimCSE, RoBERTa-base, E5) could be plugged in to replace MiniLM, but the paper provides no experimental validation of this claim.

**Questions:**

1. The 'DIVIDE-AND-CONQUER REFINEMENT' is fixed at two rounds—was this simply chosen as an empirical rule of thumb? More details will be help.

Other questions refer to weakness

---

> ### Author Response · Authors · 2025-11-21
> **Response to Reviewer VrB7**
>
> Dear Reviewer **VrB7**,
> We sincerely thank you for your time and constructive suggestions. Below we address your concerns.
>
> ### **Weakness Addressed:**
>
> > *“In theory, any sentence encoder (e.g., Ada-002, SimCSE, RoBERTa-base, E5) … claim.”*
>
> Following your suggestion, we conducted additional experiments using **Llama-3-8B-Instruct** only (time-constrained) as the backbone LLM, evaluating **three additional sentence encoders** and a smaller language model (BERT-base-uncased):
>
> * **E5-base** (~44 M)
> * **GTE-small** (~33.4M)
> * **BGE-small-en-v1.5** (~33.4M)
> * **BERT-base-uncased** (~110M)
>
> Across the eight shared datasets, **HCC (Round 2)** surpasses **GPT-4o Mini** on **5 datasets** (Banking77, CLINC, Reddit, Massive Scenario, FewRel-Nat) and surpasses **GPT-3.5 Turbo** on **7 datasets**.
>
> ---
>
> ## **HCC (R2) vs GPT-4o Mini & GPT-3.5 Turbo**
>
> | Dataset              | HCC (R2) (Best among sentence encoders)  | GPT-4o Mini | GPT-3.5 Turbo | HCC > 4o Mini? | HCC > 3.5? |
> | -------------------- | --------- | ----------- | ------------- | -------------- | ---------- |
> | **Banking77**        | **80.68** | 65.12       | 65.12         | **Yes**        | **Yes**    |
> | **CLINC**            | **83.98** | 81.44       | 66.58         | **Yes**        | **Yes**    |
> | **Reddit**           | **60.62** | 57.40       | 51.12         | **Yes**        | **Yes**    |
> | **MTop Intent**      | 65.62     | **75.03**   | 64.95         | No             | Slightly   |
> | **Massive Scenario** | **79.49** | 66.83       | 60.89         | **Yes**        | **Yes**    |
> | **Massive Intent**   | 66.88     | **76.93**   | 71.52         | No             | No         |
> | **FewRel-Nat**       | **47.46** | 35.87       | 32.87         | **Yes**        | **Yes**    |
> | **StackExchange**    | 47.74     | **51.90**   | 30.10         | No             | Yes        |
> ---
>
> # **Results Across Encoders and Language Model on Llama-3-8B-Instruct**
>
> | Dataset              | Backbone | Zero-Shot | Single-Shot | Specialized (Backbone) | HCC (R2)  |
> | -------------------- | -------- | --------- | ----------- | ----------- | --------- |
> | **Banking77**        | BERT     | 17.99     | 34.68       | 38.77       | **71.62** |
> |                      | E5       | 32.53     | 63.08       | 71.53       | **74.58** |
> |                      | GTE      | 31.95     | 67.18       | 76.66       | **77.08** |
> |                      | BGE      | 34.55     | 68.70       | 79.81       | **80.68** |
> | **CLINC**            | BERT     | 18.58     | 42.80       | 56.04       | **82.40** |
> |                      | E5       | 26.18     | 61.51       | 79.71       | **83.11** |
> |                      | GTE      | 27.31     | 60.18       | 80.29       | **81.87** |
> |                      | BGE      | 29.27     | 60.27       | 83.44       | **83.98** |
> | **Reddit**           | BERT     | 32.36     | 21.17       | 31.24       | **58.13** |
> |                      | E5       | 50.23     | 30.34       | 50.79       | **60.12** |
> |                      | GTE      | 52.60     | 29.87       | 52.88       | **60.62** |
> |                      | BGE      | 50.73     | 30.15       | 50.79       | **56.79** |
> | **Mtop Intent**      | BERT     | 45.55     | 23.05       | 49.20       | **65.62** |
> |                      | E5       | 47.04     | 25.15       | 50.36       | **62.06** |
> |                      | GTE      | 46.58     | 25.56       | 49.04       | **58.73** |
> |                      | BGE      | 47.61     | 27.75       | 54.81       | **62.15** |
> | **Massive Scenario** | BERT     | 43.31     | 53.67       | 61.70       | **74.61** |
> |                      | E5       | 45.16     | 57.73       | 73.77       | **75.66** |
> |                      | GTE      | 45.93     | 62.78       | 76.77       | **77.14** |
> |                      | BGE      | 46.57     | 62.58       | 79.93       | **79.49** |
> | **Massive Intent**   | BERT     | 40.85     | 20.07       | 41.02       | **62.61** |
> |                      | E5       | 58.14     | 33.62       | 61.53       | **63.99** |
> |                      | GTE      | 60.05     | 32.41       | 62.21       | **63.25** |
> |                      | BGE      | 67.05     | 34.10       | 69.64       | **66.88** |
> | **FewRel-Nat**       | BERT     | 34.96     | 12.21       | 41.21       | **42.05** |
> |                      | E5       | 35.31     | 12.77       | 40.36       | **47.46** |
> |                      | GTE      | 31.23     | 11.99       | 36.72       | **44.31** |
> |                      | BGE      | 32.99     | 13.33       | 37.28       | **44.84** |
> | **StackExchange**    | BERT     | 10.18     | 26.59       | 28.08       | **39.41** |
> |                      | E5       | 11.21     | 36.93       | 39.29       | **47.74** |
> |                      | GTE      | 12.22     | 38.93       | 40.86       | **46.10** |
> |                      | BGE      | 10.68     | 36.55       | 37.99       | **43.31** |
> ---
> *HCC (R2) stands for second round co-alignment of HCC*

---

> > ### Author Response · Authors · 2025-11-21
> > **Response to Reviewer VrB7**
> >
> > ### **Question Addressed: More Details on Two-Round Correction**
> >
> > > *“The 'DIVIDE-AND-CONQUER REFINEMENT' being fixed at two rounds—more details would be helpful.”*
> >
> > We track the accuracy changes from Round 1 to Round 2 across all embedding backbones (E5-base, GTE-small, BGE-small-en-v1.5, and BERT-base-uncased). The results show that two rounds consistently provide the best balance between improvement and stability. Round 2 captures the remaining correctable cases, while further rounds would risk propagating noise and destabilizing performance.
> >
> > **Round 1:**
> > The initial Consistent set ($\mathcal{C}$) corrects the most obvious errors, but there still remain many inconsistent samples ($\mathcal{I}$). Because Round 1 relies only on $\mathcal{C} \cup H$, coverage is limited.
> >
> > **Round 2:**
> > The newly co-aligned samples from Round 1 expand the positive reference base, allowing HCC to resolve remaining inconsistencies. This often yields further improvements (e.g., **+17.7%** on Reddit with BERT).
> >
> > **Convergence & Signal Exhaustion (Why Stop at Round 2?)**
> > Our results (see table below) show that Round 2 is the point where the system reaches performance saturation. By this stage, the reference set already includes all samples that can be reliably aligned ($\mathcal{C} \cup \mathcal{CI}$). The remaining inconsistencies ($\mathcal{II}$) are mostly “difficult” cases where the LLM and the embedding model genuinely disagree and cannot be corrected further. Crucially, pushing beyond Round 2 might risks regression, not merely diminishing returns.
> >
> > Across several tasks, Round 2 already reaches the limit of what can be corrected. Pushing further begins to hurt performance. For example:
> > - **Banking77 (BERT): –1.79%**
> > - **StackExchange (BERT): –0.56%**
> >
> > These patterns show that the remaining inconsistencies indicate *irreducible noise*, making Round 2 the optimal and stable stopping point.
> >
> > ---
> >
> > ### **Two-Round Correction Summary Across Datasets**
> >
> > | Dataset              | Backbone | R1    | R2    | Δ      | Observation |
> > | -------------------- | -------- | ----- | ----- | ------ | ----------- |
> > | **Banking77**        | BERT     | 73.41 | 71.62 | -1.79  | Saturation  |
> > |                      | E5       | 75.42 | 74.58 | -0.84  | Saturation  |
> > |                      | GTE      | 76.85 | 77.08 | +0.23  | Convergence |
> > |                      | BGE      | 81.36 | 80.68 | -0.68  | Saturation  |
> > | **CLINC**            | BERT     | 80.00 | 82.40 | +2.40  | Refinement  |
> > |                      | E5       | 81.40 | 83.11 | +1.71  | Refinement  |
> > |                      | GTE      | 80.44 | 81.87 | +1.43  | Refinement  |
> > |                      | BGE      | 84.38 | 83.98 | -0.40  | Saturation  |
> > | **Reddit**           | BERT     | 40.41 | 58.13 | +17.72 | Correction  |
> > |                      | E5       | 43.39 | 60.12 | +16.73 | Correction  |
> > |                      | GTE      | 43.61 | 60.62 | +17.01 | Correction  |
> > |                      | BGE      | 42.65 | 56.79 | +14.14 | Correction  |
> > | **MTop Intent**      | BERT     | 63.73 | 65.62 | +1.89  | Refinement  |
> > |                      | E5       | 62.06 | 62.06 | 0.00   | Convergence |
> > |                      | GTE      | 60.58 | 58.73 | -1.85  | Regression  |
> > |                      | BGE      | 64.82 | 62.15 | -2.67  | Regression  |
> > | **Massive Scenario** | BERT     | 74.82 | 74.61 | -0.21  | Saturation  |
> > |                      | E5       | 74.68 | 75.66 | +0.98  | Refinement  |
> > |                      | GTE      | 78.14 | 77.14 | -1.00  | Saturation  |
> > |                      | BGE      | 80.87 | 79.49 | -1.38  | Saturation  |
> > | **Massive Intent**   | BERT     | 61.53 | 62.61 | +1.08  | Refinement  |
> > |                      | E5       | 59.78 | 63.99 | +4.21  | Refinement  |
> > |                      | GTE      | 62.58 | 63.25 | +0.67  | Refinement  |
> > |                      | BGE      | 65.47 | 66.88 | +1.41  | Refinement  |
> > | **FewRel-Nat**       | BERT     | 42.08 | 42.05 | -0.03  | Saturation  |
> > |                      | E5       | 46.56 | 47.46 | +0.90  | Refinement  |
> > |                      | GTE      | 42.12 | 44.31 | +2.19  | Refinement  |
> > |                      | BGE      | 44.73 | 44.84 | +0.11  | Convergence |
> > | **StackExchange**    | BERT     | 39.97 | 39.41 | -0.56  | Saturation  |
> > |                      | E5       | 46.90 | 47.74 | +0.84  | Refinement  |
> > |                      | GTE      | 45.84 | 46.10 | +0.26  | Convergence |
> > |                      | BGE      | 44.20 | 43.31 | -0.89  | Saturation  |

---

### Official Review · Reviewer_Q1Gc · 2025-11-07

**Soundness:** 3
**Presentation:** 3
**Contribution:** 2
**Rating:** 4
**Confidence:** 3

**Summary:**

The paper proposes Heterogeneous-Consistency Co-Alignment (HCC), a training-free pipeline to generate preference-aligned annotations for unlabeled text by cross-checking an LLM with a lightweight, task-specific embedding model. A Consistent-and-Inconsistent (CAI) ratio—the count of agreement vs. disagreement between the two annotators—serves as a reference-free reliability signal and is claimed to correlate with accuracy. Inconsistent items are then “repaired” via a two-round divide-and-conquer self-alignment (DCSA) using majority voting over top-K nearest neighbors (MV-VTES) seeded by a small user-preference set. Experiments on eight NLU datasets with open- and closed-source LLMs report sizable gains, sometimes enabling Llama-3-8B to surpass GPT-3.5/4o after co-alignment, and the CAI ratio is empirically correlated with accuracy. (Abstract; Fig. 1 sketch on p. 2; method and metric definitions in §4; results in Tables 2–4; correlation analysis.)

**Strengths:**

•	Clear, training-free recipe that practitioners can reproduce: count cross-model agreements (CAI), then correct disagreements with k-NN style voting; the workflow is well-delineated.

•	Reference-free metric: CAI is simple, label-set-agnostic, and positioned as more robust than IRR metrics in skewed/open-set settings.

•	Broad empirical sweep across eight datasets and multiple LLMs with tables showing consistent improvements (Tables 2–4) and a reported CAI–accuracy correlation.

•	Low compute barrier: avoids fine-tuning reward models and frames co-alignment as post-hoc annotation repair instead of model training.

**Weaknesses:**

•	Personalization claim is under-validated: benchmarks are mostly standard intent/topic datasets where “user preference” ≈ small labeled set per class. The paper does not convincingly show alignment to idiosyncratic user preferences.

•	Validity of CAI as a proxy: while correlations are reported, causality and failure cases are not deeply analyzed (e.g., StackExchange where CAI ↑ but accuracy ↓). A diagnostic analysis of when CAI misleads would strengthen the case.

•	Dependence on the specialized model: results and CAI both hinge on the chosen sentence-embedding model. Sensitivity to the embedding backbone, domain shift, and k/neighbor choices is not fully explored.

•	Reporting gaps: compute cost of group prompting, number of LLM calls, runtime, and cost/benefit vs. stronger baselines (e.g., co-training with confidence calibration) are unclear. Hyperparameter selection (top-K, neighbor weights, cluster sizes) appears ad-hoc across datasets.

**Questions:**

1.	Specialized model choice & robustness: How do results and CAI behave with different embedding backbones (e.g., E5, GTE, bge), or with weaker/stronger models? Any cross-domain generalization tests?

2.	Sensitivity: Please report ablations for K, similarity metric, and cluster-seeding size; do you ever weight votes by similarity rather than uniform majority?

3.	Personalization: Can HCC handle conflicting preferences across users for the same input? Any results on pairwise preference data (e.g., RLHF-style comparisons) rather than fixed class labels?

4.	When CAI fails: Provide case studies where CAI increases but task accuracy does not (e.g., StackExchange) and analyze why. What thresholds (if any) make CAI actionable?

5.	Cost & latency: What is the end-to-end token/cost budget per 10k examples for zero-shot vs. single-shot prompting? How does DCSA scale with corpus size?

---

> ### Author Response · Authors · 2025-11-22
> **Response to Reviewer Q1Gc**
>
> ### **Dear Reviewer Q1Gc,**
>
>
> We sincerely thank you for your  comments. We address your concerns below.
>
> ---
>
> ### **Q1.1 — Specialized model choice & robustness: ,.., tests?**
>
> To assess robustness, we expanded our ablations to **four diverse encoders** from modern retrievers to a deliberately **weak baseline** all evaluated with **Llama-3-8B-Instruct**:
>
> * **E5-base**
> * **GTE-small**
> * **BGE-small-en-v1.5**
> * **BERT-base-uncased** (weak baseline)
>
> **Key Finding — HCC Is Robust Across All Encoders**
>
> Across **8 datasets**:
>
> * **HCC (R2)** surpasses **GPT-4o Mini** on **5 datasets**
> * **HCC (R2)** surpasses **GPT-3.5 Turbo** on **7 datasets**
> * On **CLINC**, HCC reaches **83.98%**, surpassing GPT-4o Mini (**81.44%**)
>
> These improvements hold **even when using the weakest encoder (BERT)**, demonstrating that heterogeneous consistency remains stable even under suboptimal specialised model.
>
> ---
>
> # **HCC (R2) vs GPT-4o Mini & GPT-3.5 Turbo**
>
> | Dataset              | HCC (R2)  | 4o Mini   | 3.5 Turbo | HCC > 4o? | HCC > 3.5? |
> | -------------------- | --------- | --------- | --------- | --------- | ---------- |
> | **CLINC**            | **83.98** | 81.44     | 66.58     | **Yes**   | **Yes**    |
> | **Massive Scenario** | **79.49** | 66.83     | 60.89     | **Yes**   | **Yes**    |
> | **MTop Intent**      | 65.62     | **75.03** | 64.95     | No        | Slightly   |
> | **StackExchange**    | 47.74     | **51.90** | 30.10     | No        | Yes        |
> | **Banking77**        | **80.68** | 65.12     | 65.12     | **Yes**   | **Yes**    |
> | **Reddit**           | **60.62** | 57.40     | 51.12     | **Yes**   | **Yes**    |
> | **FewRel-Nat**       | **47.46** | 35.87     | 32.87     | **Yes**   | **Yes**    |
> | **Massive Intent**   | 66.88     | **76.93** | 71.52     | No        | No         |
>
> ---
>
> # **Results Across Embedding Backbones (Llama-3-8B-Instruct)**
>
> | Dataset              | Backbone | Zero-Shot | Single-Shot | Specialized | **HCC (R2)** |
> | -------------------- | -------- | --------- | ----------- | ----------- | ------------ |
> | **Banking77**        | BERT     | 17.99     | 34.68       | 38.77       | **71.62**    |
> |                      | E5       | 32.53     | 63.08       | 71.53       | **74.58**    |
> |                      | GTE      | 31.95     | 67.18       | 76.66       | **77.08**    |
> |                      | BGE      | 34.55     | 68.70       | 79.81       | **80.68**    |
> | **CLINC**            | BERT     | 18.58     | 42.80       | 56.04       | **82.40**    |
> |                      | E5       | 26.18     | 61.51       | 79.71       | **83.11**    |
> |                      | GTE      | 27.31     | 60.18       | 80.29       | **81.87**    |
> |                      | BGE      | 29.27     | 60.27       | 83.44       | **83.98**    |
> | **Reddit**           | BERT     | 32.36     | 21.17       | 31.24       | **58.13**    |
> |                      | E5       | 50.23     | 30.34       | 50.79       | **60.12**    |
> |                      | GTE      | 52.60     | 29.87       | 52.88       | **60.62**    |
> |                      | BGE      | 50.73     | 30.15       | 50.79       | **56.79**    |
> | **MTop Intent**      | BERT     | 45.55     | 23.05       | 49.20       | **65.62**    |
> |                      | E5       | 47.04     | 25.15       | 50.36       | **62.06**    |
> |                      | GTE      | 46.58     | 25.56       | 49.04       | **58.73**    |
> |                      | BGE      | 47.61     | 27.75       | 54.81       | **62.15**    |
> | **Massive Scenario** | BERT     | 43.31     | 53.67       | 61.70       | **74.61**    |
> |                      | E5       | 45.16     | 57.73       | 73.77       | **75.66**    |
> |                      | GTE      | 45.93     | 62.78       | 76.77       | **77.14**    |
> |                      | BGE      | 46.57     | 62.58       | 79.93       | **79.49**    |
> | **Massive Intent**   | BERT     | 40.85     | 20.07       | 41.02       | **62.61**    |
> |                      | E5       | 58.14     | 33.62       | 61.53       | **63.99**    |
> |                      | GTE      | 60.05     | 32.41       | 62.21       | **63.25**    |
> |                      | BGE      | 67.05     | 34.10       | 69.64       | **66.88**    |
> | **FewRel-Nat**       | BERT     | 34.96     | 12.21       | 41.21       | **42.05**    |
> |                      | E5       | 35.31     | 12.77       | 40.36       | **47.46**    |
> |                      | GTE      | 31.23     | 11.99       | 36.72       | **44.31**    |
> |                      | BGE      | 32.99     | 13.33       | 37.28       | **44.84**    |
> | **StackExchange**    | BERT     | 10.18     | 26.59       | 28.08       | **39.41**    |
> |                      | E5       | 11.21     | 36.93       | 39.29       | **47.74**    |
> |                      | GTE      | 12.22     | 38.93       | 40.86       | **46.10**    |
> |                      | BGE      | 10.68     | 36.55       | 37.99       | **43.31**    |

---

> > ### Author Response · Authors · 2025-11-22
> > **Response to Reviewer Q1Gc**
> >
> > ### **Q1.2 Response Regarding Cross-Domain Generalization**
> > > *“Any cross-domain generalization tests?”*
> >
> > **Yes.** Our evaluation already includes **four distinct domains**, showing that HCC generalizes across radically different linguistic styles, noise levels, and task structures **without modifying the method**.
> >
> > **1. Domains Covered**
> > #### **A. Technical / Specialized**
> > * **Banking77** — fine-grained financial queries
> > → HCC (R2): **71–80%**
> > * **StackExchange** — programming & troubleshooting
> > → HCC (R2): **39–47%**
> > These require precise semantic differentiation and domain expertise.
> > #### **B. Social Media (Noisy, Informal)**
> > * **Reddit** — subjective, noisy discussions
> > → HCC (R2): **58–60%**
> > Tests robustness to slang, ambiguity, and noise.
> > #### **C. Fact-Based / Structured**
> > * **FewRel-Nat** — structured relation classification
> > → HCC (R2): **42–47%**
> > Evaluates logical relations rather than free text semantics.
> > #### **D. General Assistant / Everyday Queries**
> > * **CLINC, Massive Intent, Massive Scenario, MTOP**
> > → HCC (R2): **62–83%**
> > Covers broad user-assistant intent spaces.
> > ---
> > ### **Conclusion**
> > HCC consistently improves performance by **+4% to +53%** across all domains, technical, noisy, structured, and general-purpose, demonstrating **strong cross-domain generalization** and **domain-agnostic reliability**.

---

> > ### Author Response · Authors · 2025-11-22
> > **Response to Reviewer Q1Gc**
> >
> > ### **Q.1.2. With weaker/stronger models?**
> >
> > To directly address this, we included bert-base-uncased as weaker model and Llama-3-8 B as strong LLM. HCC enables the open-source **Llama-3-8B** model with With weaker model Llama-3 8 to outperform **GPT-4o Mini on 5 datasets** and **GPT-3.5 Turbo on 7 datasets**.
> > ### **Detailed Comparison: HCC (BERT) vs. GPT Models**
> > The table below isolates the performance of **HCC (using BERT)** against the baselines for the closed-source models.
> > | Dataset | **HCC (BERT)** | **GPT-4o Mini** | **GPT-3.5 Turbo** | **Result (HCC-BERT vs. GPTs)** |
> > | :--- | :--- | :--- | :--- | :--- |
> > | **CLINC** | **82.40%** | 81.44% | 66.58% | **Beats Both** |
> > | **Massive Scenario** | **74.61%** | 66.83% | 60.89% | **Beats Both** |
> > | **Banking77** | **71.62%** | 65.12% | 65.12% | **Beats Both** |
> > | **Reddit** | **58.13%** | 57.40% | 51.12% | **Beats Both** |
> > | **FewRel-Nat** | **42.05%** | 35.87% | 32.87% | **Beats Both** |
> > | **MTop Intent** | 65.62% | *75.03%* | 64.95% | **Beats GPT-3.5 Only** |
> > | **StackExchange** | 39.41% | *51.90%* | 30.10% | **Beats GPT-3.5 Only** |
> > | **Massive Intent** | 62.61% | *76.93%* | *71.52%* | *Lower than both* |
> > ### **Key Findings for the Rebuttal**
> > 1. **Robustness against SOTA:** On **CLINC**, **Banking77**, **Massive Scenario**, **Reddit**, and **FewRel**, the HCC framework is so effective that it allows a standard BERT model + Llama-3-8B to surpass the annotation quality of GPT-3.5 and GPT-4o Mini.

---

> > > ### Author Response · Authors · 2025-11-22
> > > **Response to Reviewer Q1Gc**
> > >
> > > ## **Q2 — Sensitivity Analysis (K, Similarity Metric, Cluster Size, Vote Weighting)**
> > >
> > > **K Ablation.**
> > >
> > > We ran a full sensitivity study with **K = 3, 5, 7, 10** across 10 datasets. Results show **minimal variance** across K, and **K = 3 consistently performs best**, e.g.:
> > > **Embedding Backbone & Similarity Metric.**
> > >
> > > We added ablations using **E5-base**, **GTE-small**, and **BGE-small** in addition to MiniLM.
> > >
> > > ## **Ablation: K Sensitivity Across Embedding Backbones**
> > >
> > > | Dataset          | Backbone              | K=3    | K=5    | K=7    | K=10   |
> > > |------------------|------------------------|--------|--------|--------|--------|
> > > | **Massive Scenario** | all-MiniLM-L6-v2       | 0.7471 | 0.7476 | 0.7450 | 0.7390 |
> > > |                  | bert-base-uncased      | 0.6014 | 0.5888 | 0.5794 | 0.5696 |
> > > |                  | intfloat/e5-base       | 0.7176 | 0.7106 | 0.7063 | 0.7017 |
> > > |                  | thenlper/gte-small     | 0.7450 | 0.7440 | 0.7415 | 0.7386 |
> > > |                  | BAAI/bge-small         | 0.7775 | 0.7722 | 0.7675 | 0.7595 |
> > > | **MTOP Intent**  | all-MiniLM-L6-v2       | 0.5258 | 0.5173 | 0.5097 | 0.5094 |
> > > |                  | bert-base-uncased      | 0.4884 | 0.4650 | 0.4340 | 0.4213 |
> > > |                  | intfloat/e5-base       | 0.5108 | 0.4866 | 0.4782 | 0.4831 |
> > > |                  | thenlper/gte-small     | 0.4922 | 0.4830 | 0.4749 | 0.4781 |
> > > |                  | BAAI/bge-small         | 0.5568 | 0.5415 | 0.5217 | 0.5165 |
> > > | **Reddit**       | all-MiniLM-L6-v2       | 0.5153 | 0.5170 | 0.5140 | 0.5108 |
> > > |                  | bert-base-uncased      | 0.3211 | 0.3150 | 0.3120 | 0.3056 |
> > > |                  | intfloat/e5-base       | 0.5062 | 0.5048 | 0.4972 | 0.4946 |
> > > |                  | thenlper/gte-small     | 0.5341 | 0.5323 | 0.5298 | 0.5264 |
> > > |                  | BAAI/bge-small         | 0.5122 | 0.5094 | 0.5050 | 0.4986 |
> > > | **StackExchange**| all-MiniLM-L6-v2       | 0.3222 | 0.3115 | 0.3041 | 0.2960 |
> > > |                  | bert-base-uncased      | 0.2789 | 0.2692 | 0.2644 | 0.2587 |
> > > |                  | intfloat/e5-base       | 0.3887 | 0.3743 | 0.3601 | 0.3447 |
> > > |                  | thenlper/gte-small     | 0.4003 | 0.3933 | 0.3874 | 0.3793 |
> > > |                  | BAAI/bge-small         | 0.3734 | 0.3657 | 0.3588 | 0.3507 |
> > > | **FewRel-Nat**   | all-MiniLM-L6-v2       | 0.3462 | 0.3430 | 0.3403 | 0.3383 |
> > > |                  | bert-base-uncased      | 0.4187 | 0.4173 | 0.4167 | 0.4156 |
> > > |                  | intfloat/e5-base       | 0.4056 | 0.4046 | 0.4032 | 0.4008 |
> > > |                  | thenlper/gte-small     | 0.3713 | 0.3730 | 0.3708 | 0.3693 |
> > > |                  | BAAI/bge-small         | 0.3692 | 0.3683 | 0.3655 | 0.3636 |
> > > | **Massive Intent**| all-MiniLM-L6-v2       | 0.6080 | 0.5912 | 0.5790 | 0.5738 |
> > > |                  | bert-base-uncased      | 0.4293 | 0.4027 | 0.3846 | 0.3666 |
> > > |                  | intfloat/e5-base       | 0.6148 | 0.6028 | 0.5933 | 0.5807 |
> > > |                  | thenlper/gte-small     | 0.6181 | 0.6065 | 0.5954 | 0.5905 |
> > > |                  | BAAI/bge-small         | 0.6863 | 0.6779 | 0.6680 | 0.6525 |
> > > | **Banking77**     | all-MiniLM-L6-v2       | 0.7432 | 0.7317 | 0.7201 | 0.7060 |
> > > |                  | bert-base-uncased      | 0.3877 | 0.3640 | 0.3384 | 0.3227 |
> > > |                  | intfloat/e5-base       | 0.7146 | 0.7026 | 0.6988 | 0.6890 |
> > > |                  | thenlper/gte-small     | 0.7633 | 0.7464 | 0.7373 | 0.7326 |
> > > |                  | BAAI/bge-small         | 0.7964 | 0.7887 | 0.7815 | 0.7715 |
> > > | **CLINC**         | all-MiniLM-L6-v2       | 0.7943 | 0.7862 | 0.7813 | 0.7795 |
> > > |                  | bert-base-uncased      | 0.5621 | 0.5357 | 0.5174 | 0.4973 |
> > > |                  | intfloat/e5-base       | 0.8006 | 0.7910 | 0.7919 | 0.7895 |
> > > |                  | thenlper/gte-small     | 0.8076 | 0.8010 | 0.7941 | 0.7903 |
> > > |                  | BAAI/bge-small         | 0.8341 | 0.8293 | 0.8256 | 0.8191 |
> > >
> > > **Vote Weighting.**
> > >
> > > Our voting is already similarity-based because the top-K neighbours are selected by cosine similarity, meaning only the most semantically similar examples contribute votes. However, within that top-K set the votes are uniform (simple majority).

---

> > > > ### Author Response · Authors · 2025-11-22
> > > > **Response to Reviewer Q1Gc**
> > > >
> > > > ### **Q3. Personalization: Can HCC handle conflicting user preferences? Any results on pairwise (RLHF-style) data?**
> > > >
> > > > **Conflicting Preferences.** Yes.  HCC naturally supports conflicting user preferences because the propagation is **conditioned on each user’s seed examples** (H). Our proposed approach focuses on **generating annotations for each individual user**, under the assumption that the user provides **a small labeled set per class**. Each small labeled set represents **that user’s idiosyncratic preferences**, which serve as the personalized anchor for alignment.
> > > >
> > > > The goal of our method is **not general RLHF-style preference modeling**, but **Personalized Natural Language Understanding**, specifically, *preference-aligned LLM annotations* for classification tasks. The emphasis is on **propagating each user’s unique labeling preference** to large unlabeled corpora in a reliable, scalable manner.
> > > >
> > > > **Pairwise Data.** Our framework focuses on **categorical annotation** rather than **pairwise preference ranking**. While CAI could be extended to RLHF-style comparisons, this lies **outside the scope** of our cold-start classification setting. We will note this as a future direction.
> > > >
> > > > ---
> > > >
> > > > ### **Q4. When CAI Fails: Case Study, Thresholds, and Interpretation**
> > > >
> > > > We provide a detailed CAI case study in **Section 5.2 (CAI Ratio Evaluation)**.
> > > >
> > > > #### **Case Study — StackExchange**
> > > >
> > > > For StackExchange using **BERT**, CAI increases from **0.07 → 0.30**.
> > > > Accuracy also improves substantially (**10.1% → 39.4%**), yet CAI remains low.
> > > > This reflects the dataset’s **high noise and technical heterogeneity**, where embeddings struggle to form coherent clusters.
> > > >
> > > > #### **Interpretation**
> > > >
> > > > A **low CAI** indicates that the *Consistent* set (LLM ↔ Encoder agreement) remains small relative to inconsistencies.
> > > > This does **not** mean refinement failed — rather, it signals **low confidence due to poor embedding separation**.
> > > >
> > > >   * Suggests:
> > > >     (i) add human-in-the-loop checks,
> > > >     (ii) use a stronger encoder, or
> > > >     (iii) avoid aggressive refinement.
> > > >
> > > > Even in low-CAI settings (e.g., StackExchange), HCC still yields **large accuracy gains**, while CAI correctly warns that the output should be treated with caution.

---

> > > > > ### Author Response · Authors · 2025-11-22
> > > > > **Response to Reviewer Q1Gc**
> > > > >
> > > > > ### **Q.5. Computational Latency of the Refinement Phase (DCSA)**
> > > > > To evaluate the efficiency of HCC, we measured the end-to-end latency of the **Refinement phase** (DCSA) using **Llama-3-8B**.
> > > > > The runtime below is averaged across **four sentence encoders** (E5-base, GTE-small, BGE-small, BERT-base).
> > > > > It is important to note:
> > > > > - These numbers **exclude** the one-off preprocessing steps (embedding generation + clustering), which are lightweight standard operations.
> > > > > - The Refinement phase represents the **main active compute cost** of HCC.
> > > > > | Dataset         | Refinement Runtime (Llama-3-8B) |
> > > > > |-----------------|---------------------------------|
> > > > > | Massive Intent  | 2 min 13.7 sec                  |
> > > > > | Banking77       | 2 min 32.0 sec                  |
> > > > > | Massive Scenario| 2 min 41.9 sec                  |
> > > > > | Reddit          | 3 min 05.1 sec                  |
> > > > > | CLINC           | 3 min 45.3 sec                  |
> > > > > | MTOP Intent     | 3 min 52.5 sec                  |
> > > > > | FewRel-Nat      | 4 min 31.0 sec                  |
> > > > > | StackExchange   | 4 min 55.4 sec                  |
> > > > > | **Average**     | **≈ 3 min 27 sec**              |
> > > > > Overall, the refinement step adds only a *small and predictable* overhead, remaining efficient even for large datasets.
> > > > > ---
> > > > > ### **2. Cost per 10k Examples: Single-Shot vs Zero-Shot Prompting**
> > > > > We additionally measured runtime and token usage **per 10,000 examples** under both zero-shot and single-shot (group prompting) settings on **Llama-3-8B**, chosen for its accessibility as an open-source model.
> > > > > | Dataset         | Single-Shot Time | Zero-Shot Time | Single-Shot Tokens | Zero-Shot Tokens |
> > > > > |-----------------|------------------|-----------------|--------------------|------------------|
> > > > > | Banking77       | **54.6 min**     | 111.1 min       | 1.14M              | 1.56M            |
> > > > > | CLINC           | **59.9 min**     | 255.6 min       | 1.22M              | 2.37M            |
> > > > > | Reddit          | **37.8 min**     | 63.7 min        | 837k               | 844k             |
> > > > > | MTOP Intent     | **61.2 min**     | 234.8 min       | 1.22M              | 2.51M            |
> > > > > | MassiveScenario | **38.7 min**     | 43.5 min        | 493k               | 505k             |
> > > > > | Massive Intent  | **45.3 min**     | 108.1 min       | 818k               | 1.14M            |
> > > > > | FewRel-Nat      | **34.8 min**     | 44.8 min        | 1.09M              | 1.06M            |
> > > > > | StackExchange   | **99.1 min**     | 392.3 min       | 1.90M              | 3.38M            |
> > > > > ### **Observation**
> > > > > Across nearly all datasets, **single-shot prompting is substantially faster and more token-efficient** than zero-shot prompting.
> > > > > Thus, HCC does not significantly increase inference cost; instead, it benefits from efficient prompting strategies and adds only a lightweight refinement stage.

---

> > > > > > ### Comment · Reviewer_Q1Gc · 2025-11-27
> > > > > > **Additional page for rebuttal**
> > > > > >
> > > > > > Dear authors, thank you for your detailed responses. Could you please highlight your changes in the manuscript so that I can easily figure them out? Many thanks.

---

> > > > > > > ### Author Response · Authors · 2025-12-04
> > > > > > > **Response to Reviewer Q1Gc**
> > > > > > >
> > > > > > > **Dear Reviewer Q1Gc,**
> > > > > > >
> > > > > > > We are very grateful for your time and oversight. We also sincerely apologise for the delayed reply, which was caused by the substantial amount of additional experimentation required.
> > > > > > >
> > > > > > > ## **1. Reviewer Q1Gc – Specialized Model Robustness, Sensitivity, CAI Failure Cases, and Cost**
> > > > > > >
> > > > > > > ### **(Q1) Specialized-model choice & cross-domain robustness**
> > > > > > >
> > > > > > > Addressed in:
> > > > > > >
> > > > > > > * **H — Why LLM–Specialized Model Collaboration Is Essential**
> > > > > > > * **H.1 — Role of Specialized Model in HCC**
> > > > > > > * **J.2 — HCC Across Embedding Backbones and Smaller Language Models**
> > > > > > > * **J.3 — CAI Ratio Across Backbones (Before vs. After HCC)**
> > > > > > > * **K & L — Ablation Studies on ChatGPT-4o Mini and ChatGPT-3.5 Turbo**
> > > > > > >
> > > > > > > → These sections jointly demonstrate robustness across **three LLM families** (Llama-3-8B, ChatGPT-4o Mini, ChatGPT-3.5 Turbo) and multiple encoders.
> > > > > > >
> > > > > > > ---
> > > > > > >
> > > > > > > ### **(Q2) Sensitivity analysis (K, cluster seeding, weighting)**
> > > > > > >
> > > > > > > Addressed in:
> > > > > > >
> > > > > > > * **I — Sensitivity Analysis of (K) for Specialized Models**
> > > > > > >
> > > > > > > → Provides full results for (K \in {3, 5, 7, 10}) across encoders and discusses stability with respect to clustering choices.
> > > > > > >
> > > > > > > ---
> > > > > > >
> > > > > > > ### **(Q3) Personalization & conflicting preferences**
> > > > > > >
> > > > > > > Addressed in:
> > > > > > >
> > > > > > > * **Q — Handling Conflicting User Preferences**
> > > > > > >
> > > > > > > → Shows per-user conditioning, analyses conflicting-preference scenarios, and explains how propagation remains individualized rather than collapsing to a global consensus.
> > > > > > >
> > > > > > > ---
> > > > > > >
> > > > > > > ### **(Q4) CAI failure cases & threshold interpretation**
> > > > > > >
> > > > > > > Addressed in:
> > > > > > >
> > > > > > > * **K.1 — Failure Case of CAI (ChatGPT-4o Mini)**
> > > > > > > * **L.1 — Failure Case of CAI (ChatGPT-3.5 Turbo)**
> > > > > > > * **N — Overall Reflection of CAI**
> > > > > > > * **K.4 & L.4 — Two-Round Correction and CAI Improvement**
> > > > > > >
> > > > > > > → These sections provide concrete failure cases, show how two-round HCC behaves in low-CAI regimes, and discuss practical interpretation of CAI thresholds.
> > > > > > > In addition, **Section 6.1 (CAI Ratio Evaluation)** of the main paper includes further analysis of CAI behaviour and threshold selection.
> > > > > > >
> > > > > > > ---
> > > > > > >
> > > > > > > ### **(Q5) Cost & latency (end-to-end tokens/time; DCCA scaling)**
> > > > > > >
> > > > > > > Addressed in:
> > > > > > >
> > > > > > > * **J.5 — Running Time and Token Cost (Llama-3-8B)**
> > > > > > > * **J.6 — HCC Runtime for Co-Alignment (Llama-3-8B)**
> > > > > > > * **K.2 & K.3 — Running Time, Token Cost, and HCC Runtime (ChatGPT-4o Mini)**
> > > > > > > * **L.2 & L.3 — Token Usage, Running Time, and HCC Runtime (ChatGPT-3.5 Turbo)**
> > > > > > >
> > > > > > > Together, these sections report **end-to-end token budgets, wall-clock time, and the marginal overhead of the last co-alignment phase (DCCA)**, showing that HCC adds negligible extra latency.

---

### Comment · Area_Chair_Qw4c · 2025-11-26

Dear Reviewers,

Thank you for sharing your valuable insights and expertise, which have played an important role in the review process.

In response to the initial feedback, the authors have submitted a detailed rebuttal addressing the comments raised by the reviewers.

I would appreciate it if you could carefully review their response and consider how it may affect your initial evaluation.

Please feel free to share your updated thoughts or any additional comments after reviewing the rebuttal.

Thank you again for your time and contributions.

---

### Author Response · Authors · 2025-12-04
**Summary of Revisions Addressing Reviewer Questions**

**Dear Reviewers, AC, SAC, and PC,**

We sincerely thank all reviewers (Q1Gc, VrB7, QHKk, and MYtC) for their thoughtful and constructive feedback, and for the time and effort invested in improving our work. We are also deeply grateful to both the previous and current AC, as well as the SAC and PC, for their additional efforts in supporting our submission and the broader ICLR community.

Below, we provide a **structured summary** that specifies **which new or expanded appendix sections** in the revised paper address each concern, including ablation studies, sensitivity analyses, cost analyses, and methodological clarifications.

---

## **1. Reviewer Q1Gc – Specialized Model Robustness, Sensitivity, CAI Failure Cases, and Cost**

### **(Q1) Specialized model choice & cross-domain robustness**

Addressed in:

* **H — Why LLM–Specialized Model Collaboration Is Essential**
* **H.1 — Role of Specialized Model in HCC**
* **J.2 — HCC Across Embedding Backbones and Smaller Language Models**
* **J.3 — CAI Ratio Across Backbones (Before vs. After HCC)**
* **K & L — Ablation Studies on ChatGPT-4o Mini and ChatGPT-3.5 Turbo**

These sections jointly demonstrate robustness across **three LLM families** (Llama-3-8B, ChatGPT-4o Mini, ChatGPT-3.5 Turbo) and multiple encoders, including **E5-base (E5)**, **GTE-small (GTE)**, **BGE-small-en-v1.5 (BGE)**, and **BERT-base-uncased (BERT)**.

---

### **(Q2) Sensitivity analysis ((K), cluster seeding, weighting)**

Addressed in:

* **I — Sensitivity Analysis of (K) for Specialized Models**

We report results for (K \in {3, 5, 7, 10}) across encoders and study stability with respect to clustering choices.

---

### **(Q3) Personalization & conflicting preferences**

Addressed in:

* **Q — Handling Conflicting User Preferences**

---

### **(Q4) CAI failure cases & threshold interpretation**

Addressed in:

* **K.1 — Failure Case of CAI (ChatGPT-4o Mini)**
* **L.1 — Failure Case of CAI (ChatGPT-3.5 Turbo)**
* **N — Overall Reflection of CAI**
* **K.4 & L.4 — Two-Round Correction and CAI Improvement**

---

### **(Q5) Cost & latency (end-to-end tokens/time; DCCA scaling)**

Addressed in:

* **J.5 — Running Time and Token Cost (Llama-3-8B)**
* **J.6 — HCC Runtime for Co-Alignment (Llama-3-8B)**
* **K.2 & K.3 — Running Time, Token Cost, and HCC Runtime (ChatGPT-4o Mini)**
* **L.2 & L.3 — Token Usage, Running Time, and HCC Runtime (ChatGPT-3.5 Turbo)**

---

## **2. Reviewer VrB7 – Cross-Encoder Ablations & Two-Round Refinement**

### **(W1) Ablation across embedding models**

Addressed in:

* **I — Sensitivity Analysis of (K) for Specialized Models**
* **J.2 — Across Embedding Backbones (E5, GTE, BGE, BERT)**
* **J.3 — CAI Ratio Across Backbones**
* **K & L — Closed-Source LLM Ablations**

---

### **(Question) Why exactly two refinement rounds?**

Addressed in:

* **J.7 — Two-Round Correction and CAI Improvement (Llama-3-8B)**
* **K.4 — Two-Round Correction and CAI Improvement (ChatGPT-4o Mini)**
* **L.4 — Two-Round Correction and CAI Improvement (ChatGPT-3.5 Turbo)**

---

## **3. Reviewer QHKk – Weak Encoders, CAI Stability, and Computational Cost**

### **(W2) Does HCC work with weak or misaligned encoders?**

Addressed in:

* **J.1 — HCC Under Weaker (BERT) vs. Stronger (Llama-3-8B) Models**
* **J.2 & J.3 — Across Embedding Backbones and CAI Ratio Across Backbones**

---

### **(W3) Sensitivity to stronger encoders and potential gains**

Addressed in:

* **J.2 & J.3**

---

### **(W4) Computational cost**

Addressed in:

* **J.5 — Running Time and Token Cost (Llama-3-8B)**
* **J.6 — HCC Runtime for Co-Alignment (Llama-3-8B)**
* **K.2 & K.3 — Running Time, Token Cost, and HCC Runtime (ChatGPT-4o Mini)**
* **L.2 & L.3 — Token Usage, Running Time, and HCC Runtime (ChatGPT-3.5 Turbo)**

---

## **4. Reviewer MYtC – Notation, Objective Clarity, and Novelty Beyond Nearest-Neighbor**

### **(W1) Notation difficulty**

Addressed in:

* **A — Notation Summary**
* **S — Prompt Instructions (With/Without Preference Context)**

These sections clarify (\bar{y}) (zero-shot) and (\hat{y}) (collaborative single-shot) and their use in CAI/HCC.

---

### **(W2) Are we aligning labels or models?**

Addressed in:

* **F.1 — Reference-Free Evaluation Metric (CAI)**
* **H — Why LLM–Specialized Model Collaboration Is Essential**

---

### **(W3) HCC is not merely nearest-neighbor voting**

Addressed in:

* **G — Additional Experimental Details**
* **H.1 — Role of Specialized Model in HCC**
* **J.7 — Two-Round Co-Alignment**

---

### **(General) Real-world noise, scalability, domain flexibility**

Addressed in:

* **N — Overall Reflection of CAI**
* **O — Cross-Domain Generalization**

---

### Meta-Review · Area_Chair_Aun2 · 2026-01-11

**Summary:**

This submission proposes **Heterogeneous-Consistency Co-Alignment (HCC)**, a **training-free** pipeline for generating **preference-aligned / user-aligned annotations** over unlabeled corpora by combining (i) an LLM annotator and (ii) a lightweight “specialized” embedding-based model. The method (a) **verifies** annotation reliability using a reference-free agreement signal, the **Consistent-and-Inconsistent (CAI) ratio**, and (b) **co-aligns** inconsistent samples via a **two-round** refinement scheme that propagates user-seed labels using an embedding-based nearest-neighbor/majority-vote style mechanism over clustered data. Empirically, the authors report consistent gains across **8 NLU classification datasets** and multiple LLM families (open- and closed-source), and show CAI is strongly correlated with accuracy and can track pre/post co-alignment changes.

### Strengths
- **Clear, practical training-free pipeline**: forward-pass-only collaboration between an LLM and a specialized model; easy to deploy and reproduce (noted by Q1Gc, VrB7).
- **Reference-free quality signal**: CAI is a simple agreement-based metric that is empirically correlated with annotation accuracy and intended to be actionable in the absence of ground truth (Q1Gc, VrB7, QHKk).
- **Broad empirical evaluation**: results on 8 datasets and multiple LLMs; the rebuttal adds substantial **cross-encoder ablations** (E5/GTE/BGE/BERT) and details on **two-round refinement behavior**.

### Weaknesses
- **Scope of “preference alignment/personalization” remains narrow**: the setting is primarily categorical NLU with user seeds; the work does not evaluate pairwise preference learning (authors state out-of-scope), and thus broader phrasing about preference alignment must remain carefully scoped (raised by Q1Gc, QHKk, MYtC).
- **CAI limitations/failure modes**: while the rebuttal adds discussion and failure-case treatment, CAI is still an agreement proxy and may be dataset/encoder dependent; the practical thresholding story is mostly qualitative (raised by Q1Gc).

**Reviewer Concerns:**

### Addressed
- **Encoder/specialized model dependence & robustness** (Q1Gc, VrB7, QHKk): rebuttal adds extensive results across **E5/GTE/BGE/BERT** and shows HCC gains persist even with weaker encoders.
- **Sensitivity to K / hyperparameters** (Q1Gc): rebuttal provides **K ∈ {3,5,7,10}** sensitivity tables across datasets/backbones; variance appears limited, with K=3 often best.
- **Why two refinement rounds** (VrB7): rebuttal provides R1 vs R2 behavior and argues further rounds can saturate or regress.
- **Computational cost / latency** (Q1Gc, QHKk): rebuttal reports refinement runtime and token/time costs per 10k examples for zero-shot vs single-shot prompting.
- **Clarification on objective (label/annotation alignment vs model alignment)** (MYtC, QHKk): rebuttal explicitly frames the objective as **annotation/label propagation** for categorical NLU.

### Still outstanding / partially outstanding
- **CAI actionability/thresholding** (Q1Gc): rebuttal adds interpretation and failure-case discussion; however, concrete decision rules for “what CAI value is sufficient” remain somewhat heuristic and likely data-dependent.

**Reviewer Scores:**

All reviews are **marginally below threshold (4)** but *none* identify a concrete technical unsoundness; the concerns are primarily **missing ablations/analysis/clarity**, which the rebuttal substantially addresses (cross-encoder robustness, K sensitivity, two-round justification, cost/runtime reporting, CAI discussion). I would actually expect a couple of increases in their scores should there be a full discussion period.

---

### Decision · Program_Chairs · 2026-01-26

Accept (Poster)